# The vacuolar fusion regulated by HOPS complex promotes hyphal initiation and penetration in *Candida albicans*

Yu Liu[1,5], Ruina Wang[1,5], Jiacun Liu[1,5], Mengting Fan[2,5], Zi Ye[1], Yumeng Hao[1], Fei Xie[1], Ting Wang [1], Yuanying Jiang[3], Ningning Liu [4,6] ✉, Xiaoyan Cui [2,6] ✉, Quanzhen Lv [1,6] ✉ & Lan Yan [1,6] ✉

The transition between yeast and hyphae is crucial for regulating the commensalism and pathogenicity in *Candida albicans*. The mechanisms that affect the invasion of hyphae in solid media, whose deficiency is more related to the pathogenicity of *C. albicans*, have not been elucidated. Here, we found that the disruption of *VAM6* or *VPS41* which are components of the homotypic vacuolar fusion and protein sorting (HOPS) complex, or the Rab GTPase *YPT72*, all responsible for vacuole fusion, led to defects in hyphal growth in both liquid and solid media, but more pronounced on solid agar. The phenotypes of *vac8*Δ/Δ and *GTR1*^OE-*vam6*Δ/Δ mutants indicated that these deficiencies are mainly caused by the reduced mechanical forces that drive agar and organs penetration, and confirmed that large vacuoles are required for hyphal mechanical penetration. In summary, our study revealed that large vacuoles generated by vacuolar fusion support hyphal penetration and provided a perspective to refocus attention on the role of solid agar in evaluating *C. albicans* invasion.

As the fourth most common hospital bloodstream infection, candidemia occurs in organ transplant recipients, cancer patients, and patients with severe infections caused by peritoneal leakage, HIV or COVID-19[1,2]. The major cause is the opportunistic pathogen *Candida albicans*, which colonizes mucosal surfaces as a commensal. In immunocompromised patients, *C. albicans* can invade solid organs such as kidneys, liver, spleen, and bone marrow, with associated mortality of 10% to 47%[3]. Morphological transition, induced by nutrients, high temperature, neutral pH, and hypoxia, is critical for *C. albicans* adaption to the microenvironment[4,5]. Neither yeast-locked nor hyphae-locked *C. albicans* is fully virulent in models of systemic infections[6]. Since some virulence-defective mutants have a normal yeast-hyphal transition, there is still much to learn about the association between morphogenesis and virulence in *C. albicans*[7].

Early studies of the ability of yeast and hyphal form switching as a virulence factor focused on the pathogenicity of *C. albicans* mutants' defect in hyphal growth. The cyclic adenosine monophosphate/protein kinase A (cAMP/PKA), mitogen-activated protein kinases (MAPK), and pH pathways have been identified as the main signaling transduction cascades to regulate the yeast-hyphae transition. Downstream targets of these pathways are transcriptional activators including Efg1, Cph1, Tec1, Flo8, and Rim101, which result in hyphal-specific gene transcription. The cell cycle arrest Hgc1/Cdc28 pathway suppresses the degradation of the septin ring, promoting hyphal growth.

[1]Center for Basic Research and Innovation of Medicine and Pharmacy (MOE), School of Pharmacy, Naval Medical University, Shanghai 200433, PR China. [2]School of Chemistry and Molecular Engineering, East China Normal University, Shanghai 200241, PR China. [3]School of Medicine, Tongji University, Shanghai 200092, PR China. [4]State Key Laboratory of Systems Medicine for Cancer, Center for Single-Cell Omics, School of Public Health, Shanghai Jiao Tong University School of Medicine, Shanghai 200025, PR China. [5]These authors contributed equally: Yu Liu, Ruina Wang, Jiacun Liu, Mengting Fan. [6]These authors jointly supervised this work: Ningning Liu, Xiaoyan Cui, Quanzhen Lv, Lan Yan. ✉e-mail: fenghu704@163.com; xycui@chem.ecnu.edu.cn; lvquanzhen2011@163.com; ylansmmu@sina.com

Moreover, Tup1, Nrg1 and Rfg1 are repressors of filamentous growth[8–10]. A large number of mutant libraries were used to screen for deficiencies in hyphal growth. For instance, Noble et al. have identified 115 virulence-attenuated *C. albicans* mutants. Among which, 53 mutants showed defects in morphogenesis, 18 mutants had slow proliferation, and 46 mutants exhibited deficiency in infectivity without defects in morphogenesis or proliferation, indicating that virulence and morphogenesis are related in a complex, non-linear process[11]. In addition, many virulence factors associated with hyphal morphogenesis have been identified, such as GPI-anchored proteins, degradative enzymes, and cell surface adhesins, among which the identification of candidalysin (Ece1) was influential[12]. The invasive *ece1*Δ/Δ hyphae do not damage epithelia because the lack of candidalysin causes an inability to destroy cellular membranes[13,14].

In addition to studies on signal transduction and the function of virulence factors, mutants with different hyphae formation abilities in liquid and solid media have been of interest. The Blankenship group screened 124 *C. albicans* mutants in vitro for distinct filamentation induction conditions and concluded that filamentation in solid versus liquid media represent distinct biological and transcriptional programs[15]. Subsequent studies showed that morphogenesis on solid surfaces depends on several proteins, such as Cwt1, Tpk1, Dfi1, Cek1, and Dck1/Rac1[16–19]. These genes are required for invasive filamentous growth on agar but do not influence hyphae formation in liquid media. Notably, null mutants of some genes showed varying degrees of reduction in virulence in mouse models of systemic infection. For example, the virulence of the *cwt1* null mutant was not reduced, while the *dfi1* null mutant or the *cek1* null mutant was significantly attenuated. These studies indicated that the presence of hyphal defects on solid agar was more closely related with the decreased virulence in *C. albicans*. Morphogenesis on solid media is not only influenced by signal transduction but also related to hyphal penetration. Penetration of host organs or tissues is critical for *C. albicans* pathogenicity and requires adhesion factors and the generation of sufficient force[20]. Furthermore, the rigidity of hyphae is crucial for immune escape, as macrophages can damage *C. albicans* by folding hyphae[21]. For invading the epithelial cells, the endocytosis of host cells and the active penetration initiated by *C. albicans* are equally important[22,23]. Elastic moduli of host cells are ranged from 1 to 100 kPa[24]. Overcoming the mechanical force of the host cell membrane is critical for *C. albicans* penetration of the host cell and immune escape. Arkowitz and colleagues determined the threshold for *C. albicans* penetration to be about 200 kPa[25]. However, the source of the force that supports hyphal penetration is still not well defined.

In this study, we unexpectedly discovered that the ability of hyphal growth in the *vam6*Δ/Δ mutant was different in liquid and solid media. And further study indicates that HOPS-complex-mediated vacuolar fusion promotes hyphal initiation and penetration and thus is required for the invasion of *C. albicans* in systematic and skin infection models. We predict that targeting vacuolar fusion or vacuole integrity may be a potential strategy to reduce the pathogenicity in *C. albicans*.

## Results

### The loss of *VAM6* delays hyphal initiation in liquid medium and weakens hyphal formation on solid medium

Hyphae formation is necessary for *C. albicans* to invade host cells. While investigating the function of uncharacterized genes in *C. albicans*, we unexpectedly found a significant difference in the hyphal defects in the *vam6* null mutant in liquid and solid media. The disruption of *VAM6* was verified by genomic-specific PCR (Supplementary Fig. 1a, b). Then, we compared the hyphae in the wild-type *C. albicans* SC5314 and the *vam6*Δ/Δ mutant. In liquid hyphae-inducing medium, no matter RPMI1640 plus 10% fetal bovine serum (FBS) or Spider medium induced the hyphae formation in both SC5314 and the *vam6*Δ/Δ mutant. Moreover, calcofluor white (CFW) staining did not show obvious structural defects in hyphal branching and septum. In terms of hyphal

length, the disruption of *VAM6* resulted in shorter hyphae (Fig. 1a). After induction in liquid RPMI1640 plus 10% FBS or Spider medium for 4 h, the average length of hyphae in the *VAM6*-deficient mutant was 53.7 μm or 64.7 μm, in contrast to 86.25 μm or 103.2 μm in the wild type. The average hyphal length in the mutant was about 60% of that in the wild type SC5314 (Fig. 1b). Further statistics of hyphal compartments' number showed that either the first branching or the secondary branching in the *VAM6*-deficient mutant were less (Fig. 1c). The time-lapse analysis of hyphal length and induction time showed that the initiation of germ tube in the mutant was around 30 min, which was nearly 10 min later than the wild-type strain. The average elongation rate in the *VAM6*-disrupted mutant was 0.018 μm/min within 2 h of induction, slightly slower than that of 0.022 μm/min in the wild-type SC5314. In particular, with the extension of induction time, the elongation rate decreased in the *vam6* null mutant (Fig. 1d). To further examine the defects caused by the disruption of *VAM6*, we investigated the hyphae formation on solid media. As shown in Fig. 1e, the defects of hyphae growth in the *VAM6*-disrupted mutant on the solid media were more obvious than those in the liquid media. On solid RPMI1640 plus 10% FBS medium, the wild-type colony formed obvious hyphal extension around the smooth center. In contrast, the hyphal extensions in the *vam6*Δ/Δ mutant were much shorter and fewer. Similarly, on Spider medium, unlike the parental strain forming wrinkled colonies with hyphal extension around the colony, the *vam6*Δ/Δ mutant only formed wrinkles without hyphal extension. These results indicated that the *VAM6*-disrupted mutant possessed impaired hyphal production.

As hyphae are important components in maintaining the structure of *C. albicans* biofilms, we explored biofilm formation. Side-view and axial confocal projections revealed presence of hyphae in biofilms of the wild-type strain and the *vam6* null mutant. Strains at the base of the silicone produced hyphae rising to a dense zone, and longer hyphae above the dense zone. The long hyphae folded over on the surface of the biofilm. The depth of biofilm in the *vam6* null mutant was shorter than that in the wild type, which could be due to the shorter hyphae in the mutant (Fig. 1f). An XTT colorimetric assay showed that there was no significant difference in the number of living cells between the *VAM6*-deficient and wild-type biofilms (Fig. 1g). Our results indicated that the disruption of *VAM6* reduced the rate of hyphal elongation which affected the branching appearance and the length of hyphae. The more pronounced deficiency of hyphae formation on solid media in the *vam6* null mutant prompted us to speculate whether Vam6 plays a more significant role in maintaining hyphae penetration into agar.

### The fragmented vacuoles caused by the loss of HOPS complex impairs hyphal initiation and penetration

*VAM6* has been reported as a vacuole and mitochondria patch protein in *C. albicans*, which is crucial for autophagy[26]. Meanwhile, Vam6 (Vps39), Vps41 and Class C proteins form the HOPS complex, which bind the Rab GTPase Ypt7 via its subunits Vps39 and Vps41 in the tethering step of vacuolar fusion in *Saccharomyces cerevisiae* (Fig. 2a)[27]. Based on the hyphal phenotypes of mutants with fragmented vacuoles[28], we speculated that the lost function of HOPS complex caused by the disruption of *VAM6* may be the main reason for the hyphal penetration defects on solid media. To test this hypothesis, one of the components of HOPS complex, *VPS41*, and the Rab GTPase *YPT72* in *C. albicans* were knocked out, respectively; in addition, revertant strains were constructed (Supplementary Fig. 2a–f). The specific dye FM4-64 was used to detect the morphology of vacuoles. Our results showed that vacuoles in the *vam6*Δ/Δ, *vps41*Δ/Δ and *ypt72*Δ/Δ mutants exhibited fragmentation without any discernable normal vacuole structures, while one or more large oval vacuoles were present in the wild-type SC5314 and the revertant strains (Supplementary Fig. 2g). To further image the structure of the fragmented vacuoles, the probe VacuRed, a silicon-substituted coumarin-based

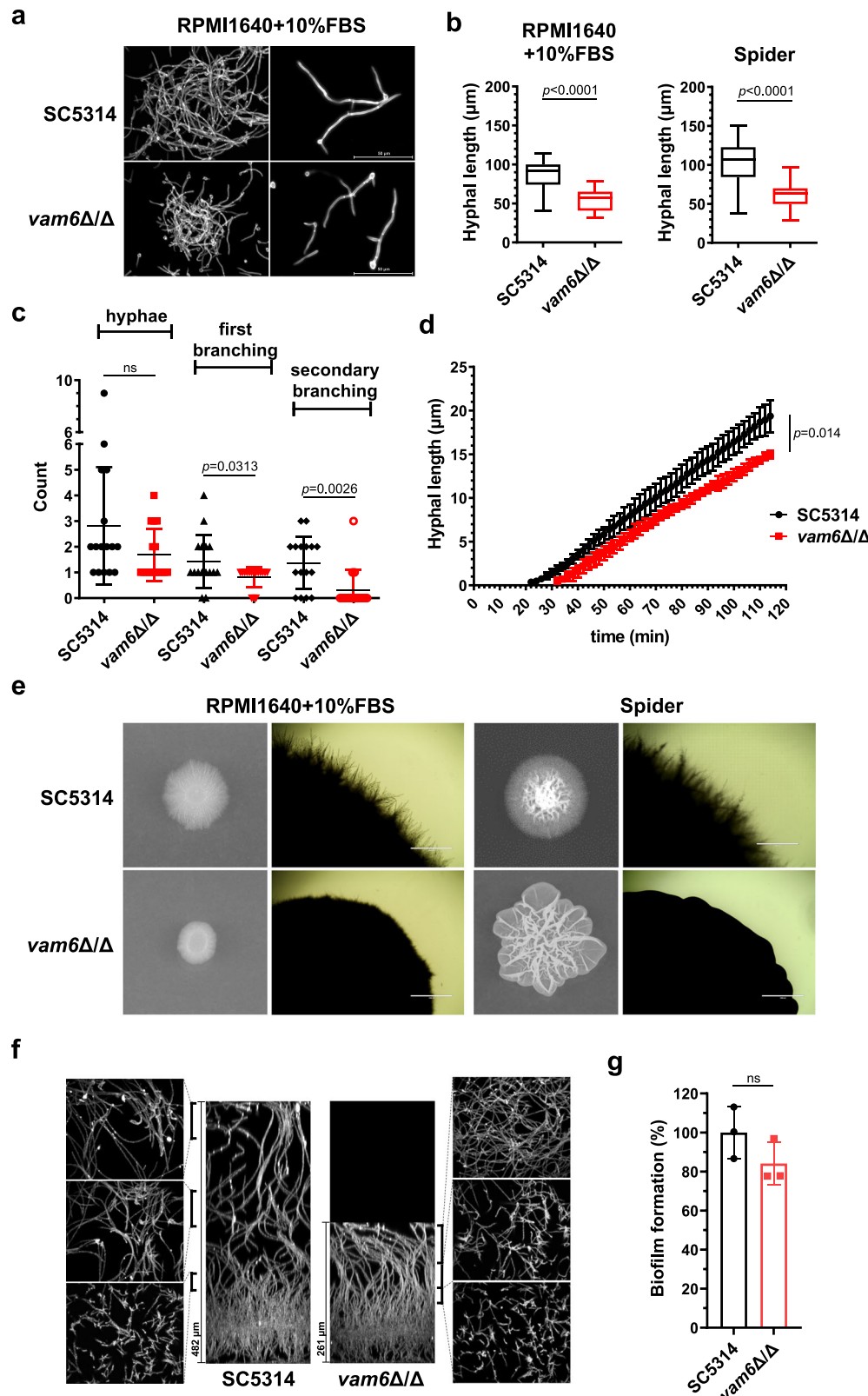

fluorophore developed by our group[29,30], targeting fungal vacuoles was designed and synthesized, which improved the imaging quality of small vesicles. VacuRed was explicitly enriched in yeast and hyphal vacuoles (Fig. 2b and Supplementary 2h). The large vacuoles were clearly observed in the wild-type yeast-form mother cells while budding or hyphal germinating. The volume of large vacuoles occupied almost all the entire cytoplasm. There were more fragmented vacuoles in the yeast-form daughter cells than those in the wild-type mother cells. Similarly, during germ tube extension, there were several vacuoles in the hyphal compartment and apical region. In contrast, the vacuoles in *vam6Δ/Δ*, *vps41Δ/Δ* or *ypt72Δ/Δ* mutants were even more fragmented. With the high resolution of the VacuRed probe and STED imaging, some small vesicles in the HOPS-complex-deficient mutants were clearly distinguished (Fig. 2c). In the wild-type yeast-form *C.*

**Fig. 1 | The disruption of *VAM6* results in defective filamentous growth in *C. albicans*.** *C. albicans* were cultured in RPMI1640 + 10% FBS or Spider medium at 37 °C for 4 h and stained with calcofluor white. **a** The hyphal morphology of *C. albicans* strains in liquid RPMI1640 + 10% FBS medium. Scale bars = 50 μm. **b** The hyphal length in the wild-type strain and *vam6*Δ/Δ mutant in RPMI1640 + 10% FBS or Spider medium. Box plots indicate median (middle line), 25th, 75th percentile (box), and minimum and maximum value as well as outliers (single points); *n* = 50 hyphae. **c** The number of clearly separated hyphae and their first and secondary branching in RPMI1640 + 10% FBS medium was counted. Data were presented as mean ± SD; *n* = 45 hyphae in SC5314, *n* = 27 hyphae in *vam6*Δ/Δ. **d** The dynamics of hyphal growth in *C. albicans* strains in liquid RPMI1640 + 10% FBS medium. The lengths of hyphal elongation were recorded for 120 min upon the initiation of hyphal induction. Images were collected for 50 time points, with one image every 2 min. Data were presented as mean ± SD; *n* = 6 hyphae. **e** The hyphal morphology of *C. albicans* strains on solid media. The representative images shown are from 3 biological repeats. Scale bars = 1000 μm. **f** *C. albicans* strains were assayed biofilm formation in RPMI1640 + 10% FBS at 37 °C for 24 h. The biofilms were fixed, stained with ConA Alexa Fluor 594, and clarified using the solvent exchange protocol into methyl salicylate. Axial and side-view projections of biofilms are shown. Scalebar indicates depth of the biofilms. **g** The viable strains in biofilms were quantified by XTT. Data were presented as mean ± SD; *n* = 3 samples. Comparison between the mutant and the wild-type strain was analyzed with mixed linear model calculated by SPSS (**d**), two-tailed unpaired *t*-test (**b**, **c**, **g**). Source data are provided as a Source Data file.

*albicans*, the large vacuoles were irregularly shaped and the inner diameter of 50 randomly selected large vacuoles was about 2782 ± 1259 nm. In contrast, the vacuoles in the *vam6*Δ/Δ, *vps41*Δ/Δ, and *ypt72*Δ/Δ mutants were mostly regular round with diameters of 436 ± 91 nm, 484 ± 112 nm, and 464 ± 99 nm, respectively. The STED imaging of vacuolar membrane suggested that the HOPS complex did not influence the integrity of the membrane. To further confirm the integrity of vacuolar membrane in the mutants, quinacrine, a dye that accumulates inside acidic vacuoles, was used to determine whether the compartments of the fragmented vacuoles were acidic and separated by membranes[31]. As shown in Fig. 2d and Supplementary Fig. 2i, the vacuolar membrane of the wild-type strain was stained by VacuRed, while green fluorescence of quinacrine was distributed within the vacuolar membrane. In the HOPS-complex-deficient mutants, the distribution of fluorescence of the red VacuRed and the green quinacrine of each fragmented vacuole did not be clearly resolved due to the limits of microscopic resolution. However, the obvious colocalization of the two probes indicated that the fragmented vacuolar compartments were also acidic and isolated from the cytoplasm by the vacuolar membranes. Transmission electron microscopy also showed that the refraction inside the vacuole was different from that of other organelles (Fig. 2e). Large vacuoles and intact membrane edges were seen in the wild-type *C. albicans*. In contrast, the vacuolar volume was significantly smaller in the HOPS-complex-disrupted mutants, whereas the vacuolar edge contour was still distinguishable.

To clarify the HOPS complex is primarily localized on the vacuolar membrane to mediate fusion, we constructed the GFP-Vam6, GFP-Vps41, and GFP-Ypt72 strains and performed fluorescence colocalization assays (Supplementary Fig. 2j–n). The colocalization of VacuRed with GFP-Vam6, GFP-Vps41, and GFP-Ypt72 (Fig. 2f), on the one hand, confirmed the specific distribution of VacuRed on the vacuolar membranes, and on the other hand, verified the colocalization of three target proteins on the vacuole membrane to mediate the fusion of vacuoles. To better observe the vacuolar fusion process, the Yvc1-GFP strain (NKF105) from *Yu* group, which is a highly abundant Ca²⁺ channel protein located on the vacuolar membrane, was used[32]. Unexpectedly, we found that Yvc1-GFP was accumulated at the vacuolar fusion site to form a bright region, which was used as a marker for vacuolar fusion (Fig. 2g). Real-time imaging revealed that the aggregated highlight of Yvc1-GFP disappeared once the process of vacuolar fusion was complete (Supplementary Video 1 and 2). Next, we disrupted *VAM6* gene in the Yvc1-GFP strain (Supplementary Fig. 2o, p). In Yvc1-GFP-*vam6*Δ/Δ mutant, green fluorescence exhibited as diffusion-like distribution. Real-time imaging for 10 min revealed the absence of localized aggregation regions, indicating the lack of vacuolar fusion process (Fig. 2g, Supplementary Video 3 and 4). Collectively, these results suggested that the HOPS complex mainly promoted the fusion of vesicles but did not affect the integrity of the vacuolar membrane.

Having established the function of *VAM6*, *VPS41*, and *YPT72* in vacuolar fusion, we then investigated whether the three mutants showed similar defects in hyphae formation. As expected, the disruption of *VPS41* or *YPT72* also resulted in approximately 40%

shortening of hyphae length in *C. albicans* in liquid media (Fig. 2h, i). While, complementation of *VAM6*, *VPS41*, and *YPT72* restored the hyphal length of the disrupted mutants, respectively (Supplementary Fig. 2q). The colony morphology of these HOPS-complex-deficient strains on solid hyphae-inducing media was similar. To describe the morphology of colonies in more detail, we defined the two distinct regions as the peripheral region and the central region. We measured the diameter of the central region (C) and determined the distance between hyphal insertion and the edge of the central region (P) (Fig. 2j). The quantitative analysis showed that the peripheral length of three mutant colonies was significantly shorter than that of the wild-type SC5314, which only accounted for about 15–30% on the RPMI1640 plus 10% FBS medium (Fig. 2k, l). To evaluate hyphal penetration ability, the colonies on solid agar were sectioned. The width and depth of colonies were labeled as W and D, respectively (Fig. 2m). When cultured on agar plates for 5 days, the penetration depth in the wild-type SC5314 was about 2.31 mm on RPMI1640 + 10% FBS agar and 5.23 mm on Spider agar (Fig. 2n). In contrast, the penetration depth of *vam6*Δ/Δ, *vps41*Δ/Δ and *ypt72*Δ/Δ mutants was only about 0.58 mm, 1.05 mm, and 1.35 mm on RPMI1640 + 10% FBS agar, and about 0.98 mm, 1.78 mm and 1.89 mm on Spider agar, respectively. According to the radiative morphology of invading hyphae, the penetration direction of the wild-type SC5314 was evenly distributed from horizontal 0° to vertical 90°. In contrast, most of the penetrated hyphae of HOPS-complex-deficient strains were vertical (90°). From the physical and mechanical analysis, due to the vertical downward effect of gravity, the penetrating force required for vertical penetration is less, which further reflected that the penetration capability of HOPS-complex-deficient mutants was significantly lower than that of the wild-type SC5314 (Fig. 2n). The *VAM6*, *VPS41*, and *YPT72* revertant strains showed similar hyphal morphology and invasion depth to those of the wild-type SC5314 on solid agar, which confirmed that the hyphal deficiencies were specifically caused by the disruption of the HOPS complex (Supplementary Fig. 2r). Collectively, the disruption of the HOPS complex resulted in reduced initiation, elongation and penetration of hyphae in *C. albicans*.

## HOPS complex is required for *C. albicans* to invade and damage the host

Impaired agar invasion may be associated with reduced virulence in *C. albicans*, as the active penetration is a mechanism of *C. albicans* invasion of epithelial cells[33]. Weakened penetration of hyphae in *C. albicans* might indicate the reduced damage to host cells and tissues, so the interaction of HOPS-complex-disrupted mutants with human umbilical vein endothelial cells (HUVECs) and peritoneal macrophages (PM) was evaluated[34]. We assayed the release of lactate dehydrogenase (LDH), which rapidly accumulates in the cell culture when the plasma membrane is damaged[35]. As shown in Fig. 3a, C. *albicans* SC5314 caused the highest LDH release rate when cultured with HUVECs for 6 h and 9 h. The LDH release rate caused by *vam6*Δ/Δ, *vps41*Δ/Δ or *ypt72*Δ/Δ mutants was about 15% lower than that of SC5314 in cells co-incubated for 9 h. Quantification of LDH release also showed

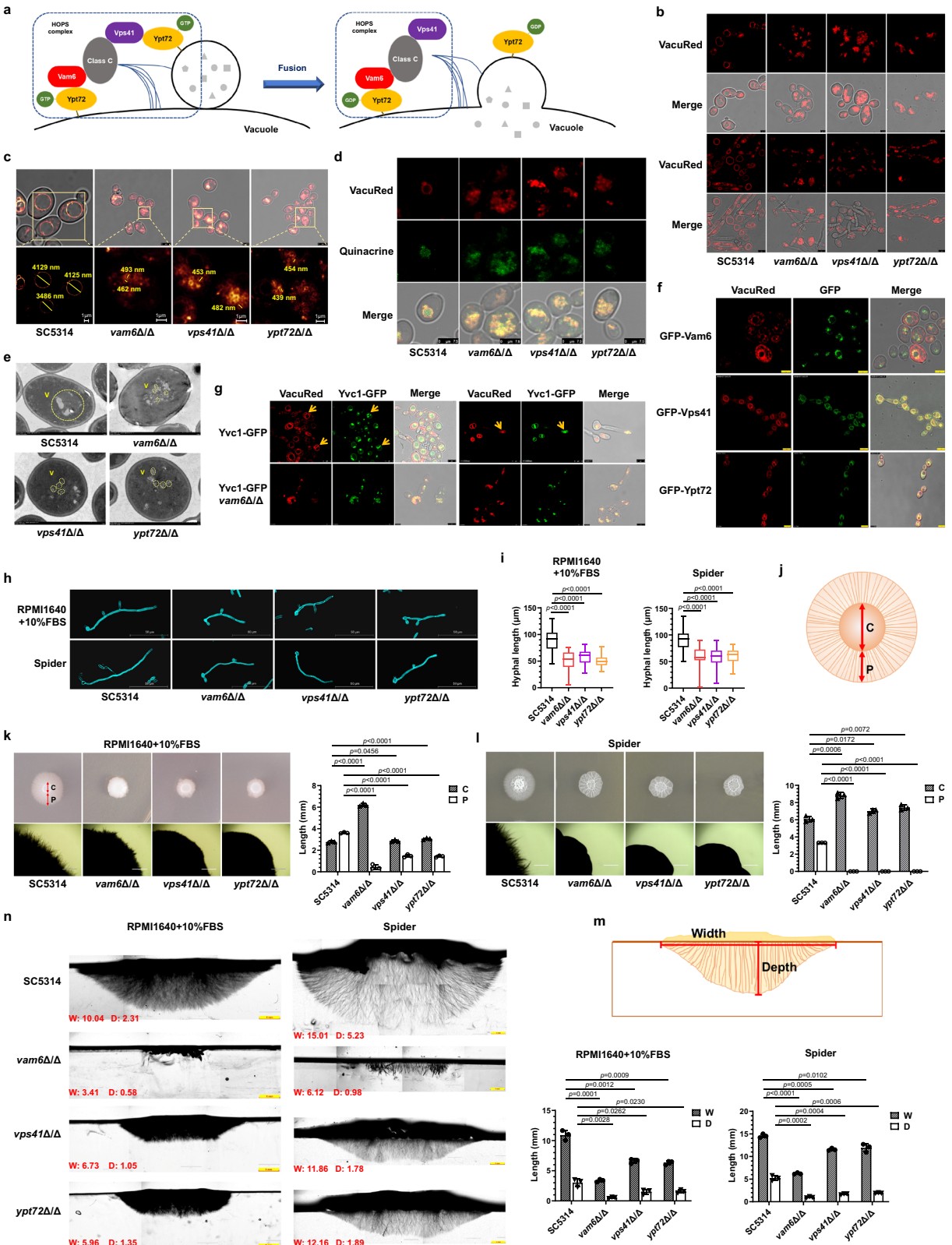

that the killing effect of HOPS-complex-disrupted mutants on HUVEC cells was weakened. In addition to invading epithelial cells, cell membrane penetration by hyphae is important to escape the killing of innate immune cells[36]. The fluorescence of propyl iodide (PI) in peritoneal macrophages co-incubated with *C. albicans* was used to investigate cell death. Most *C. albicans* strains were phagocyted after 3 h of co-incubation. Also, the hyphae in *C. albicans* SC5314 penetrating

macrophages were significantly much more and longer than those in *vam6*Δ/Δ, *vps41*Δ/Δ or *ypt72*Δ/Δ mutants (Fig. 3b). Correspondingly, the percentage of PI-positive macrophages was decreased in cells cultured with *vam6*Δ/Δ, *vps41*Δ/Δ or *ypt72*Δ/Δ mutants for 3 and 6 h, while the revertant mutants resulted in more dead macrophages similar to that of WT *C. albicans* (Fig. 3c and Supplementary Fig. 3a, b). These data suggested that the HOPS complex was also required for

Fig. 2 | The deficiency in HOPS complex abolishes vacuole fusion and impairs filamentous growth in *C. albicans*. **a** Identified process of HOPS-complex-mediated vacuole fusion in fungal cells. **b** Imaging of vacuole membrane stained by VacuRed. Scale bars = 5 μm. **c** Measurement of vacuolar diameters in *C. albicans*. Scale bars = 1 μm (lower panel). **d** Imaging of vacuole compartment stained by quinacrine. Scale bars = 7.5 μm. **e** Transmission electron microscope images in *C. albicans*. The yellow circles represent vacuoles (V). Scale bars = 1 μm. **f** The colocalization of Gfp-tagged Vam6, Vps41 and Ypt72 with VacuRed. Scale bars = 5 μm. **g** Imaging of GFP-tagged Yvc1 with VacuRed. The yellow arrows represent the site of fusion. Scale bars = 5 μm. **h** The hyphal morphology in *C. albicans*. Scale bars = 50 μm. **i** The hyphal length in *C. albicans*. Box plots indicate median (middle line), 25th, 75th percentile (box) and minimum and maximum value as well as outliers (single points). *n* = 50 hyphae. **j** The model of top view of hyphal colony on

solid agar. The diameter of center part without hyphae was defined as C. The distance of visible hyphal part scattering around the central region were defined as P. **k, l** The hyphal morphology on solid media. The C and P values were quantified by Image J. Data were presented as mean ± SD. *n* = 3 colonies. Scale bars = 1000 μm. **m** The model of vertical section of hyphal colony on solid agar. Width (W), the width of colony ranges under the surface of agar; Depth (D), the maximum distance of hyphae penetrating into agar from the surface of agar. **n** Vertical sections of hyphal colonies on solid media were observed by microscopy. The W and D values (mm) were quantified by Image J. Scale bars = 1 mm. Data were presented as mean ± SD. *n* = 3 colonies. Two-tailed unpaired *t*-test (**i, k, l, n**). The representative images shown are from 3 biological repeats (**b–g, k, l, n**). Source data are provided as a Source Data file.

maintenance the ability to kill macrophages. Overall, the HOPS complex participates in active penetration through host cells.

Subsequently, we evaluated the effect of the HOPS complex on *C. albicans* pathogenicity in murine models of systemic and cutaneous candidiasis. In the systemic infection models, *C. albicans* strains were injected through the tail vein. Mice injected with the wild-type SC5314 were all died within 11 days. However, almost all mice injected with *vam6Δ/Δ*, *vps41Δ/Δ* or *ypt72Δ/Δ* mutants were alive, with a survival rate of 90–100% within 21 days (Fig. 3d). As kidneys are the primary target organs in systemic candidiasis, decreased kidney function can also reflect the severity of *C. albicans* infections[37]. Our results showed that the levels of serum urea and creatinine in mice infected with *vam6Δ/Δ*, *vps41Δ/Δ* or *ypt72Δ/Δ* mutants were significantly lower than those in mice infected with SC5314, which indicated the importance of HOPS complex in causing kidney injury (Fig. 3e, f). Consistently, the fungal burden in kidneys in mice infected with *vam6Δ/Δ*, *vps41Δ/Δ* or *ypt72Δ/Δ* mutants for 5 days was significantly decreased (Fig. 3g). The periodic acid-Schiff (PAS) stained sections showed that the hyphal length in kidneys infected with *vam6Δ/Δ*, *vps41Δ/Δ* or *ypt72Δ/Δ* mutants was significantly shorter than that in kidney infected with the wild-type SC5314, which was consistent with the observed hyphal defects in vitro (Fig. 3h, i). The virulence of the revertant strains of *VAM6*, *VPS41*, and *YPT72*, respectively, was similar to that of the wild-type *C. albicans* in mice. All the mice infected with the revertant and the wild-type strains died within 9 days (Supplementary Fig. 3c). The fungal load in kidneys of mice infected with the revertant strains showed no significant difference in comparison with the mice infected with the wild-type *C. albicans* (Supplementary Fig. 3d), although the re-introduction of the three genes did not completely restore the hyphal length in mice (Supplementary Fig. 3e). These results confirmed that the hyphal penetration maintained by the HOPS complex is critical for *C. albicans* invasion to solid organs. A similar phenomenon was observed in a mouse model of *C. albicans* skin infection (Fig. 3j). Mice infected with *vam6Δ/Δ*, *vps41Δ/Δ* or *ypt72Δ/Δ* mutants for 5 days had less exfoliation and visible skin damage (Fig. 3k). Consistent with the skin lesions, mice infected with the HOPS-complex-deficient mutants had lower fungal load in the skin than that infected with wild-type *C. albicans* (Fig.3l). PAS staining showed that the length of hyphae invading the skin in mice infected with *vam6Δ/Δ*, *vps41Δ/Δ* or *ypt72Δ/Δ* mutants was shorter and the invasion depth also decreased, which indicated a more minor lesion and reduced skin damage (Fig. 3m, n). Similarly, the complement of *VAM6*, *VPS41*, and *YPT72* restored the damage, fungal load, hyphae length, and invasion, in the skin of mice infected with the revertant mutants (Supplementary Fig. 3f–i). Collectively, these results of infection models indicate that the HOPS complex is critical in maintaining hyphal invasion and injury to solid tissue.

## Inactivation of TOR signaling in HOPS-complex-deficient mutants results in delayed hyphae initiation

The disruption of *VAM6*, *VPS41*, or *YPT72* resulted in fragmented vacuoles and reduced hyphae initiation and penetration. To clarify the

main reason for the reduced virulence of HOPS-complex-deficient mutants, the secretion of some virulence factors and the activation of target of rapamycin (TOR) signaling pathways were examined. Fragmented vacuoles may influence the protein secretion, such as secreted aspartyl proteases (SAPs), which secretes simultaneously with yeast-to-hyphae transition and contributes to *C. albicans* virulence[38]. In our study, SAPs secretion was assayed on the YCB-BSA agar. The halo around of each colony was produced by the SAP-mediated hydrolysis of BSA. By measuring the size of each halo, we found that there was no significant difference in SAP secretion between the wild type, the HOPS-complex-deficient mutants, and the revertant strains (Supplementary Fig. 4a).

In addition to SAPs secretion, the activity of TOR pathway is also a central regulator of filamentation in *C. albicans*, which is dependent on a functional HOPS complex in *S. cerevisiae*[39,40]. At first, we examined the sensitivity of the *vam6Δ/Δ*, *vps41Δ/Δ*, and *ypt72Δ/Δ* mutants to rapamycin. Growth of each mutant was significantly reduced after the treatment with 3 nM or 5 nM rapamycin (Fig. 4a). And the complement of these three genes has fully restored the sensitivity to rapamycin (Supplementary Fig. 4b). Under nitrogen starvations, as shown in Fig. 4b, low dose of ammonium sulfate or arginine induced significantly decreased P-S6 signals in the *vam6Δ/Δ*, *vps41Δ/Δ*, and *ypt72Δ/Δ* mutants when compared with the wild-type SC5314, while the revertant mutants restored the increased activation of P-S6 (Supplementary Fig. 4c, d). These results indicated that TOR activity was weakened in *C. albicans* mutant strains with the defective HOPS complex. Since Tor1 can regulate the expression of adhesin genes, we then detected the transcription of genes encoding virulence factors, such as *ALS1*, *ALS3*, *ALS5*, *HWP1*, *ECE1*, *SAP2*, and *SAP6*[39]. The wild-type *C. albicans* and the HOPS-complex-deficient mutants were cultured in RPMI1640 plus 10% FBS medium with or without HUVECs (MOI = 1) for 1.5 h and 3 h. The relative expression of *ALS3*, *HWP1*, *ECE1* and *SAP6* were significantly up-regulated during incubation, either with or without the HUVECs, but there was no significant difference between the wild type and the HOPS-complex-deficient mutants (Supplementary Fig. 4e). Our results suggested that the reduced virulence in the HOPS-complex-deficient strains may not be caused by the decreased expression or secretion of virulence factors. Meanwhile, the impaired TOR activity did not reduce the normal proliferation of *C. albicans* (Supplementary Fig. 4f), indicating that the phenotypes of the HOPS-complex-deficient mutants were not caused by the decreased growth.

To determine whether the TOR pathway is involved in regulating the hyphae formation and penetration on solid agar, the effect of rapamycin on hyphal morphology was then evaluated. The hyphal length of *C. albicans* was decreased in dose-dependent manner in both liquid RPMI1640 plus 10% FBS or Spider medium supplemented with rapamycin (Supplementary Fig. 4g). As for the hyphae formed on solid agar in the wild-type SC5314, our results showed that rapamycin reduced significantly both the width of colony and the hyphal length embedded in solid medium (Fig. 4c). Interestingly, the hyphae penetrating into the agar were still radial rather than vertical, suggesting that

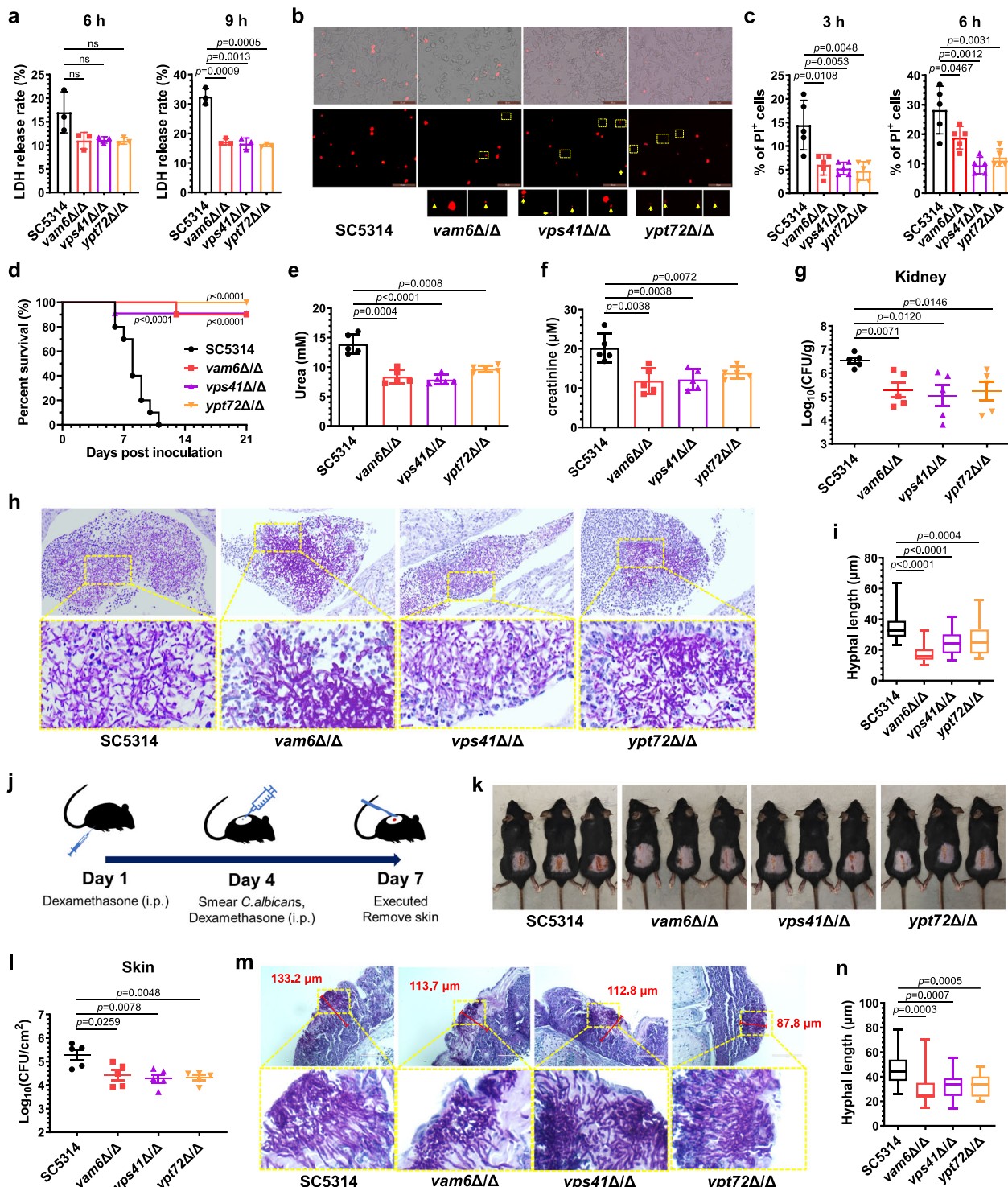

**Fig. 3 | The disruption of HOPS complex reduces pathogenesis in *C. albicans*.**
**a** The LDH release rate of HUVECs co-incubated with *C. albicans* (MOI = 1). Three biological replicates were carried out. *n* = 3 samples. **b** Murine peritoneal macrophages were co-incubated with *C. albicans* (MOI = 1) for 3 h, stained with propyl iodide (PI). The yellow arrows in the rectangle boxes zoomed in represent the nucleus of *C. albicans* which were not counted as PI-positive macrophages. Scale bars = 50 μm. **c** The percentage of PI-positive murine peritoneal macrophages cells co-incubated with strains. *n* = 5 fields. **d** The survival of mice infected with *C. albicans*. *n* = 10 mice. **e**, **f** The concentrations of urea (**e**) and creatinine (**f**) in the serum of mice were determined 5 days after fungal inoculation. *n* = 5 mice. **g** The fungal burden in kidneys of mice were determined 5 days after fungal inoculation. *n* = 5 mice. **h** The kidneys of mice were taken out 5 days after fungal inoculation and

made pathological sections with PAS staining. Scale bars = 20 μm. **i** The hyphal length in kidneys was measured by Image J. *n* = 30 hyphae. **j** The experiment process for cutaneous candidiasis model. **k** The exposed back skin with fungal-infected damage on day 7. **l** The fungal burden in damaged skin of mice were determined 3 days after fungal inoculation. *n* = 5 mice. **m** The pathological images of infected skin stained by PAS. The dimension of skin damage was quantified by the maximum depth of penetrated hyphae measured by Image J. Scale bars = 100 μm. **n** The hyphal length in damaged skin was measured by Image J. *n* = 20 hyphae. Data were presented as mean ± SD (**a**, **c**, **e**, **f**, **g**, **l**). Box plots indicate median (middle line), 25th, 75th percentile (box), and minimum and maximum value as well as outliers (single points) (**i**, **n**). Log-rank (Mantel-Cox) test (**d**), two-tailed unpaired *t*-test (**a**, **c**, **e**, **f**, **g**, **i**, **l**, **n**). Source data are provided as a Source Data file.

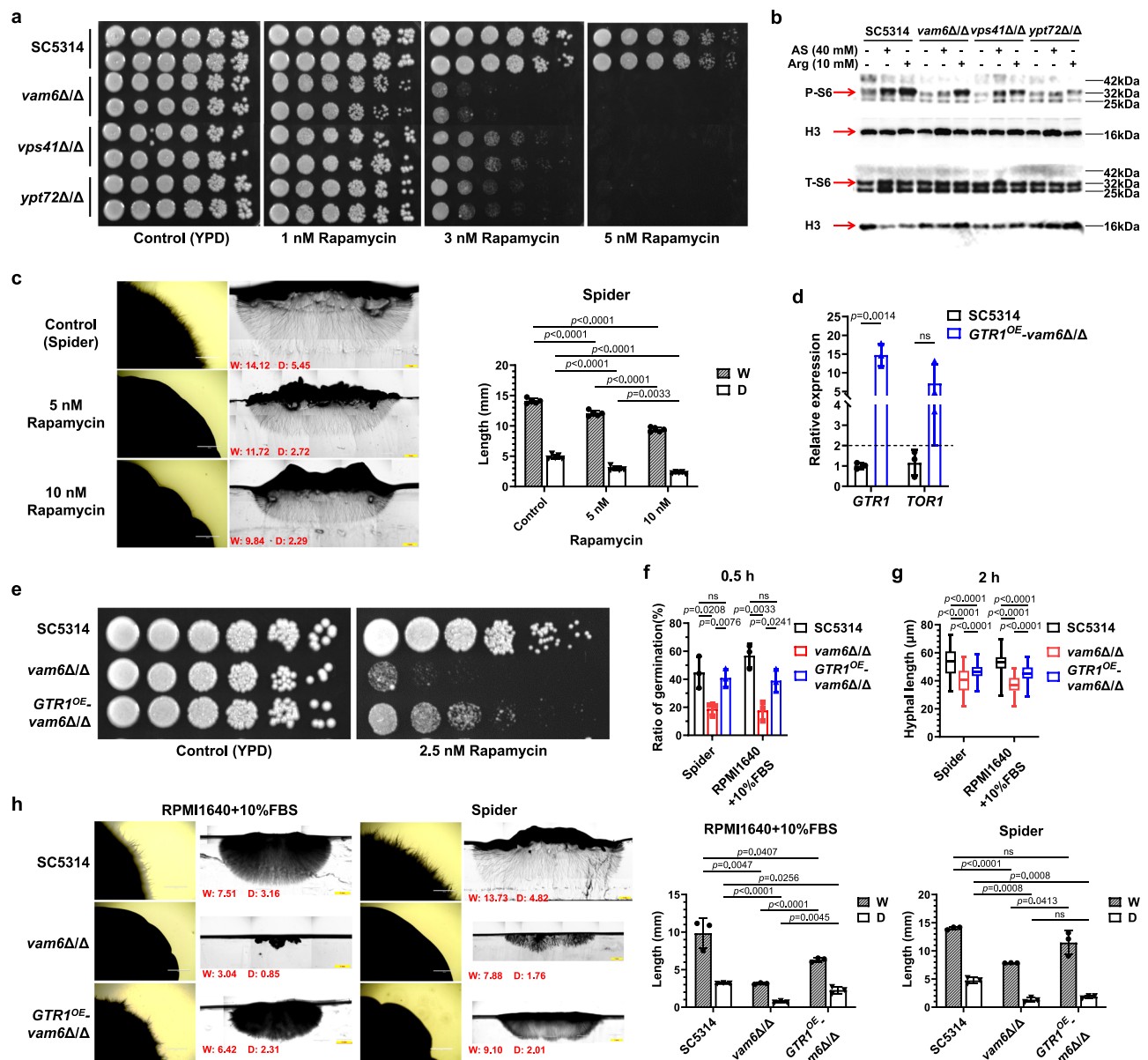

**Fig. 4 | The HOPS complex is required for the activation of TOR pathway.**
**a** Rapamycin sensitivity examined by spot assay. *C. albicans* strains were serial 5-fold diluted, spotted on YPD agar containing rapamycin and cultured at 30 °C for 48 h. **b** The phosphorylation of RPS6 protein (P-S6) examined by western blotting. The representative images shown are from 3 biological repeats. **c** The hyphal morphology of *C. albicans* strains on solid medium. Three biological replicates were carried out. One of the representative images are shown. Width (W) and depth (D) (mm) were quantified by Image J. *n* = 5 colonies. Scale bars = 1 mm. **d** Relative expression of *GTR1* and *TOR1* in *GTR1*^OE-*vam6*Δ/Δ mutants compared to SC5314 examined by quantitative real-time RT-PCR. *n* = 3 samples. **e** Rapamycin sensitivity examined by spot assay. **f** The ratio of germination in *C. albicans* cultured in hyphal-

inducing liquid media at 37 °C for 0.5 h. The ratio of germinated strains to all the strains observed in the microscopic field (approximately 50–100 fungal cells in each field under the 40× objective) was considered as the ratio of germination. *n* = 3 fields. **g** The hyphal length quantified by Image J. *n* = 50 hyphae. **h** The hyphal morphology of *C. albicans* strains on solid medium. Three biological replicates were carried out. One of the representative images are shown. Width (W) and depth (D) (mm) were quantified by Image J. Scale bars = 1 mm. *n* = 3 colonies. Data were presented as mean ± SD (**c, d, f, h**). Box plots indicate median (middle line), 25th, 75th percentile (box) and minimum and maximum value as well as outliers (single points) (**g**). Two-tailed unpaired *t*-test (**c, d, f, g, h**). Source data are provided as a Source Data file.

that decreased TOR signaling activity is not the main reason for impaired hyphal penetration in the *vam6*Δ/Δ, *vps41*Δ/Δ, and *ypt72*Δ/Δ mutants.

Overexpression of the small GTPase Gtr1, a subunit of the TORC1-activating EGO complex, has been shown to restore TOR activity in *C. albicans*[41,42]. To further clarify the effect of TOR signaling in hyphal initiation and penetration, the *GTR1* gene was overexpressed in the *vam6*Δ/Δ mutant. The *GTR1*^OE-*vam6*Δ/Δ mutant was confirmed by the genomic PCR, the high expression of *GTR1* detected by RT-qPCR, and increased resistance to rapamycin (Supplementary Fig. 4h, Fig. 4d, e).

Our results showed that overexpression of *GTR1* restored the delayed germination caused by the disruption of *VAM6* in liquid medium. Either in RPMI1640 plus 10% FBS or Spider medium, there was no significant difference in the proportion of strains formed germ tubes between *GTR1*^OE-*vam6*Δ/Δ mutant and the wild-type strain, which suggested that the signaling transduction of hyphae initiation was recovered (Fig. 4f). Moreover, overexpression of *GTR1* in the *vam6*Δ/Δ mutant promoted the hyphal elongation through measuring the average hyphal length after incubation for 2 h (Fig. 4g). Subsequently, we investigated whether *GTR1* overexpression could recover the

hyphal deficiency in the *vam6Δ/Δ* mutant on solid agar. As shown in Fig. 4h, the *GTR1*[OE]-*vam6Δ/Δ* mutant did not deeply embed into the agar or form a radial hyphal distribution similar to the wild-type SC5314, either on RPMI1640 plus 10% FBS or Spider agar. On the contrary, the *GTR1*[OE]-*vam6Δ/Δ* mutant had a similar colony morphology to the *vam6Δ/Δ* mutant, which indicated that *GTR1* overexpression did not restore the penetration in *C. albicans* to solid agar. Overall, our results suggest that overexpression of *GTR1* restores the TOR activity of the *vam6Δ/Δ* mutant during the hyphal initiation but did not promote hyphal penetration into the agar.

## Vacuole fusion supports hyphal mechanical penetration into the solid agar

The phenotypes of *GTR1*[OE]-*vam6Δ/Δ* mutant indicated that the slightly delayed hyphal initiation in HOPS-complex-deficient mutants is not the main reason for the obvious penetrating defects on solid agar. Therefore, the morphological integrity of large vacuoles maintained by vacuole fusion may be more critical, possibly by changing the compartment size to provide hyphal penetration forces. As shown in Fig. 5a, vacuoles stained with VacuRed probes indicated the presence of multiple large vacuoles in hyphae induced in either liquid or solid media. In particular, the morphology of vacuoles located in the hyphal tips on solid media is significantly different from that in liquid medium. In hyphal cells scraped from solid RPMI1640 plus 10% of FBS medium, the hyphae tips were enlarged and elliptic in shape. The interior of the hyphal tip was filled with a large vacuole that almost occupies the leading septum, which suggested that the large vacuole may be required for the expansion of hyphal tips. In order to accurately discern the presence of large vacuoles localized at the tips of hyphae, the vacuolar morphology in Yvc1-GFP strains in the agar sections was observed. Consistent with the hyphae scraped from agar, the tips of some but not all elongated hyphae were distributed with large vacuoles (Fig. 5b). When the Yvc1-GFP strains were incubated with HUVECs, some small dense vacuoles (solid globules in fluorescence) were accumulated at the tips of hyphae, but they did not completely fuse into hollow large vacuoles (Supplementary Fig. 5a). This result indicates that hyphae can penetrate the cell membrane without large hollow vacuoles in the liquid medium. Compared with the wild-type *C. albicans*, the vacuoles in the HOPS-complex-deficient mutants scraped from the agar still showed fragmentation without obvious large vacuoles (Fig. 5c and Supplementary Fig. 5b). To verify whether the large vacuoles in the tips were enlarged by vacuole fusion, the dynamic changes of vacuoles in hyphae were observed. Using live cell imaging, we captured vacuolar fusion in hyphae. Especially in the hyphal tip, two vacuoles fused to form a larger one in SC5314 (Fig. 5d), and fusing vacuoles were also observed in the hyphal tips in the Yvc1-GFP strains (Supplementary Video 1 and 2). In contrast, there was no fusion of the fragmented vacuoles in the *vam6Δ/Δ*, *vps41Δ/Δ* and *ypt72Δ/Δ* mutants (Supplementary Video 5, 6, and 7). These results suggested that large vacuoles produced by fusion in hyphae may play an important role in promoting hyphal penetration to solid agar.

The disruption of the HOPS complex caused both delayed hyphal initiation and fragmented vacuoles. To exclude the effects of delayed hyphal initiation on the hyphal penetrating into agar, we tried various hyphae-inducing media to activate different intracellular signaling pathways to induce hyphal initiation in *vam6Δ/Δ*, *vps41Δ/Δ*, and *ypt72Δ/Δ* mutants. Fortunately, we found that the addition of 10 mM or 50 mM of calcium chloride in Spider medium significantly promoted hyphae formation in *vam6Δ/Δ*, *vps41Δ/Δ* and *ypt72Δ/Δ* mutants on solid agar (Supplementary Fig. 5c). Moreover, at the early stage of hyphal induction (24 h and 48 h), *vam6Δ/Δ*, *vps41Δ/Δ* and *ypt72Δ/Δ* mutants formed burr invasion to the agar containing calcium with longer hyphae (reflected by the parameter D), in comparison with the hyphae on the agar without calcium, which indicated that calcium would effectively promote the hyphal initiation in the HOPS-complex-

deficient mutants. But, from the morphologies of colonies induced for 5 days, the hyphal invasion depth in *vam6Δ/Δ*, *vps41Δ/Δ* and *ypt72Δ/Δ* mutants into agar was significantly shorter than that in the wild-type SC5314, as the depth of invasion was only about 1.79 mm, 1.82 mm and 2.00 mm, and was similar to that of 1.69 mm, 1.74 mm and 2.19 mm in the agar without calcium (Fig. 5e, f). This result indicated that although calcium restored the initiation of hyphae growth induced for 48 h, it did not restore the penetration depth of HOPS-complex-deficient mutants into 2% agar. To further verify the defect in HOPS-complex-deficient mutants on solid medium resulting from the insufficient mechanical force of hyphae, we investigated the influence of different concentrations of agar on the invasive zones, as the invasion zone was decreased with the increase of agar concentration[43]. As shown in Fig. 5g, we adjusted the agar concentrations to 0.5%, 1%, and 2% in Spider medium with 10 mM calcium, respectively. With the concentrations of agar increased, the invading hyphae of all strains were sparser, and the length of hyphae around the colony was slightly reduced. The peripheral region (P) and central region (C) value of different colonies were shown in Supplementary Fig. 5d and the ratios of peripheral region (P) of HOPS-complex-disrupted mutants to that of the wild-type *C. albicans* at the same agar concentration were shown in Fig. 5h. In *vam6Δ/Δ*, *vps41Δ/Δ*, and *ypt72Δ/Δ* mutants, the deficiency of hyphal invasion in 2% agar is significantly larger than that in 0.5% agar. Similar inverse relationship between the length of hyphal extension and agar concentration was further demonstrated in RPMI1640 plus 10% FBS medium supplemented with 0.5% to 2% agar. The hyphal invasion ratio in *vam6Δ/Δ*, *vps41Δ/Δ*, and *ypt72Δ/Δ* mutants had greater deficiencies in 2% agar than in 0.5% or 1% agar (Fig. 5i, j, Supplementary 5e). Penetration of solid medium with higher concentrations of agar needs stronger mechanical force of hyphae. Therefore, the reduced peripheral region suggested that the HOPS-complex-mediated vacuole fusion is necessary to maintain the hyphal mechanical force. Thus, our results suggest that fragmented vacuoles may result in reduced mechanical forces supporting hyphal penetration in *C. albicans*.

## Hyphal mechanical penetration is required for *C. albicans* pathogenicity in vivo

The disruption of the HOPS-complex delayed hyphal initiation and weakened the mechanical support of large vacuoles. Next, we examined which of them had a greater effect on the pathogenicity in *C. albicans*. As shown in Fig. 6a, the overexpression of *GTR1* failed to recover the vacuole fusion caused by *VAM6* disruption, and no large vacuoles were observed in *GTR1*[OE]-*vam6Δ/Δ* mutant. The hyphae initiation and hyphal length of *GTR1*[OE]-*vam6Δ/Δ* in liquid medium was restored (Fig. 4f, g), but the ability to penetrate agar was still defective (Fig. 4h). So, the pathogenicity of *vam6Δ/Δ* and *GTR1*[OE]-*vam6Δ/Δ* mutants was compared. The co-incubation of the wild-type SC5314, *vam6Δ/Δ* or *GTR1*[OE]-*vam6Δ/Δ* mutant with HUVECs showed that the overexpression of *GTR1* almost completely recovered the invasion and killing effect of the *vam6Δ/Δ* mutants, as the proportion of PI-positive cells and the release of LDH caused by *GTR1*[OE]-*vam6Δ/Δ* mutant was similar to those caused by SC5314 (Fig. 6b, c). However, in the model of systemic *C. albicans* infection, mice infected with either *GTR1*[OE]-*vam6Δ/Δ* or *vam6Δ/Δ* mutant exhibited higher survival rates and decreased fungal burdens in kidneys compared to the wild-type group. No significant difference was observed among mice infected with *GTR1*[OE]-*vam6Δ/Δ* or *vam6Δ/Δ* mutants (Fig. 6d, e). Consistently, kidney sections stained by PAS showed that *GTR1*[OE]-*vam6Δ/Δ* and *vam6Δ/Δ* mutants were unable to form elongated radial hyphae, and the extent of kidney damage caused by the mutants was significantly smaller than that caused by the wild-type SC5314 (Fig. 6f, g). In the mouse model of skin infection, *GTR1*[OE]-*vam6Δ/Δ* mutant did not cause more severe skin damage or deeper tissue invasion compared to the *vam6Δ/Δ* mutant (Fig. 6h). Mice infected with *vam6Δ/Δ* and *GTR1*[OE]-*vam6Δ/Δ* had similar

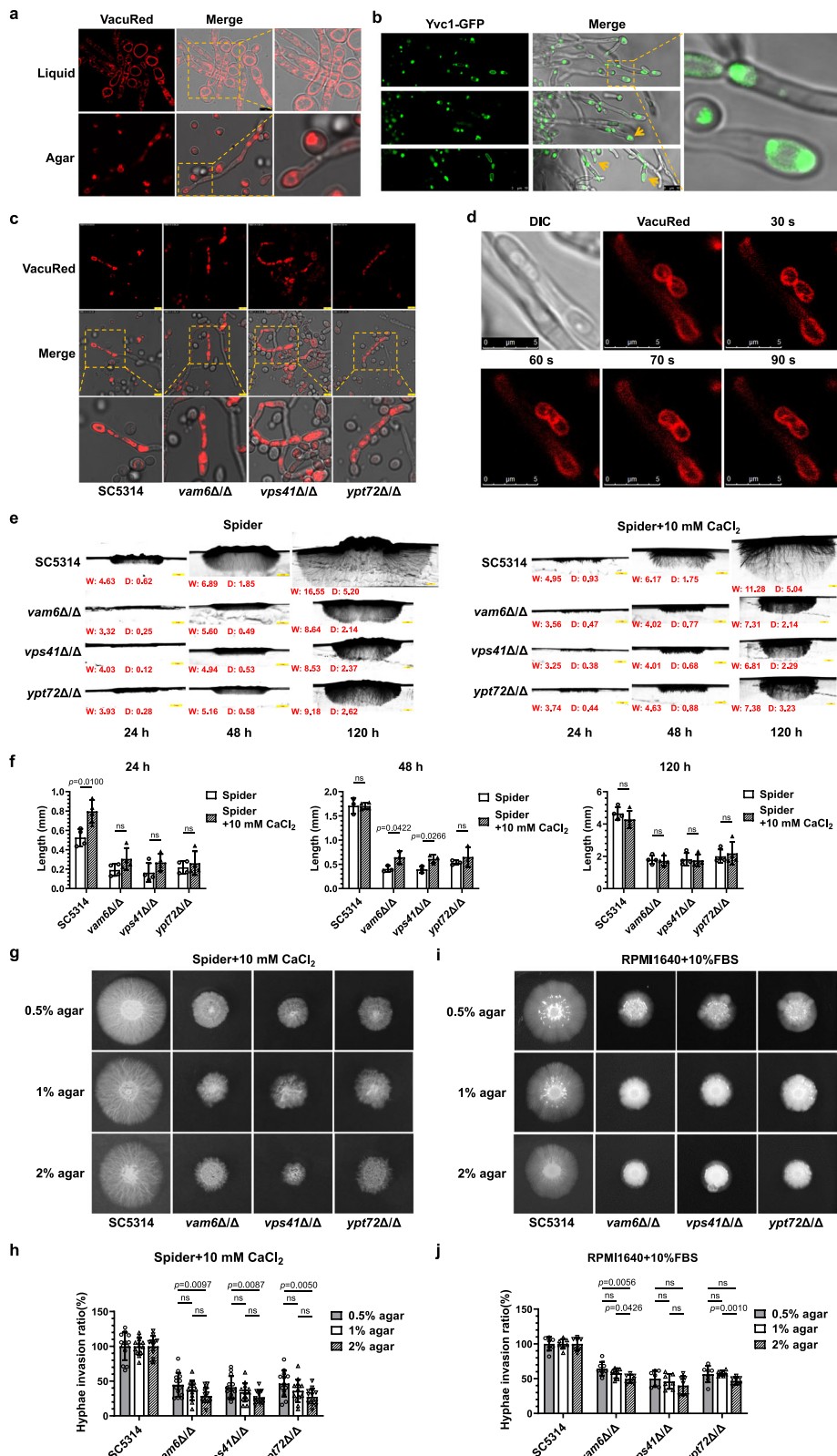

fungal load and hyphae length in skin, but both were lower than those of mice infected with the wild-type *C. albicans* (Fig. 6i–k). Overall, our results showed that the overexpression of *GTR1* partially restored the mammalian cell killing in the *vam6*Δ/Δ mutant in vitro. However, this overexpression did not enhance the invasion of solid organs or alleviate the reduced virulence caused by the disruption of *VAM6* in vivo.

It is reported that the *vac8* null mutants have deficiencies in the movement of vacuoles from mother cells to daughter cells[44]. The disruption of *VAC8* resulted in depleted vacuoles in the emerging yeast buds and smaller vacuoles in hyphae induced in liquid serum[45]. Thus, we hypothesized that the deletion of *VAC8* would lead to fewer and smaller vacuoles in hyphae and hyphae tips, which in turn reduce hyphal penetration and virulence. To further confirm the important

**Fig. 5 | The deficiency in HOPS complex leads to reduced mechanical penetration of hyphae. a** Vacuoles in hyphae of *C. albicans* SC5314 cultured in liquid RPMI1640 + 10% FBS medium for 90 min and solid Spider media for 48 h. Scale bars = 5 μm. **b** Imaging of Yvc1-GFP strains in hyphae sliced from Spider agar. Scale bars = 10 μm. **c** Vacuoles stained by VacuRed in hyphae scraped from Spider agar. Scale bars = 5 μm. **d** The fusion process of vacuoles during hyphal growth in *C. albicans*. SC5314 were cultured in liquid RPMI1640 + 10% FBS medium at 37 °C for 1 h, and stained with VacuRed. The vacuoles in hyphae were observed every 30 s. A 90 s time period is shown. Scale bars = 5 μm. The representative images shown are from 3 biological repeats (**a**–**d**). **e**, **f** Vertical sections of hyphal colonies on solid agar. Width (W) and depth (D) (mm) were quantified by Image J. *n* = 4 colonies.

Scale bars = 1 mm. **g** Top view of hyphal colony on Spider + 10 mM CaCl₂ solid plates containing different concentrations of agar. **h** The hyphae invasion ratio calculated by P$_{HOPS-complex-deficient\ mutants}$/P$_{SC5314}$ in the same concentrations of agar. Parameters C and P of hyphal colonies in (**g**) were measured by Image J. *n* = 13 colonies. **i** Top view of hyphal colony on RPMI1640 + 10% FBS solid plates containing different concentrations of agar. Three biological replicates were carried out. One of the representative images are shown (**e**, **g**, **i**). **j** The hyphae invasion ratio calculated by P$_{HOPS-complex-deficient\ mutants}$/P$_{SC5314}$ in the same concentrations of agar. Parameters C and P of hyphal colonies in **i** were measured by Image J. *n* = 7 colonies. Data were presented as mean ± SD (**h**, **j**). Two-tailed unpaired *t*-test (**f**, **h**, **j**). Source data are provided as a Source Data file.

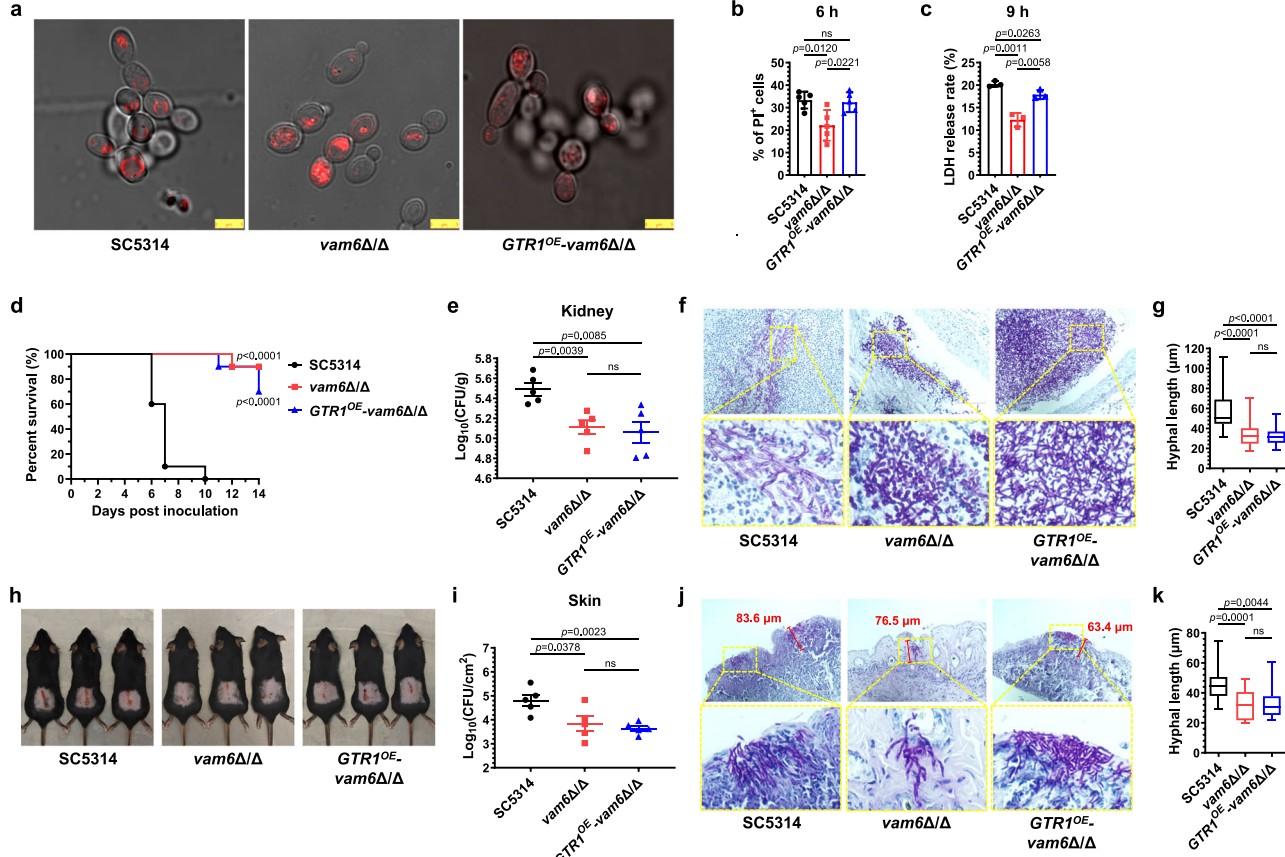

**Fig. 6 | Hyphal penetration forces supported by large vacuoles are required for the virulence in *C. albicans*. a** Imaging of vacuole membrane stained by VacuRed in *C. albicans* cultured in YPD medium at 30 °C for 90 min. Three biological replicates were carried out. One of the representative images are shown. Scale bars = 5 μm. **b** The percentage of propyl iodide (PI)-positive murine peritoneal macrophages co-incubated with *C. albicans* for 6 h (MOI = 1). *n* = 5 fields. **c** The LDH release rate of HUVECs co-incubated with *C. albicans* for 9 h (MOI = 1). *n* = 3 samples. **d** The survival percentage of mice infected with *C. albicans*. *n* = 10 mice. **e** The fungal burden in kidneys of mice were determined 5 days after fungal inoculation. *n* = 5 mice. **f** The kidneys of mice were taken out 5 days after fungal inoculation and made pathological sections with PAS staining. Scale bars = 100 μm.

**g** The hyphal length in kidneys was measured by Image J. *n* = 20 hyphae. **h** The exposed back skin with fungal-infected damage on day 7. **i** The fungal burdens in damaged skin of mice were determined 3 days after fungal inoculation. *n* = 5 mice. **j** The pathological images of infected skin stained by PAS. The dimension of skin damage was quantified by the maximum depth of penetrated hyphae measured by Image J. Scale bars = 100 μm. **k** The hyphal length in damaged skin was measured by Image J. *n* = 20 hyphae. Data were presented as mean ± SD (**b**, **c**, **e**, **i**). Box plots indicate median (middle line), 25th, 75th percentile (box), and minimum and maximum values as well as outliers (single points) (**g**, **k**). Log-rank (Mantel-Cox) test (**d**), two-tailed unpaired *t*-test (**b**, **c**, **e**, **g**, **i**, **k**). Source data are provided as a Source Data file.

role of large vacuoles, the disruption of *VAC8* was constructed in our study (Supplementary Fig. 6a, b). As show in Fig. 7a, the hyphal germination rate of *vac8*Δ/Δ mutants was lower than that of wild-type *C. albicans* after a 40-min induction period, similar to that observed in the *vam6*Δ/Δ mutants. However, after being induced for 4 h, the length of hyphae formed by the *vac8*Δ/Δ mutants was longer compared to that in the *vam6*Δ/Δ mutants (Fig. 7b, c). For agar penetration, the initial behavior of *vac8*Δ/Δ mutants on solid agar resembled that of *vam6*Δ/Δ. After 48 h of cultivation, the invaded depth of *vac8*Δ/Δ and *vam6*Δ/Δ was significantly smaller than that of the wild-type

*C. albicans*. But to our surprise, when the cultivation time extended to 5 days, the *vac8*Δ/Δ mutant exhibited a significantly greater penetration depth compared to the *vam6*Δ/Δ, indicating that mechanical penetration of *vac8*Δ/Δ was not significantly impaired (Fig. 7d, e).

The vacuolar morphology in the *vac8* null mutant were then explored. In consistence with the previous studies, in the hyphae induced for 1.5 h in liquid RPMI1640 plus 10%FBS medium, the disruption of *VAC8* resulted in an augmentation of vacuole volume in yeast cells, while both the number and volume of vacuoles decreased in the hyphal tubes. However, when the hyphae were induced for 4 h,

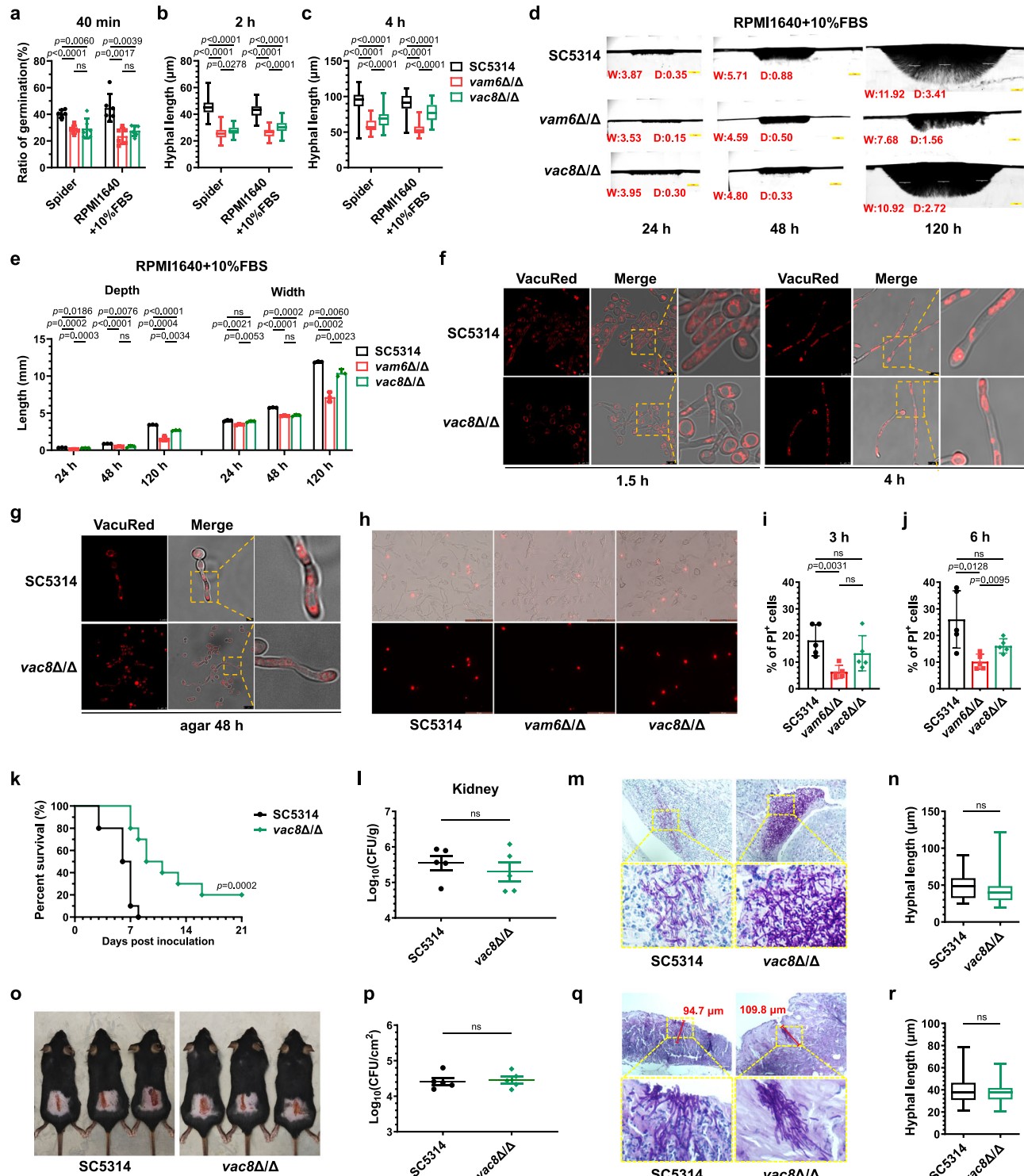

**Fig. 7 | The *vac8* null mutant with large vacuoles showed normal ability to penetrate solid media and slightly decreased virulence. a** The ratio of germination in *C. albicans* induced for 40 min. *n* = 6 fields. **b, c** The hyphal length induced for 2 h and 4 h. *n* = 50 hyphae. **d, e** Vertical sections of hyphal colonies on solid agar. Colonel width (W) and depth (D) (mm) were quantified by Image J. *n* = 3 colonies. Scale bars = 1 mm. **f, g** Vacuoles in hyphae cultured in liquid RPMI1640 + 10% FBS or solid Spider medium. Scale bars = 7.5 μm in (**f**), 2.5 μm (upper), and 10 μm (lower) in (**g**). The representative images shown are from 3 biological replicates (**d, f, g**). **h** Murine peritoneal macrophages were co-incubated with *C. albicans* (MOI = 1) for 3 h and stained with PI. Scale bars = 50 μm. **i, j** The percentage of PI-positive murine macrophages co-incubated with fungi (MOI = 1). *n* = 5 fields. **k** The survival percentage of mice infected with *C. albicans*. *n* = 10 mice. **l** The fungal burden in

kidneys. *n* = 5 mice. **m** The kidneys of mice were made pathological sections with PAS staining. Scale bars = 100 μm. **n** The hyphal length in kidneys was measured by Image J. *n* = 35 hyphae. **o** The exposed back skin with fungal-infected damage on day 7. **p** The fungal burden in skin of mice. *n* = 5 mice. **q** The pathological images of infected skin stained by PAS. The dimension of skin damage was quantified by the maximum depth of penetrated hyphae measured by Image J. Scale bars = 100 μm. **r** The hyphal length in damaged skin was measured by Image J. *n* = 25 hyphae. Data were presented as mean ± SD (**a, e, i, j, l, p**). Box plots indicate median (middle line), 25th, 75th percentile (box), and minimum and maximum value as well as outliers (single points) (**b, c, n, r**). Log-rank (Mantel-Cox) test (**k**), two-tailed unpaired *t*-test (**a–c, e, i, j, l, n, p, r**). Source data are provided as a Source Data file.

there were no significant differences observed in the number or size of hyphal vacuoles between the *vac8Δ/Δ* mutants and the wild-type *C. albicans* (Fig. 7f). The *vac8Δ/Δ* mutants scraped from solid agar were also filled with large vacuoles in the hyphal tubes and tips (Fig. 7g). These results were contrary to our initial hypothesis that the *vac8* null mutants would lack apical vacuoles, but our findings suggested that the newly synthesized and fused large vacuoles within the hyphae were efficient to facilitate the hyphae extension and penetration into the agar.

The pathogenicity of the *vac8Δ/Δ* mutant was consistent with the agar-invasive phenotypes. There was no apparent reduction in its virulence compared to the wild-type *C. albicans*. In the *C. albicans*-peritoneal macrophages co-incubation assays, the proportion of PI-positive cells damaged by *vac8Δ/Δ* was not different from that by the wild-type *C. albicans*, but was higher than that damaged by *vam6Δ/Δ* (Fig. 7h–j). In the systemic infection model, mice infected with *vac8Δ/Δ* from the tail vein had a survival rate of approximately 30% within 21 days, which was higher than that in mice infected with the wild-type *C. albicans* (Fig. 7k), yet lower than that in mice infected with the HOPS-complex-deficient mutants (Fig. 3d). These results indicated that the pathogenicity of the *vac8Δ/Δ* mutant was reduced compared with that of wild-type *C. albicans*, but was still stronger than that of the HOPS-complex-deficient mutants. The renal fungal load did not show significant difference between *vac8Δ/Δ* and the wild-type *C. albicans* infected mice for 5 days (Fig. 7l). Consistently, the length of hyphae in kidneys of mice infected with *vac8Δ/Δ* mutant was similar to that of mice infected with the wide type *C. albicans* (Fig. 7m, n). Subsequently, we assessed the pathogenicity and invasion of *vac8Δ/Δ* in the murine skin infection model. Notably, no significant difference was observed in the dorsal skin lesions and fungal load within the skin homogenates between mice infected with *vac8Δ/Δ* and the wild-type *C. albicans* (Fig. 7o, p). Through the PAS staining, we found that the hyphae length of *vac8Δ/Δ* mutants invading to the skin was similar to the wild-type *C. albicans* (Fig.7q, r). Collectively, although the hyphae initiation was slowed, but the ability to penetrate agar was not significantly reduced in the *VAC8*-disrupted mutant. As a result, the lethality of *vac8Δ/Δ* in systemic infection models was reduced in comparison with the wild-type strain but was still higher than that of the HOPS-complex-deficient strains. The results of *GTR1*[OE]-*vam6Δ/Δ* (normal hyphae initiation but defect in hyphae penetration) and *vac8Δ/Δ* (slowed hyphae initiation but normal in hyphae penetration) mutants indicated that both hyphae initiation and penetration are required for the pathogenicity of *C. albicans*, but the large vacuoles produced by vacuole fusion play a decisive role in maintaining the penetration of *C. albicans* into solid tissues in host.

## Discussion

*C. albicans* can transform into different morphologies during commensalism and infection, such as yeast, hyphal, pseudohyphal, white, opaque, and gray cells[46]. Among them, the regulation of yeast-to-hyphae transition and its influence on virulence have been studied widely. During the hyphal initiation and elongations, the critical transcription factors and signaling pathways, including MAPK, Ras/cAMP/PKA, Cdc28/Hgc1, TOR, and pH sensory pathways, as well as the dynamics of septin ring and actin structures, have been extensively identified[16]. Furthermore, novel bioinformatics strategies have also been constructed to elucidate the hyphae forming transcriptional and protein networks[9,47]. These studies clarified the hyphae formation pathways activated by exogenous and endogenous signals. In the process of signaling transduction, the morphology of fungal cells changes dramatically. However, the changes in organelles, including the vesicle transports in maintaining hyphal initiation and extension, have been poorly studied[18,48]. *Gow* et al. have observed the vacuolar movements and segregation during the morphogenetic transition in *C. albicans* by FM4-64 staining, while the biological significance of

vacuolar dynamics and the mechanism of action has not been studied in depth[49]. In this study, we identified the defect in vacuole fusion in the HOPS-complex-deficient mutants and characterized their insufficient ability of hyphae formation. Compared with the slightly delayed initiation and shortened length in the liquid medium, we found that the hyphae of HOPS-complex-disrupted mutants showed more significant defects on solid agar. The effects of defective vacuole fusion on *C. albicans* invasion and virulence were then evaluated systematically in fungi-cell co-incubation and mouse infection models. The decreased virulence is primarily due to the impaired hyphal initiation and penetration in HOPS-complex-deficient mutants. On one hand, we elucidated the necessity of HOPS-complex-mediated vacuole fusion in maintaining hyphal initiation by facilitating TOR signaling activation. Overexpression of *GTR1* restored the delayed hyphal initiation caused by the disruption of *VAM6*. On the other hand, by real-time observing of the vacuole fusion in hyphae, we proved that the fusion of vacuoles creates large vacuoles in the hyphal septum, which are essential for the active penetration of hyphae into solid agar and host cells. Active penetration was used to describe the process of *C. albicans* hyphae physically pushing their way into the epithelial cell, which was also an important way for engulfed *C. albicans* to destroy the immune cells membrane[33]. Our study also elucidated the essential role of large vacuoles in maintaining the hyphal initiation and active penetration of *C. albicans*, which provides new insights into the pathogenic mechanism of dimorphic or filamentous fungi.

Fungal vacuole is critical for maintaining osmotic pressure, degrading macromolecules, storing nutrients, and resisting stresses[50]. Cellular vacuolar space is associated with the cell cycle changes of yeast and hyphae, attributed to the size-regulated functions in eukaryotic cells. Many studies have found that fragmented vacuoles lead to changes in hyphal morphology, such as the hyphal branching phenotypes of *VAC1*, *VAM2*, *VAM3*, and *VAC8* null mutants[45,51]. As an opportunistic pathogen, it is significant to elucidate the regulatory mechanisms of fragmented vacuoles on the pathogenicity of *C. albicans*. Our study revealed the discriminative effect of fragmented vacuoles on the growth of hyphae in liquid and solid media. Fragmented vacuoles led to deficiencies in both TOR signaling activation and hyphal penetration forces. Previous studies have shown that the TOR pathway senses multiple nutritional signals, whose activation relies on complex regulatory networks, such as the EGO complex, *PIB2*, and phosphatidylinositol 3-phosphate (PI(3)P) pathways[41,52,53]. Our studies indicated that the TOR signaling activation maintained by large vacuoles is also important for the timely responses of *C. albicans* to hyphae-inducing signals. All the *vam6Δ/Δ*, *vps41Δ/Δ*, and *ypt72Δ/Δ* mutants showed decreased activation of TOR signaling and delayed hyphal initiation. Consistently, TOR pathway inhibitor rapamycin inhibited the hyphal initiation and elongation in the wild-type strain, which is consistence with previous results[28]. Overexpression of *GTR1*, upstream of the TOR pathway, restored the slow initiation caused by the disruption of *VAM6*. The effect of fragmented vacuoles on the TOR pathways activation may be due to the distribution of TORC1 complex on the surface of vacuoles membrane, as the kinase activity of TORC1 is activated by the small GTPase at the lysosome (vacuole)[54,55]. When there is no large vacuolar membrane surface, the interaction of TORC1 and EGO complex may be affected to some extent[56]. But to our surprise, overexpression of *GTR1* partially recovered the function of TOR signaling in the *vam6* null mutant. In *S. cerevisiae*, Vam6 protein is not only involved in the formation of the HOPS complex and mediates the tethering process for vacuole fusion, but also functions as a guanylate exchange factor that regulates the GDP- or GTP-binding status of Gtr1 protein[57]. The restoration effect of *GTR1* overexpression in our experiments suggested that *GTR1* activation may not be limited to the functional Vam6 but may also be regulated by other guanosine triphosphates or proteins, especially in the hyphae-inducing conditions, as there have been reported that leucyl-tRNA synthetase (LeuRS)

Cdc60 could interact with Gtr1 to regulate the leucine-dependent TOR activation[58]. On the other hand, overexpression of *GTR1* may elevate its low intrinsic GTPase activity and prompt the activation of TOR signaling[59]. Further studies on the regulation of TOR signaling under hyphae-induced conditions will provide a deeper understanding of the pathogenic mechanism of *C. albicans*.

Overexpression of *GTR1* compensated for the reduced TOR activity as well as hyphal initiation and cell damage caused by hyphae, but it did not fully restore the filamentation defect in the *vam6* null mutant penetrating to agar or solid organs, such as kidney and skin. In contrast, the disruption of *VAC8*, which encodes a gene required for vacuolar inheritance, resulted in slower germination and hyphae growth in liquid medium. To our surprise, *vac8*Δ/Δ also formed large vacuoles in hyphae when penetrating agar. For penetrating solid kidneys and skin, the *vac8*Δ/Δ mutant also showed no obvious defect, and its pathogenicity in vivo was significantly higher than that of the HOPS-complex-deficient mutants. The opposite phenotypes of *GTR1*^OE-*vam6*Δ/Δ and *vac8*Δ/Δ mutants suggests that the production of the HOPS-complex-mediated large vacuoles plays a more dominant role in maintaining *C. albicans* pathogenicity than hyphal initiation. The HOPS-complex-deficient mutants did not show significant difference in secretion and expression of virulence factors, so it was suggested that the large vacuoles may provide the mechanical support for hyphae penetrating the solid matter. In our study, the deficiency of vacuolar support is mainly manifested by the impaired hyphal invasion to the solid agars. With the decrease of agar concentration, the invasion zone of hyphae on solid agar was increased, which indirectly reflect the forces of hyphal penetration. When compared with the wild-type *C. albicans* invasion zones at the equivalent agar plates, the penetration deficiency of HOPS-complex-disrupted mutants in 0.5% agar were significantly smaller than that in 2% agar. These results suggest that the ability of *C. albicans* invading agar is mainly affected by the mechanical support provided by large vacuoles. Some studies have reported the driving forces and turgor pressure produced by vacuolar expansion. In *Basidiobolus ranarum*, the driving forces produced by enlarging vacuole pushes cytoplasm into developing conidia, which eventually break through the spore walls and disperse into the air[60]. In the appressorium cell of *Magnaporthe grisea*, large vacuoles were also required for generating the massive turgor pressure to invade plant. The turgor pressure could promote *M. grisea* to penetrate plant tissue via the infection peg, which indicated the large mechanical forces provided by vacuoles. The turgor pressure was derived from the accumulation of glycerol in large vacuoles[61]. In addition, plant vacuolar protein PIEZO could sense mechanical forces and modulate vacuole morphology[62]. Many studies indicate that vacuoles are indispensable for sensing and regulating mechanical forces in fungi. Our study mainly demonstrates the necessity of large vacuoles in *C. albicans* hyphae to penetrate solid agar and organs. However, the detection and modulation of the vacuolar turgor pressure in *C. albicans* remains to be further investigated.

More precise methods are needed for the determination of mechanical forces. To accurately investigate the forces exerted by hyphae, micro-fabrication approaches employ the elastomer polydimethylsiloxane (PDMS) as single-cell force sensors for fungal cells[25]. In addition, they found that filament buckling angle also reflects the response to resistive forces. However, no significant hyphal buckling was observed in hyphae induced by normal liquid or solid medium. The unobserved hyphal buckling indicates the limitation of PDMS in determining the mechanical penetration of individual cells, which cannot be accurately determined for slight penetration differences. In addition, with the development of fluorescent probes, N-BDP can be used to detect the turgor pressure and the tension of the hyphal plasma membrane by measuring the fluorescence lifetime. The wide synthesis and application of such probes will provide more convenient tools to detect the penetration forces[63].

And, the advancement of novel techniques for single-cell force real-time detection will offer a more efficient approach to investigate the virulence of filamentous fungi. In addition to impaired mechanical forces, fragmented vacuoles may also affect other functions of normal vacuoles. *Li* et al. reported that deletion of *VAM6* caused the defect of autophagy[26]. However, it has been reported that autophagy was not required for *C. albicans* yeast-to-hypha differentiation[64]. Dysfunction caused by fragmented vacuoles is not limited to autophagy. *Li* et al. also identified the role of *VAM6* in the maintenance of mitochondria function under oxidative stress resistance[65,66]. The deletion of *VAM6* attenuated the virulence in *C. albicans*. The phenotypes of shorted hyphae in liquid RPMI1640 medium and the smooth colonies with very short wrinkles on solid agar in the *vam6* null mutant were similar to our data. The appearances of colonies in both the wild type and the *vam6* null mutant on the solid Spider medium were consistence with ours as well. Moreover, their study focused primarily on the effects of *VAM6* on autophagy, mitochondrial function, and response to oxidative stress, suggesting the functional diversity of vacuoles in *C. albicans*. However, comparing the functions of normal vacuoles and fragmented vacuoles one by one is challenging. Our results indicated that fragmented vacuoles did not influence the secretion of SAPs and the expression of virulence-related genes. Meanwhile, the membrane integrity and acidic compartments of small vacuoles was also demonstrated, which in turn partially ruled out the defects in the function of small vacuoles. Our results provide some evidence that large vacuoles produced by vacuolar fusion provide sufficient mechanical force supporting hyphal to penetrate the host solid tissues. Recovery of TOR signaling does not restore vacuolar morphology and pathogenicity, which further confirmed the importance of large vacuoles in maintaining *C. albicans* virulence. However, our study has some limitations in quantitatively assessing the relationship between vacuolar size and hyphal invasion. It would be beneficial to better understand the role of vacuoles in *C. albicans* pathogenicity by precisely controlling the movement of vacuoles to the tips of hyphae or by determining the correlation between vacuole size and hyphal mechanical forces.

During determining the in vivo virulence of *C. albicans* mutants, we also used the yeast-blocked *EFG1* and *CPH1* double-deleted mutant (*efg1*Δ/Δ *cph1*Δ/Δ), which showed normal vacuolar morphology in yeast-form cells in the hyphal-inducing medium (Supplementary Fig. 7). However, C57/BL6 mice infected with $5 \times 10^5$ CFU of *efg1*Δ/Δ *cph1*Δ/Δ mutant did not show any abnormal locomotion and death during systemic infection[67]. Compared with the yeast-blocked *efg1*Δ/Δ *cph1*Δ/Δ mutant, HOPS-complex-deficient mutants is more virulent. But our study indicates that in addition to secreting virulence factors, hyphae can also cause tissue damage through mechanical penetration, which was promoted by the large vacuoles located in the hyphae tubes and tips. The production of large vacuoles was relied on the HOPS-complex-mediated vacuolar fusion (Fig. 8). Our finding suggested that targeting vacuolar fusion or HOPS complex could be explored as a novel antifungal strategy to reduce damage to host tissues.

## Methods

### Ethics statement

All animal experiments were performed according to institutional guidelines and have been approved by the Committee on Ethics of Medicine, Naval Medical University.

### Media and cultural conditions

All strains and mutants were maintained in 15% glycerol stocks and stored at −80 °C. Before experiments, strains were streaked on YPD (2% Bacto peptone, 2% dextrose, 1% yeast extract) for 48 h at 30 °C. Each colony was then grown overnight in liquid YPD at 30 °C with shaking. Transformants were selected on complete synthetic media

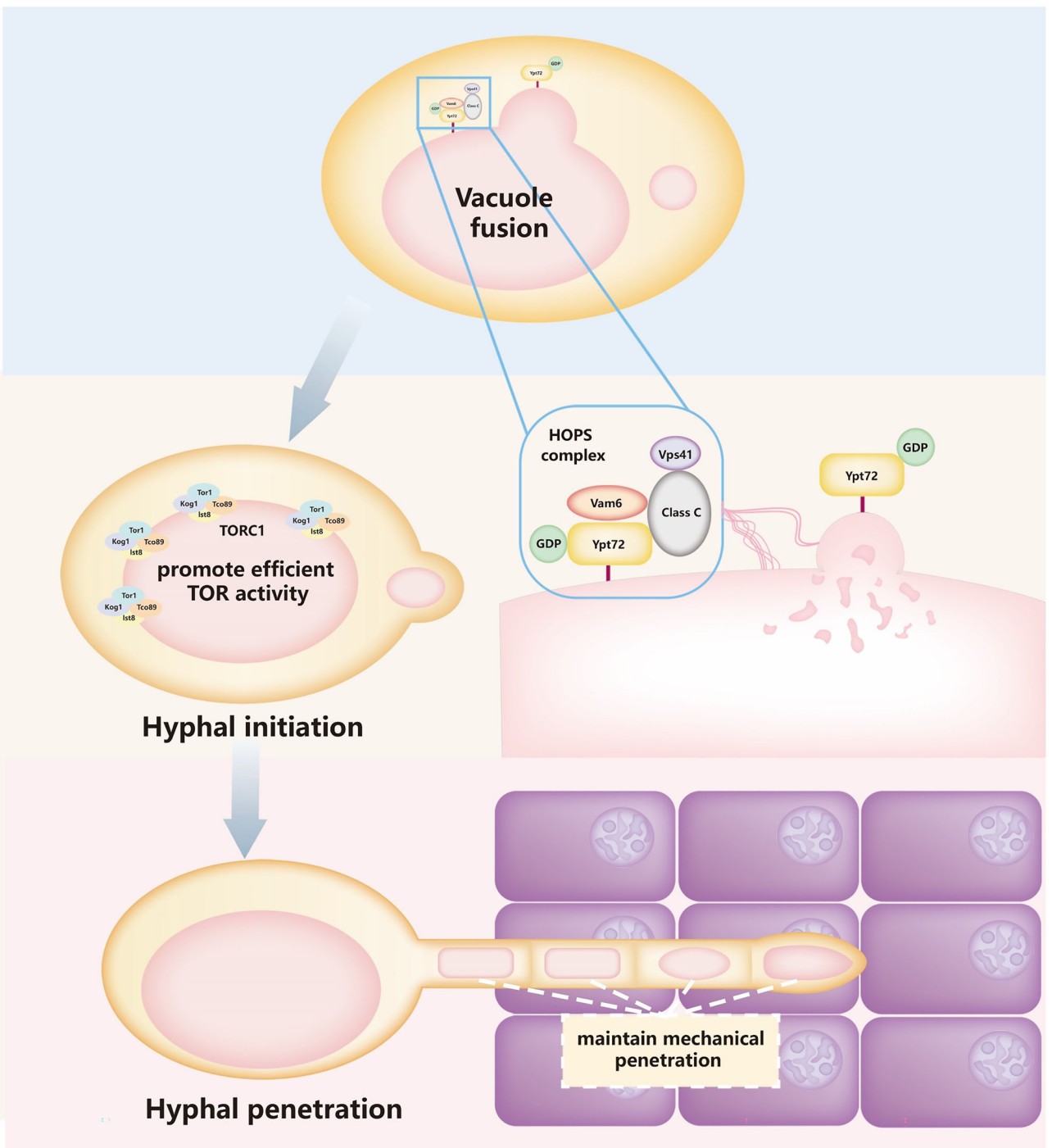

**Fig. 8 | The role of HOPS-complex-mediated vacuolar fusion in hyphae formation.** The HOPS complex, composed of Vam6, Vps41, Class C proteins, and Rab GTPase Ypt72 regulates the tethering of vacuoles in the process of vacuolar fusion. Since the TORC1 complex is located on the vacuolar membranes, large vacuoles generated by fusion are facilitators of efficient activation of TOR signaling, which also promotes signaling transduction during hyphal initiation. In addition, large vacuoles in hyphae would contribute *C. albicans* to penetrating through the host cells by maintaining mechanical support forces.

plates (CSM; 2% dextrose, 1.7% Difco yeast nitrogen base with ammonium sulfate and auxotrophic supplements) lacking histidine or selected for nourseothricin-resistance on YPD supplemented with nourseothricin (Sigma, Catalog #74667). For hyphal induction assays, overnight cultured strains were grown in liquid Spider medium (1% nutrient broth, 1% mannitol, 0.2% potassium phosphate dibasic) or RPMI1640 medium (Gibco BRL) adjusted with MOPS (Yuanye, Catalog #1132-61-2) to pH 7.4 plus 10% fetal bovine serum (FBS; Gibco) at 37 °C. Solid media plates routinely contained 2% agar, otherwise, the proportion of agar was prepared as indicated. For biofilm formation assays, RPMI1640 medium supplemented with 10% FBS was used. For nitrogen starvation, overnight cultured strains were washed twice and resuspended in the synthetic medium without amino acids (SD; 1.7% Difco yeast nitrogen base without ammonium sulfate and 2% dextrose). Low-quality nitrogen SD medium contained 40 mM ammonium sulfate or 10 mM arginine. For co-incubation assays with macrophage or human umbilical vein endothelial cells (HUVECs), strains were cultured in Dulbecco's modified eagle medium (DMEM, 4.5 g/L glucose,

L-glutamine, sodium pyruvate, Meilunbio) supplemented with 10% FBS, 100 U/ml penicillin, and 10 μg/ml streptomycin in a humidified 5% $CO_2$ incubator at 37 °C. For secretory aspartate proteases (SAPs) secretion assay, YCB-BSA (11.7 g/L yeast carbon base), 0.01% yeast extract, and 0.2% BSA (bovine serum albumin) medium is used.

## Strain constructions
All strains and plasmids used in this study are listed in Supplementary Table 1. All primers used in this study are included in Supplementary Table 2. A detailed description of strain construction is included in Supplementary Methods.

Target gene deletions were performed using a transient CRISPR-Cas9 system as reported[68]. Plasmid pV1093 was used to amplify the CAS9 expression and sgRNA expression cassettes. Plasmids pMH01 and pMH02 with the recyclable *Candida dubliniensis HIS1* marker and pNAT with the nourseothricin-resistance marker were used to amplify the deletion cassettes with the corresponding marker. The deletion cassette contains 80 bp homology sequences of the target ORF upstream and downstream regions. PCR products were then transformed into *C. albicans* SC5314 with *HIS1* auxotrophic parental strain using the lithium acetate transformation method. The positive colonies were confirmed by colony PCR followed by genomic PCR. To validate the construction of our deletion mutants, one copy of the SC5314 allele of each target gene was reintroduced at the original *HIS1* locus. The reintroduced fragment contains the whole target ORF, together with ~1000 bp of the target upstream and ~400 bp of the downstream region. The *NAT1* marker amplified from pNAT was fused to the above-reintroduced fragment.

To construct the GFP-tagged strain, plasmid pCPC160 or pCPC158 was used to amplify the fragment containing marker and GFP, with the long primers homology to the upstream of the target gene, resulting in the N-terminal fused GFP to the target gene[69]. The two alleles of *VAM6* were disrupted in the Yvc1-GFP[32] strain to obtain the strain Yvc1-GFP-*vam6*Δ/Δ.

To overexpress *GTR1* in the *vam6*Δ/Δ null mutant, plasmid pJK1277 was linearized with BsrGI enzyme and then transformed into the *vam6* null mutant screened by nourseothricin-resistance[70].

## Hyphal induction assays and imaging
Hyphal induction and then calcofluor white (CFW, Fluid analytics 18909) staining assay in liquid medium were carried out according to reported[71]. Overnight cultured strains were diluted to an $OD_{600}$ of 0.4 in RPMI1640 medium plus 10% FBS or Spider medium and were grown for 4 h at 37 °C. Strains were harvested, fixed with 4% formaldehyde for 15 min, stained with 0.2 mg/ml of CFW, and then resuspended in PBS. Stained strains were observed and photographed with 1.5-fold magnification on a Zeiss Axiovert 200 microscope with a 63× objective, or on a Leica Stellaris 8 STED scanning microscope with 40× water-immersion objective. To quantify the length of hyphae, the distance from a mother yeast cell to its hyphal tip was analyzed by Image J (1.53e). At least 50 separate hyphae were evaluated. The number of clearly separated hyphae and their first and secondary branching from at least 25 fields of view was counted.

The elongation of hyphae was recorded by time-lapse phase-contrast scanning on the Zeiss Axiovert 200 microscope with a 40× objective. Strains were cultured in RPMI1640 plus 10% FBS medium in the cell chamber maintained at 37 °C. The images were collected for 50 time points, with one image every 2 min. A total of 15 planes were scanned for each record. Usually, the center plane 8 or 9 gave the ideal focus. The lengths of six clearly separated hyphae at each time point were measured using Image J.

For determination of the ratio of germination, *C. albicans* were cultured in RPMI1640 plus 10% FBS or Spider medium at 37 °C. The ratio of germinated cells to all the cells observed in the microscopic field was considered as the ratio of germination. In each field,

approximate 50–100 fungal cells were observed under the microscope (EVOS xl, AMG) with a 40× objective.

Hyphal induction assays on solid medium were carried out according to previously reported[72] with some modifications. Overnight grown *C. albicans* strains were washed and adjusted to an $OD_{600}$ of 0.1 with RPMI1640 plus 10% FBS or Spider medium. A volume of 1 μl of each strain suspension was spotted on RPMI1640 plus 10% FBS or Spider solid medium. The plates were static incubated at 37 °C for 5 days. The top view or quarter edge of colonies were photographed by camera or inverted phase-contrast microscope (EVOS xl, AMG) with a 4× objective. To image the penetration of hyphae into agar, the colonies were vertically sliced across the center with the hyphae invading into agar for 0.5–1.0 mm thickness sections, and the vertical sections were photographed under the microscope (EVOS xl, AMG) with a 4× objective. Different fields were recorded and jointed together to integrate to obtain a full view of the whole slice. The depth and width of hyphae invading into agar in the vertical section were analyzed by Image J. The diameter of the central region of the colony and the distance between the hyphal insertion and the edge of the central region were measured. Where indicated, rapamycin (Meilunbio; J0304A) or $CaCl_2$ were added into agar media before solidification, and the concentrations of agar were adjusted to 0.5% or 1%.

## Biofilm formation and imaging
Biofilm growth was carried out according to previously reported[71] with appropriate modifications. Overnight cultured strains were inoculated to an $OD_{600}$ of 0.5 in 2 ml of RPMI1640 medium plus 10% FBS containing a 1.5 cm × 1.5 cm silicone square (Bentec Medical Inc., Woodland) in a 12-well plate (Costar). The plate was incubated at 37 °C for 90 min. After two washes with PBS, the silicone square was transferred to a fresh medium and incubated at 37 °C for 24 h to allow biofilm formation. The mature biofilms attached to the silicone were then washed and fixed with 4% formaldehyde and 2% glutaraldehyde in PBS. Fixed biofilms were stained overnight with Concanavalin A, Alexa Fluor 594 conjugate (Life Technologies) diluted to 25 μg/ml in PBS. Biofilms were washed once more in PBS to remove excess dyes, then transferred to glass scintillation vials and index-matched through subsequent passages through 50:50 PBS and methanol, 100% methanol, 50:50 methanol and methyl salicylate solution, and 100% methyl salicylate. Biofilms were then visualized using a slit-scan confocal optical unit on a Zeiss Axiovert 200 microscope with a Zeiss 40×/0.85 NA oil-immersion objective. The apical and basal regions of the serial-focus image stack were set to span the inverted biofilm. Imaging processing, axial, and side-view projections were described as previously reported[73]. Image data were converted to 32-bit format in Image J, then processed using the Subtract Background function with a 50-pixel rolling ball radius, setting negative pixels to zero, reslicing, and selecting Max Intensity projection to produce side views.

## Vacuolar membrane staining and imaging
*C. albicans* strains grown overnight were adjusted to an $OD_{600}$ of 0.5 in fresh YPD medium or RPMI1640 plus 10% FBS and allowed to grow for 90–120 min. Both yeast and hyphal forms of *C. albicans* strains were incubated with VacuRed (500 nM) for 10 min at 37 °C in a humidified incubator and washed with PBS three times before imaging. The VacuRed was excited by a 545 nm laser (14 μW) and fluorescence was collected between 590 nm and 670 nm. Images were acquired on a Leica Stellaris 8 STED scanning microscope (LAS X 4.6.1.27508) equipped with a white light laser. A 100× oil-immersion objective was employed, and the fluorophore emission was collected with Hybrid Detectors (HyD). The fluorescence of GFP was excited by a 488 nm laser and collected at 507 nm.

For measurement of vacuole diameter, all images were processed using LAS AF Lite software (3.3.0 10134). The fluorescence signal intensity was utilized to distinguish individual vacuoles, and the

diameter of each vacuole was measured using the 'Draw scalebar' tool and annotated on the merged diagram. A total of 50 sets of diameter data were obtained for each sample, from which the mean and standard deviation were calculated.

For the imaging of the fusion process of vacuoles during hyphal growth, overnight cultured *C. albicans* SC5314 were adjusted to an $OD_{600}$ of 0.5 in fresh YPD medium. The fungal suspension was incubated at 30 °C with shaking for 4 h. About $5 \times 10^6$ CFU of *C. albicans* were added to a confocal dish with 5 ml of fresh RPMI1640 + 10% FBS medium and cultured statically at 37 °C for 1 h. Then, strains were stained with 500 nM VacuRed for 10 min and washed. Images were acquired on a Leica Stellaris 8 STED microscope every 30 s and processed using LAS AF Lite software (3.3.0 10134).

For the imaging of the integrity of vacuolar membrane stained by both VacuRed and quinacrine, overnight cultured *C. albicans* were adjusted to an $OD_{600}$ of 0.5 in fresh YPD medium. The fungal suspension was incubated at 30 °C with shaking for 4 h. Then, strains were stained with 500 nM VacuRed and 200 μM quinacrine (Sigma, Q3251) for 10 min and washed. Images were acquired on a Leica Stellaris 8 STED scanning microscope with a 100× oil-immersion objective. Fluorescence of quinacrine was excited by a 490 nm laser and collected at 525 nm. Images were processed using LAS AF Lite (3.3.0 10134).

### HUVEC or macrophages co-cultured with *C. albicans* assay

HUVECs (ATCC® CRL-1730) were cultured in DMEM supplemented with 10% FBS, 100 U/ml penicillin, and 10 μg/ml streptomycin in a humidified 5% $CO_2$ incubator at 37 °C. Approximately $1 \times 10^6$ HUVECs were added to each well in a 12-well plate with 1 ml of DMEM medium. Later, 100 μl of *C. albicans* strains ($1 \times 10^7$ CFU) were added to each well. The plates were placed at 37 °C in a humidified 5% $CO_2$ incubator for 6 h and 9 h. At each time point, 120 μl of the supernatant was measured by lactate dehydrogenase (LDH) Release Assay Kit (Beyotime, Catalog # C0017). The absorbance of each well was measured by a Thermo microplate reader at 490 nm. The absorbance used for calculation was subtracted from the background control well. The LDH release rate (%) was calculated by the absorbance of each sample divided by the absorbance of cells treated with LDH release agent.

ICR mice (female, 6–8 weeks) obtained from Bikaikeji Biotechnology Co., LTD were treated with cervical dislocation and cleaned with 75% ethanol. Each mouse was injected with 5 ml of cold PBS (containing 3% FBS). Peritoneal macrophages were collected by aspiration of the lavage fluid into a 5 ml syringe (again if necessary) and harvested by centrifugation at 160 g for 8 min. The cells were resuspended in DMEM medium and subsequently cultured in a 12-well plate with 10% FBS and 1% penicillin–streptomycin at 37 °C in a humidified 5% $CO_2$ incubator. The supernatant was discarded and the macrophages cells adhering to the wells were washed three times with PBS. 100 μl of *C. albicans* strains ($1 \times 10^7$ CFU) and 1 ml of peritoneal murine macrophages ($1 \times 10^6$ cells) were co-cultured for 3 h and 6 h, stained with 2 μl per well of 100 μg/ml of propidium iodide (PI, Meilunbio, MA0137), and incubated at 37 °C for 15 min in dark. Then the plate was photographed with a fluorescence microscope (Leica, DMLL-LED) with a 20× objective. Peritoneal murine macrophages in the absence of fungi were used as negative control. The percentage of PI-positive peritoneal murine macrophages was calculated by the number of red fluorescent dots divided by the total number of macrophages in the negative control.

For the imaging of GFP-fused Yvc1 during hyphal growth co-incubated with HUVECs, HUVECs ($1 \times 10^6$ cells) and *C. albicans* strains ($1 \times 10^6$ CFU) were co-incubated in a confocal microscope plate with 2 ml of DMEM medium at 37 °C in a humidified 5% $CO_2$ incubator for 3 h. Images were acquired on a Leica Stellaris 8 STED scanning microscope with a 100× oil-immersion objective. The fluorescence of

GFP was excited by a 488 nm laser and collected at 507 nm. Images were processed using LAS AF Lite (3.3.0 10134).

### Mouse model of *Candida* infection

All mice were housed in a 12 h light/dark cycle room with appropriate temperature of 18–25 °C and relative humidity of 30–70%. Mice were given free access to food and water throughout the study.

ICR mice (female, 6–8 weeks) obtained from Bikaikeji Biotechnology Co., LTD were used for the disseminated candidiasis model. *C. albicans* wild type and mutant strains were washed, resuspended in sterile normal saline after culturing in YPD medium at 30 °C overnight, and then were adjusted to $8 \times 10^6$ cells/ml. A total of 20 mice for each group were systemic infected with 0.2 ml of *C. albicans* strain suspensions to get $1.6 \times 10^6$ CFU per mouse by intravenous administration via tail vein. The whole blood from 5 mice from each group was collected by the eyeball extraction method to detect metabolites, including urea and creatinine 5 days after fungal inoculation. The kidneys of five mice in each group were taken out 5 days after fungal inoculation to investigate the fungal burden and to make pathological sections with periodic acid-Schiff (PAS) staining. Hyphae invading the kidney were observed through an inverted phase-contrast microscope (EVOS xl, AMG) with a 40× objective. The remaining mice were monitored daily for 2–3 weeks, followed by fungal challenges.

C57BL/6 mice (female, 6–8 weeks) obtained from Leigen Biotechnology Co., LTD were used for the cutaneous candidiasis model. Mice were intraperitoneally injected with dexamethasone (Zhengzhou Zhuofeng, H41020055) 25 mg/kg once a day for 3 consecutive days to make immunosuppression in advance and anesthetized with 4% chloral hydrate intraperitoneally injected. To expose the back skin, mice's back hair was shaved and wiped thoroughly by applying Nair on the back, followed by cleaning with 75% ethanol, then the back skin was lightly stroked with 220 grits sandpaper to remove stratum corneum. *C. albicans* wild type and mutant strains were harvested after culturing in YPD medium at 30 °C overnight, washed, and resuspended in PBS to obtain a concentration of $4 \times 10^9$ CFU/ml. Exposed back skin of each mouse was infected with 50 μl of *C. albicans* suspension to reach $2 \times 10^8$ CFU per mouse. Five mice were used for each strain-challenging group. Mice were euthanized 3 days later. About 15 mm × 15 mm size of the back skins were removed. One piece from the removed skin was used to make pathological sections with PAS staining. Hyphae invading into skin were observed through an inverted phase-contrast microscope (EVOS xl, AMG) with a 40× objective. The rest of the removed skin was cut and ground, resuspended in normal saline, plated onto YPD medium with ten-fold dilutions and grown at 30 °C for 48 h. The number of survived colonies was counted to calculate the fungal burden.

### Spot assay

Rapamycin was dissolved in DMSO, added to YPD medium, and used to prepare solid plates. Overnight *C. albicans* strains suspensions were adjusted to an $OD_{600}$ of 0.4 in YPD medium. Then each of the *C. albicans* strains was serial five-fold diluted in YPD to six concentration gradients. Each strain suspension was spotted orderly in line at a volume of 3 μl per spot on YPD solid medium containing different concentrations of rapamycin. The plates were incubated at 30 °C for 48 h and photographed.

### Quantitative real-time RT-PCR

Briefly, to prepare samples for RNA extraction, overnight growth *C. albicans* SC5314 was inoculated into 10 ml of fresh YPD at an $OD_{600}$ of 0.4, and incubated at 30 °C for 1.5 h with shaking. Strains were harvested by centrifugation at $14,000 \times g$ for 3 min in cold. For the detection of the gene expression level of strains co-cultured with HUVECs, HUVECs ($1 \times 10^7$ cells) and *C. albicans* strains ($1 \times 10^7$ CFU)

were co-incubated in a 9 cm × 9 cm plate with 10 ml of DMEM medium at 37 °C in a humidified 5% $CO_2$ incubator for 1.5 h or 3 h. The strains with or without cells were then scraped off the plate with a cell scraper and harvested. RNA was isolated using a column fungal RNAout kit (KangLang, China), and cDNA was generated using the PrimeScript RT Master Mix kit (Takara Bio Inc., Japan). Real-time RT-PCR was carried out using the SYBR Premix Ex Taq kit (Takara Bio Inc.) in ABI Quant-Studio 3 RT-PCR 96-well with the following cycle conditions: pre-incubation step was 95 °C for 150 s; the amplification step was 95 °C for 5 s and 60 °C for 34 s, repeat for 60 cycles. The melting curve was completed with the following cycle conditions: 95 °C for 15 s, 65 °C for 60 s and 97 °C for 1 s with a rate of 0.1 °C/s. Three biological replicates were carried out and at least two parallel wells for each reaction were conducted. The results were compared with Actin-1 as an internal standard, and the $2^{-(\Delta\Delta Ct)}$ method was used to indicate the fold change of gene expression level.

## Western blotting

Overnight cultured *C. albicans* strains suspensions were diluted to $OD_{600}$ of 1.0 and incubated in SD medium for 4 h. Ammonium sulfate or arginine or equal volume of ddH₂O was added and incubated for 1 h at 30 °C. The strains were harvested at 4 °C, resuspended in 500 μl of protein lysis buffer (50 mM Tris-HCl pH 7.5, 150 mM NaCl, 5 mM EDTA, 10% glycerol, 0.2% Nonidet P-40 substitute, 1 M glycerophosphate, 1 M NaF, 0.05 M Na-ortho vanadate, 1 M Pnpp, 10 mM PMSF, 100 mM DTT, Protease Inhibitor cocktail), transferred into screw-cap tubes with the addition of 500 μl of acid-washed glass beads, and homogenized by a bead-beater for 6 cycles of 45 s, with 3 min cooling on ice in each cycle. Then the tubes were centrifuged at $10,000 \times g$ for 10 min at 4 °C and supernatant was transferred to a new tube. The protein concentration was measured with a BCA protein assay kit (Beyotime; P0011). Equal amounts of protein (30 μg) were separated by 12% SDS-PAGE and then transferred onto PVDF membranes. After blocking with 5% skimmed dry milk powder dissolved in TBS containing 0.1% Tween-20 for 2 h at room temperature, the membranes were incubated with primary antibodies against P-S6 (anti-phospho (S/T) Akt substrate rabbit polyclonal antibody, Cell Signaling Technology, catalog # 9611, 1:1000 dilution), total S6 (anti-S6 sheep polyclonal antibody R&D, catalog # AF5436, 1:1000 dilution), or the loading control H3 (Anti-histone H3 rabbit polyclonal antibody, Cell Signaling Technology, catalog # 4499,1:1000 dilution) overnight at 4 °C. Then, the membranes were incubated with secondary antibodies (Peroxidase-conjugated Affini-pure goat Anti-Rabbit, Proteintech, catalog # SA00001-2, 1:2000 dilution; Peroxidase-conjugated Affinipure Rabbit Anti-Sheep, Pro-teintech, catalog # SA00001-16, 1:2000 dilution) for 2 h at room temperature and the probed protein was visualized on the Tanon ABL X5 Chemiluminescence Imaging System with AllDoc_X software. H3 was used as the loading control.

## SAPs secretion assay

*C. albicans* strains grown overnight were adjusted to an $OD_{600}$ of 0.3 in PBS. Each strain suspension was spotted at a volume of 1 μl per spot on solid YCB-BSA medium. The plates were incubated at 30 °C for 48 h and photographed. The size of each halo was measured by Image J.

## Statistics and reproducibility

All experiments were performed at least three times. Representative images are shown. The lengths of hyphae were measured using Image J version 1.53e. Statistical tests were conducted with GraphPad Prism software version 8.3.0 (538) or version 9.0.0 (121). Unpaired two-tailed Student's *t*-test was used to analyze the differences between two groups. Survival curves were compared using the Log-rank (Mantel-Cox) test. Comparison between the elongation rates of hyphae was analyzed with mixed linear model by SPSS Statistics 24. *P*-value < 0.05 was considered statistically significant.

## Reporting summary

Further information on research design is available in the Nature Portfolio Reporting Summary linked to this article.

## Data availability

The data supporting the findings from this study are available within the article file, supplementary information and source data file. Any other raw data in this study are available from the corresponding author upon request. Source data are provided with this paper.

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

## Acknowledgements

We appreciate Professor Aaron P. Mitchell for kindly providing the plasmids pV1093, pNAT, pMH01, pMH02 and for his edits of this manuscript. We appreciate Professor Julia Koehler for kindly providing the plasmid pJK1277. We appreciate Professor Jiangye Chen for kindly providing the plasmid pCPC160 and pCPC158. We appreciate Professor Qilin Yu for kindly providing the strain Yvc1-GFP. We appreciate Yuan Yu for kindly providing HUVECs. We are grateful to Frederick Lanni, Carol A. Woolford, Manning Y. Huang, Katherine Lagree for helpful discussions and technical assistance. This work was supported by the National Natural Science Foundation of China (No. 82173867 to L.Y., No. 22274056 to X.Y.C., No. 32270202 to N.N.L., No. 22178377 and 22374163 to T.W.), Shanghai International Science and Technology Cooperation Project (21430713000 to L.Y.), Shanghai Pujiang Program (No. 21PJD081 to L.Y.), and the MOST Key R&D Program of China (2022YFC2304703 and 2020YFA0907200 to N.N.L.).

## Author contributions

L.Y., Q.Z.L., X.Y.C., and N.N.L. conceived and designed the study. Q.Z.L. and L.Y. performed the data analysis and wrote the manuscript. L.Y., Y.L., R.N.W., J.C.L., M.T.F., Z.Y., Y.M.H., F.X., and T.W. conducted all the experiments. Y.L., R.N.W., and J.C.L. performed the statistical analysis of the data. M.T.F., T.W., and X.Y.C. performed VacuRed staining experiments and data analysis. L.Y., Q.Z.L., X.Y.C., N.N.L., Y.L., R.N.W., and Y.Y.J. discussed the experiments and results.

## Competing interests

The authors declare no competing interests.
