## [Peer Review File · Nature Communications]

The vacuolar fusion regulated by HOPS complex promotes hyphal initiation and penetration in *Candida albicans*REVIEWER COMMENTS

Reviewer #1 (Remarks to the Author):

The manuscript by Liu and colleagues investigated the role of the HOPS complex proteins, Vam6, Vps41 and Ypt72 in morphogenesis and pathogenesis of *Candida albicans*. Vam6 as part of this complex has been previously shown to be involved in *Candida* virulence (Mao et al *Biochem Biophys Res*; not mentioned by the authors). The authors provide some evidence that HOPS-complex-mediated vacuolar fusion promotes hyphal initiation and penetration. Some of the data is not clear and cannot substantiate the claims made by the authors. The following modifications or concerns will be addressed before publication.

Major:

- Figure 1f: Could the authors provide a picture of *vam6* mutant without a black box on its top. Concerning this biofilm experiment in the manuscript line 128-129: "the total thickness of biofilm in the *vam6* null mutant was 261 μ m, much shorter than the wild-type 482 μ m." Longitudinal picture of *Vam6* Δ/Δ does not support the statement above. Line 129-133: "Meanwhile, in terms of microscopic cross-section observation, the wild-type biofilms were denser in the lower layer and sparser in the top layer. However, with the absence of VAM6, the number of strands in the top layer of biofilms was significantly increased, and the hyphal entanglements were much denser, which may reflect the insufficient supporting force of hyphae (Fig. 1F)." Data does not support the statements for the same reason. Line 135-138: "Our results indicated that the disruption of VAM6 reduced the supporting force and thus decreased the thickness of biofilms. Combined with the deficient hyphae and biofilms in the *vam6* null mutant, we speculated that VAM6 is essential for maintaining hyphal rigidity." Data does not support the statements; data only show that *vam6* deletion reduces the speed of hyphal elongation which affects the branching appearance. There is no data showing that VAM6 reduced the supporting force (invasion into agar).
- Figure 2: Line 57: As expected, the vacuoles in *vam6* Δ/Δ , *vps41* Δ/Δ or *ypt72* Δ/Δ mutants were fragmented, with small vesicles all less than 1 μ m in diameter. Did the authors measure the vacuole or vesicle diameter please provide sufficient information for measurement. Line 162: "These results suggested that the HOPS complex can only affect the fusion of vesicles but not the integrity of the membrane (Fig. 2B)." Why a difference of vesicle size will suggest that? Please clarify. The authors claim that the HOPS complex only affects the fusion of vesicles to show that please perform TEM microscopy as well as fusion proteins marker. Hyphae in Fig 2b where grown in the same condition than in Fig 1b (RPMI 1640 + 10% FBS medium), which shows that hyphal growth was reduced in liquid. Why does *vam6* and wt hyphae have the same size in Figure 2b but not in 1a.
- Figure 3: Please provide Figure 3h with high magnification to show WT and HPOS mutants have hyphal length differences in kidneys. In Figure 3K Line 236: "Mice infected with *vam6* Δ/Δ , *vps41* Δ/Δ or *ypt72* Δ/Δ mutants for 5 days had less exfoliation and visible skin damage (Fig. 3K)" *Vam6* infected mice looks worse than the wt for me in scab size and WT mice looks more scratched for inoculation than the others. Anyway, where is the quantification? N=1? Need proper data. Line 236-239: PAS staining showed that the length of hyphae invading the skin in mice infected with *vam6* Δ/Δ , *vps41* Δ/Δ or *ypt72* Δ/Δ mutants was shorter and the invasion depth decreased by about 40%, which indicated a more minor lesion and reduced skin damage (Fig. 3L). Please provide sufficient information how the quantification of hyphal length in the PAS staining.
- Figure 4: In Figure S6, the authors show that all three mutants have reduce growth in liquid media suggesting a role in hyphal elongation which will affect secondarily penetration. Line 258: "However, it should be noted that TOR activity in HOPS-complex-deficient mutants was still maintained which promotes the normal proliferation of *C. albicans* cells, indicating that TOR activity is only partially impaired (Fig. S6A). Statement and the figure S6a don't match. Line 264: our results showed that 10 nM rapamycin, significantly inhibited the growth of *C. albicans* in Fig. 4A. 10nM rapamycin is not reported in Figure 4A and it seems 8nM rapamycin inhibits the *Candida* growth completely. Please clarify why 10nM rapamycin was used in Figure 4C. Also, Inconsistency for the spotted growth of *vam6* $^{-/-}$ strain from Figure 4A and Figure E. Please clarify 3nM rapamycin treated *vam6* has growth in 3 dilutions but less concentration rap con at 2nm inhibits the growth with first dilution. Please clarify.
- Figure 5: Line 304-305": Using continuous live cell imaging, we captured vacuolar fusion in hyphae. 305 Especially in the hyphal tip, two vacuoles fused to form a larger one within 90 s (Fig.

5C).” No data provided showing this for the mutants.

- Figure 6: Fig 6d, 5×10^5 CFU/mouse via tail has same virulence for WT than with 1.6×10^6 . Please clarify. Mutant is more virulent than in Figure 3d which seems contradictory since the inoculum is 10 times lower in fig 6d. Why did the author look at the fungal burden at day 2 while in Figure 3 they looked at day 5 with a higher inoculum. Should be consistent.

Minor

-Figure 1A Please provide the scale bar and please provide sufficient information on how an image is acquired, magnification, and length was measured. The authors should provide the statistics for Figure 1d. Authors should provide detailed information for Figure 1e.

-Line No 166: remove the word “function” verified colocalization of target proteins.

-The representative image for VAM6^{-/-} and the bar graph does not match Figure 2G. Please quantify the length or provide an alternative image.

-VacuRed has no reference and I could not find anything about this staining for vacuole.

- Line 217: “Overall, the HOPS complex is indispensable for active penetration and escape from phagocytes.” The use of the word indispensable is not accurate because there is no control uninfected so indispensability cannot be determined. Participate could be used instead.

-Authors should clarify whether the microscope images provided in Figure 3b are 3h or 6h time points.

1. How do the authors differentiate PI-positive macrophage and PI+ *C. albicans*?

Reviewer #2 (Remarks to the Author):

The manuscript by Liu et al. describes the role of *C. albicans* orthologs of the yeast HOPS complex in hyphae initiation and penetration of solid agar. Consistent with the impaired hyphal penetration in HOPS knockout *C. albicans* strains, their virulence is reduced in co-incubation with human umbilical vein endothelial cells (HUVEC) and peritoneal macrophages (PM) and in mouse models. The authors observe that vacuoles are fragmented in HOPS knockout strains. Interestingly, the authors observe that the “interior of the hyphal tip was filled with a large vacuole”, which occupies the intracellular volume of the leading septum. Notably, in HOPS knockout strains (vam6 Δ/Δ , vps41 Δ/Δ , or ypt72 Δ/Δ mutants) with a defect in vacuole fusion, these large vacuoles were absent from hyphae tips.

The study convincingly illustrates the role of the HOPS complex in *C. albicans* pathogenesis in human and mouse cells and mouse models. The manuscript presents an interesting hypothesis on the role of vacuole size in generating mechanical forces supporting hyphal penetration in *C. albicans*. However, the authors have not persuasively established that the defect in hyphal penetration in HOPS knockout *C. albicans* strains is due to vacuolar fragmentation and that large vacuoles indeed generate the mechanical forces required for hyphal penetration of *C. albicans*. To support their idea, the authors should show that abrogating vacuolar movement to the hyphal tip or inducing vacuolar localization to the tip is sufficient to change hyphal shape and penetration ability in solid media. Similarly, increasing or decreasing vacuolar size independent of HOPS function (such as by regulating vacuole fission) should be included to support the hypothesis that vacuolar size regulates the hyphal ability to penetrate.

-The data presented in Figs. 5a and b should be supported by quantification.

-The authors should rescue the HOPS expression in knockout strains and test whether the rescue strains show rescue of vacuolar size and localization at the hyphae tip and rescue of ability to penetrate solid media.

-Is the vacuolar localization to the hyphae tip also observed when *C. albicans* is co-cultured with HUVEC or macrophages? What happens in this experiment using HOPS knockout strains? This is an important experiment, as data from agar plates is not sufficient to conclude the role of vacuolar size in *C. albicans* infection under in vivo conditions.

Besides this major concern, there are some specific suggestions about some of the experiments

and their presentation. The author can consider adding these suggestions to strengthen their major conclusions:

- 1) The VAM6, VPS41, and ypt72 revertant strains should also be employed in Figure 3 to show rescue of *C. albicans* pathogenesis in cells or a mouse model.
- 2) The description for Figs. 3B and 3C is not clear as to how the authors conclude using propyl iodide (PI) staining that the HOPS complex is required for escape from phagocytes. These lines from 209–217 need to be rewritten for better explanation.
- 3) Fig. 5C: Live imaging of hyphal dynamics in the HOPS knockout strain should also be shown along with the WT strain to show whether vacuole fusion is observed or not in the hyphae in the HOPS knockout strains.
- 4) The description of Fig. 5F and G (lines 328–338) is unclear and needs to be rewritten. The graph shown in Fig. 5G should be described with more clarity in the text.
- 5) The mTORC phosphorylation activity (as shown in Fig. 4B) should be tested in the VAM6, VPS41, and ypt72 revertant strains to support the rapamycin spot assay data shown in Fig. S4.
- 6) Zoom-in images should be shown in Fig. 5B for clarity.
- 7) Line 142-143 states that "Meanwhile, Vam6 (Vps39), Vps41, Ypt72, and Class C proteins form the HOPS complex." However, Ypt72 in Fig. 2a appears to be a Rab7 ortholog and not part of the hexameric HOPS complex. Ypt72 should be introduced properly in the manuscript result section.
- 8) In the "Supplementary methods for strain construction" section, it states that "the presence of the product of primers LY22-17 and LY20-11 and the absence of the product of primers LY22-18 and LY22-18 were used to evaluate the correct strain by genomic PCR." The primer names should be corrected to LY22-17 and LY-22-18.

Reviewer #3 (Remarks to the Author):

The authors study the role of VAM6, VPS41, and YPT72 which encode components of the HOPS complex in *Candida albicans*. Here they demonstrate that mutants lacking these genes are defective in filamentation, particularly on solid media, and virulence in vivo. Rescue of the filamentation defect in the *vam6* mutant by overexpression of small GTPase encoded by GTR1, which encodes a subunit of the TORC1-activating EGO269 complex, did not restore hyphal penetration. Taken together, these results demonstrate an association between vacuolar integrity and the ability to invade solid agar, as an indirect measure of hyphal turgor pressure.

Major Concerns:

The Introduction should be presented more comprehensively with additional details. For example, in line 62, Noble et al found that virulence and hyphal morphogenesis were de-coupled, indicating that virulence and morphogenesis are related in a complex, non-linear process. Also, in line 58, the transcriptional pathways in morphogenesis are listed but not described. In line 75, key proteins involved in morphogenesis on solid surfaces are also listed but not described. A broader and more detailed background to provide substantive context is recommended.

Since individual colonies can vary substantially, were the agar invasion assays (filament length, width/depth) done in multiple instances? If not, how were the presented colonies chosen as representative samples?

The GTR1OE-*vam6* mutant seems to produce a somewhat incompletely restored phenotype, specifically, it does not appear to fully restore the filamentation defect of the *vam6* null mutant. Could this incomplete restoration be a contributing factor to the tissue penetration and virulence defect, either separately or in conjunction with the vacuolar defect? Can the authors more fully discuss this possibility? Are the filamentation defects fully rescued in the complemented control strains?

Additional images with more views of SC5314 vacuoles for Fig. 5B would be of benefit. The current image shows only a single hyphal element with stained vacuoles, thus rendering the figure more difficult to interpret.

The Discussion should provide more information and context on the mechanical role of vacuolar function and filamentation in *Candida* species, and importantly to also include studies in filamentous fungi such as *Neurospora* or several plant pathogens.

The role of degradative enzymes in tissue penetration should also be mentioned, since they play a

large role in tissue invasion. It would be straightforward to perform some simple indirect agar plate assays to assay for secreted aspartyl protease secretion, for example, in these mutants. In addition, the role of surface adhesins, which provide anchoring for *Candida* to form hyphae and invade tissues, should be mentioned, since defects in vacuolar regulatory proteins could potentially alter expression or trafficking of various secreted proteins. It would be very instructive to also include a comparison with hyphal-defective mutants unrelated to the vacuole. Overall, a tantalizing potential association between vacuolar morphology/function and hyphal penetration is revealed, but further experiments are needed to demonstrate causality related to the vacuole and turgor pressure.

Minor Concerns:

Strain construction: expected PCR sizes should be indicated and more details should be provided in Supplementary Methods and Supplemental Figure Legends. As currently presented, the expected gel findings are difficult to follow. There also seems to be a typographical error: "the absence of the product of primers LY22-18 and LY22-18".

Please use either "VacoRed" or "VacuRed" for consistency, not both. More details of this vacuolar probe should be provided.

Fig. 2D, it is difficult to compare length or other morphology of the filaments in SC5314 (compared to the mutant strains), due to clumping and the relatively low magnification used. A higher magnification image would be helpful. It would also be beneficial to include a scale bar.

Fig. 3j, please use the term "sacrificed" or "killed", not "executed".

Fig. 3h, the image resolution is sub-optimal; are higher resolution images available so the hyphae can be more adequately seen?

There are formatting/layout issues with the Supplemental Figures, such that it is difficult to discern which label belongs to which figure. These figures should be more clearly separated.

REVIEWER COMMENTS

Reviewer #1 (Remarks to the Author):

The manuscript by Liu and colleagues investigated the role of the HOPS complex proteins, Vam6, Vps41 and Ypt72 in morphogenesis and pathogenesis of *Candida albicans*. Vam6 as part of this complex has been previously shown to be involved in *Candida* virulence (Mao et al Biochem Biophys Res; not mentioned by the authors). The authors provide some evidence that HOPS-complex-mediated vacuolar fusion promotes hyphal initiation and penetration. Some of the data is not clear and cannot substantiate the claims made by the authors. The following modifications or concerns will be addressed before publication.

Response: Thanks for your constructive comments which help us a lot to improve the quality of our manuscript. In the process of studying and writing, we retrieved some relevant studies on *VAM6*, referenced and cited the article of *VAM6* involved in *C. albicans* autophagy¹. Following a reminder from reviewers, we have examined the findings about Vam6 and cited their articles in Discussion line 629-637^{2,3} *Li et al.* found that the hyphae in the *vam6* null mutants in liquid RPMI 1640 medium were shorter than those in the wild type strains, the colonies were formed without hyphal extension on the solid Spider medium, and that the pathogenicity of the *vam6* null mutants was decreased in mice. These phenotypes are consistent with those in the *vam6Δ/Δ* mutant constructed in our study. In addition, they mainly focused on the effects of *VAM6* on autophagy and mitochondrial function, which are important for *C. albicans* in response to nitrogen starvation and oxidative stresses. In contrast, we mainly focused on the regulation of HOPS complex on hyphal formation and the relationship between *C. albicans* virulence and the mechanical forces provided by large vacuoles. Although the conclusions and the mechanisms in our study are not similar to those in Li's study, some similar phenotypes can provide mutual support. Thanks again to your comments, which allow us to cite their results more normatively.

Line 629-637 “Li *et al* also identified the role of *VAM6* in the maintenance of mitochondria function under oxidative stress resistance. The deletion of *VAM6* attenuated the virulence in *C. albicans*. The phenotypes of shorted hyphae in liquid RPMI 1640 medium and the smooth colonies with very short wrinkles on solid agar in the *vam6* null mutant were similar to our data. The appearances of colonies in both the wild type and the *vam6* null mutant on the solid Spider medium were consistence with ours as well. Moreover, their study focused primarily on the effects of *VAM6* on autophagy, mitochondrial function, and response to oxidative stress, suggesting the functional diversity of vacuoles in *C. albicans*.”

1. Mao, X.; Yang, L.; Liu, Y.; Ma, C.; Ma, T.; Yu, Q.; Li, M., Vacuole and Mitochondria Patch (vCLAMP) Protein Vam6 Is Involved in Maintenance of Mitochondrial and Vacuolar Functions

under Oxidative Stress in *Candida albicans*. *Antioxidants* (Basel) 2021, 10 (1).

2. Mao, X.; Peng, L.; Yang, L.; Zhu, M.; Du, J.; Ma, C.; Yu, Q.; Li, M., Vacuole and mitochondria patch (vCLAMP) and ER-mitochondria encounter structure (ERMES) maintain cell survival by protecting mitochondrial functions in *Candida albicans*. *Biochem Biophys Res Commun* 2022, 591, 88-94.

3. Mao, X.; Yang, L.; Yu, D.; Ma, T.; Ma, C.; Wang, J.; Yu, Q.; Li, M., The Vacuole and Mitochondria Patch (vCLAMP) Protein Vam6 is Crucial for Autophagy in *Candida albicans*. *Mycopathologia* 2021, 186 (4), 477-486.

The point-by-point responses to each comment are shown in the following.

Major:

- Figure 1f: Could the authors provide a picture of *vam6* mutant without a black box on its top.

Thank you for your comments. We have provided a picture of *vam6* Δ/Δ mutant without a black box on its top. We carefully checked the morphology of biofilms in the previous confocal scan and provided all the results in the following figure. In our experiments, we set the length of z-axis scan manually and the top was set in the position with no apparent fluorescence. Therefore, the z-axis scanning was adjusted to a shorter length for *vam6* Δ/Δ due to its thinner biofilm compared to the wild type. The previously provided picture with a black box of *vam6* Δ/Δ mutant was loaded in Image J and we have removed it in our new manuscript.

Fig.1 *C. albicans* strains were assayed biofilm formation in RPMI 1640 + 10% FBS at 37 °C for 24 h. The biofilms were fixed, stained with ConA Alexa Fluor 594, and clarified using the solvent exchange protocol into methyl salicylate. Axial and side-view projections of biofilms are shown.

Concerning this biofilm experiment in the manuscript line 128-129: “the total thickness of biofilm in the *vam6* null mutant was 261 μm , much shorter than the wild-type 482 μm .” Longitudinal picture of *Vam6* Δ/Δ does not support the statement above.

Thank you for your comments. Different fields were observed and three side-view projections of biofilm images were produced. As shown in Fig. 1 above, the average thickness of biofilm of the wild type was 443.3 μm , while that of the *vam6* null mutant was 238 μm . One represented image was selected shown in the main text. We have also re-written the text in line 133-138, as follows, “Side-view and axial confocal projections revealed presence of hyphae in biofilms of the wild-type strain and the *vam6* null mutant. Strains at the base of the silicone produced hyphae rising to a dense zone, and longer hyphae above the dense zone. The long hyphae folded over on the surface of the biofilm. The depth of biofilm in the *vam6* null mutant was shorter than that in the wild type, which could be due to the shorter hyphae in the mutant”.

Line 129-133:” Meanwhile, in terms of microscopic cross-section observation, the wild-type biofilms were denser in the lower layer and sparser in the top layer. However, with the absence of VAM6, the number of strands in the top layer of biofilms was significantly increased, and the hyphal entanglements were much denser, which may reflect the insufficient supporting force of hyphae (Fig. 1F).” Data does not support the statements for the same reason. Line 135-138:” Our results indicated that the disruption of VAM6 reduced the supporting force and thus decreased the thickness of biofilms. Combined with the deficient hyphae and biofilms in the *vam6* null mutant, we speculated that VAM6 is essential for maintaining hyphal rigidity.” Data does not support the statements; data only show that *vam6* deletion reduces the speed of hyphal elongation which affects the branching appearance. There is no data showing that VAM6 reduced the supporting force (invasion into agar).

Thanks for your comments. Our previous sentence had a lot of overstatement. According to your suggestion, we have re-written the results concerning to Fig. 1F in line 133-138. “As hyphae are important components in maintaining the structure of *C. albicans* biofilms, we explored biofilm formation. Side-view and axial confocal projections revealed presence of hyphae in biofilms of the wild-type strain and the *vam6* null mutant. Strains at the base of the silicone produced hyphae rising to a dense zone, and longer hyphae above the dense zone. The long hyphae folded over on the surface of the biofilm. The depth of biofilm in the *vam6* null mutant was shorter than that in the wild type, which could be due to the shorter hyphae in the mutant (Fig. 1f).”

- Figure 2: Line 57: As expected, the vacuoles in *vam6* Δ/Δ , *vps41* Δ/Δ or *ypt72* Δ/Δ mutants were fragmented, with small vesicles all less than 1 μm in diameter. Did the authors measure the vacuole or vesicle diameter please provide sufficient information for measurement.

Thank you for your comments. We did not quantitatively measure the diameter of each vacuole in

our previous manuscript. Instead, based on the local magnification of confocal images, we provided a rough estimation that the vacuole diameter was less than 1 μm . Following your suggestion, we have employed stimulated emission depletion (STED) imaging with enhanced resolution to determine the vacuolar diameters in *C. albicans* strains. All images were processed using LAS AF Lite software. The fluorescence signal intensity was utilized to distinguish individual vacuoles, and the diameter of each vacuole was measured using the 'Draw scalebar' tool and annotated on the merged diagram. A total of 50 sets of diameter data were obtained for each strain, from which the mean and standard deviation were calculated. In the wild-type yeast form *C. albicans*, the large vacuoles were irregularly shaped and the inner diameter of 50 randomly selected large vacuoles was about 2782 ± 1259 nm. In contrast, the vacuoles in the *vam6 Δ/Δ* , *vps41 Δ/Δ* , and *ypt72 Δ/Δ* mutants were mostly regular round with diameters of 436 ± 91 nm, 484 ± 112 nm, and 464 ± 99 nm, respectively. We have updated these results in Fig. 2c, line 169-179 in the text.

Line 169-179 “With the high resolution of the VacuRed probe and STED imaging, some small vesicles in the HOPS-complex-deficient mutants were clearly distinguished (Fig. 2c). In the wild-type yeast form *C. albicans*, the large vacuoles were irregularly shaped and the inner diameter of 50 randomly selected large vacuoles was about 2782 ± 1259 nm. In contrast, the vacuoles in the *vam6 Δ/Δ* , *vps41 Δ/Δ* , and *ypt72 Δ/Δ* mutants were mostly regular round with diameters of 436 ± 91 nm, 484 ± 112 nm, and 464 ± 99 nm, respectively. The STED imaging of vacuolar membrane suggested that the HOPS complex did not influence the integrity of the membrane. To further confirm the integrity of vacuolar membrane in the mutants, quinacrine, a dye that accumulates inside acidic vacuoles, was used to determine whether the compartments of the fragmented vacuoles were acidic and separated by membranes”

Fig.2 Measurement of vacuolar diameters in *C. albicans*. Fluorescence of VacuRed were acquired

on a STED imaging. The fluorescence signal intensity was utilized to distinguish individual vacuoles, and the diameter of each vacuole was measured using the 'Draw scalebar' tool and annotated on the merged diagram.

Line 162: “These results suggested that the HOPS complex can only affect the fusion of vesicles but not the integrity of the membrane (Fig. 2B).” Why a difference of vesicle size will suggest that? Please clarify.

Thank you for your comments. To confirm the integrity of vacuolar membrane, we have added the results of STED imaging, transmission electron microscopy (TEM) observations, and quinacrine staining. All these results suggested that the vacuolar membrane of HOPS-complex-deficient mutants was intact. We have updated these results in Fig. 2c-2e (Fig. 3a-3c in below), line 169-190 in the text.

Line 169-190 “With the high resolution of the VacuRed probe and STED imaging, some small vesicles in the HOPS-complex-deficient mutants were clearly distinguished (Fig. 2c). In the wild-type yeast form *C. albicans*, the large vacuoles were irregularly shaped and the inner diameter of 50 randomly selected large vacuoles was about 2782 ± 1259 nm. In contrast, the vacuoles in the *vam6* Δ/Δ , *vps41* Δ/Δ , and *ypt72* Δ/Δ mutants were mostly regular round with diameters of 436 ± 91 nm, 484 ± 112 nm, and 464 ± 99 nm, respectively. The STED imaging of vacuolar membrane suggested that the HOPS complex did not influence the integrity of the membrane. To further confirm the integrity of vacuolar membrane in the mutants, quinacrine, a dye that accumulates inside acidic vacuoles, was used to determine whether the compartments of the fragmented vacuoles were acidic and separated by membranes¹. As shown in Fig. 2d and Supplementary Fig.2i, the vacuolar membrane of the wild type strain was stained by VacuRed, while green fluorescence of quinacrine was distributed within the vacuolar membrane. In the HOPS-complex-deficient mutants, the distribution of fluorescence of the red VacuRed and the green quinacrine of each fragmented vacuole did not be clearly resolved due to the limits of microscopic resolution. However, the obvious co-localization of the two probes indicated that the fragmented vacuolar compartments were also acidic and isolated from the cytoplasm by the vacuolar membranes. Transmission electron microscopy also showed that the refraction inside the vacuole was different from that of other organelles (Fig. 2e). Large vacuoles and intact membrane edges were seen in the wild-type *C. albicans*. In contrast, the vacuolar volume was significantly smaller in the HOPS-complex-disrupted mutants, whereas the vacuolar edge contour was still distinguishable.”

Fig.3 (a) Measurement of vacuolar diameters in *C. albicans*. Fluorescence of VacuRed were acquired on a STED imaging. The fluorescence signal intensity was utilized to distinguish individual vacuoles, and the diameter of each vacuole was measured using the 'Draw scalebar' tool and annotated on the merged diagram. **(b)** Imaging of vacuole compartment stained by quinacrine. *C. albicans* strains were cultured in YPD medium at 30 °C for 4 h. **(c)** Transmission electron microscope images in *C. albicans* strains. The yellow circles represent vacuoles (V).

The authors claim that the HOPS complex only affects the fusion of vesicles to show that please perform TEM microscopy as well as fusion proteins marker.

Thank you for your suggestion. TEM microscopy has been performed and the images of vacuoles in *C. albicans* have been updated in Fig. 2e.

Inspired by your suggestion, Yvc1, a calcium channel protein specifically distributed on the vacuolar membrane, was selected as a fusion protein marker. During real-time observations, we observed a clear colocalization of VacuRed and Yvc1-GFP. In particularly, we observed that Yvc1-GFP accumulated to form a highlighted fusion region at the site of contact between two vacuoles. However, in the *VAM6*-disrupted mutant, the fluorescence of Yvc1-GFP became diffuse and no obvious highlighted region was observed, suggesting that there was no obvious fusion event in the fragmented vacuoles (Supplementary Video 1 - 4). The results of TEM, quinacrine staining and Yvc1-GFP images suggested that the HOPS complex mainly promoted the fusion of vesicles but did

not affect the integrity of the vacuolar membrane.

Fig.4 (a) Transmission electron microscope images in *C. albicans* strains. The yellow circles represent vacuoles (V). (b) Imaging of GFP-tagged Yvc1 with VacuRed. The Yvc1-GFP strains were cultured in YPD medium at 30 °C or in RPMI 1640 + 10% FBS medium at 37 °C for 90-120 min, and stained with 500 nM VacuRed at 37 °C for 10 min. The yellow arrows represent the site of fusion.

Hypahe in Fig 2b where grown in the same condition than in Fig 1b (RPMI 1640 + 10% FBS medium), which shows that hyphal growth was reduced in liquid. Why does *vam6* and wt hyphae have the same size in Figure 2b but not in 1a.

Thanks for your careful evaluation. VacuRed is a fluorescent amphiphilic styryl dye that embeds into the membranes of synaptic vesicles as endocytosis is stimulated. Lipophilic interactions cause the dye to greatly increase in fluorescence, thus emitting a bright signal when associated with vesicles and a nominal one when in the extracellular fluid. As a result, the VacuRed can only be used for staining living cells. When we observed the vacuoles of the *vam6Δ/Δ*, *vps41Δ/Δ* and *ypt72Δ/Δ* mutants, the hyphae were induced for about 90-120 min and kept still alive. By examining the rate of hyphal elongation depicted in Fig. 1d, it showed that *vam6Δ/Δ* exhibited a mere 10-minute delay compared to the wild type in achieving equivalent hyphal growth. So, the length of hyphae may vary due to the disparity in the timing of live observation and photography during VacuRed staining. In such experiment, we mainly focused on the vacuole morphology without strict controlling over the imaging time, which might result in insignificant differences in hyphal length between the wild type and the mutant strains. So, these representative images in Fig. 2b were

selected for better displaying the morphology of vacuolar fragmentation.

- Figure 3: Please provide Figure 3h with high magnification to show WT and HOPS mutants have hyphal length differences in kidneys.

Thanks for your suggestion. We have provided a high quality of images and zoom-in images to distinguish the morphology of hyphae in kidneys in Fig. 3h, 3m, 6f, 6j, 7m, 7q.

Fig 5. Fig 3h in our new manuscript. The hyphae in kidneys stained by PAS.

In Figure 3K Line 236:” Mice infected with *vam6* Δ/Δ , *vps41* Δ/Δ or *ypt72* Δ/Δ mutants for 5 days had less exfoliation and visible skin damage (Fig. 3K)” Vam6 infected mice looks worse than the wt for me in scab size and WT mice looks more scratched for inoculation than the others. Anyway, where is the quantification? N=1? Need proper data.

Thank you for your comments. The number of mice used were shown in the figure legend of Fig. 3l. Each group contains 5 mice, and 3 of them were photographed to show the skin lesions. The backs of mice infected with the wild-type *C. albicans* exhibited a higher prevalence of crusts and shedding of dander. Mice infected with the HOPS-complex-deficient mutants showed significantly less skin damage on the back. To better quantification, we determined the fungal load in the lesion skin. Our results showed that the fungal load in the skin of mice infected with the HOPS-complex-deficient mutants was significantly lower than that of mice infected with the wild type *C. albicans*. Subsequently, in the skin PAS staining results, the HOPS-complex-deficient mutants showed shorter invasive hyphae by quantification. Collectively, these results consistently demonstrated that the ability of *C. albicans* invasion to skin was significantly reduced after the disruption of HOPS complex.

Fig.6 Fig. 3j-3n in our revised manuscript. **j** The experiment process for cutaneous candidiasis model. Female C57BL/6 mice were intraperitoneally injected with 25 mg/kg of dexamethasone (i.p.) once a day for 3 days. On day 4, mice were infected with *C. albicans* (2×10^8 CFU/mouse) on exposed back skin. The infected back skins were photographed and then removed to make pathological sections on day 7. **k** The exposed back skin with fungal-infected damage on day 7. **l** The fungal burden in damaged skin of mice (n=5) were determined 3 days after fungal inoculation. **m** The pathological images of infected skin stained by PAS. The dimension of skin damage was quantified by the maximum depth of penetrated hyphae measured by Image J. **n** The hyphal length in damaged skin was measured by Image J. At least 20 hyphae were measured.

Line 236-239: PAS staining showed that the length of hyphae invading the skin in mice infected with *vam6Δ/Δ*, *vps41Δ/Δ* or *ypt72Δ/Δ* mutants was shorter and the invasion depth decreased by about 40%, which indicated a more minor lesion and reduced skin damage (Fig. 3L). Please provide sufficient information how the quantification of hyphal length in the PAS staining.

Thank you for your comments. After PAS staining, the hyphae of *C. albicans* in kidney sections appear dark purple. We have selected at least 20 long hyphae from each section to measure the hyphal length. After imaging by a 40× objective, we used the scalebar and the length measurement function in Image J to quantify. We described the profile of hyphae by lines, and then determined the length of the profile. The diagram is as follows:

Fig.7 The kidney sections stained by PAS. *C. albicans* hyphae was stained as dark purple. The dotted yellow lines were the profile described in Image J.

- Figure 4: In Figure S6, the authors show that all three mutants have reduced growth in liquid media suggesting a role in hyphal elongation which will affect secondarily penetration. Line 258: "However, it should be noted that TOR activity in HOPS-complex-deficient mutants was still maintained which promotes the normal proliferation of *C. albicans* cells, indicating that TOR activity is only partially impaired (Fig. S6A). Statement and the figure S6a don't match.

Thank you for your suggestion. The time-growth curves of the wild type *C. albicans*, *vam6Δ/Δ*, *vps41Δ/Δ* and *ypt72Δ/Δ* mutants in YPD medium were shown in Fig. S4f, where the four lines essentially align and show no significant difference. The time-growth assay indicated that the reduced TOR signaling activity in *vam6Δ/Δ*, *vps41Δ/Δ* and *ypt72Δ/Δ* mutants did not affect cell proliferation, so the shortening of hyphae was not caused by slow proliferation. Our original sentence exhibits ambiguity. According to your suggestion, we have changed this sentence to "Our results suggested that the reduced virulence in the HOPS-complex-deficient strains may not be caused by the decreased expression or secretion of virulence factors. Meanwhile, the impaired TOR activity did not reduce the normal proliferation of *C. albicans* (Fig. S4f), indicating that the phenotypes of the HOPS-complex-deficient mutants were not caused by the decreased growth." in line 328-331.

Fig. 8 Fig S4f in our revised manuscript. The growth curves in *C. albicans* mutants in liquid YPD medium. The *C. albicans* SC5314, *VAM6Δ/Δ*, *VPS41Δ/Δ* and *YPT72Δ/Δ* strains with initial OD₆₀₀ of 0.10 were cultured in YPD liquid medium at 200 rpm, 30 °C. And OD₆₀₀ was measured at 2 h, 4 h, 6 h, 8 h, 10 h, 12 h and 24 h.

Line 264: our results showed that 10 nM rapamycin, significantly inhibited the growth of *C. albicans* in Fig. 4A. 10nM rapamycin is not reported in Figure 4A and it seems 8nM rapamycin inhibits the *Candida* growth completely. Please clarify why 10nM rapamycin was used in Figure 4C. Also, Inconsistency for the spotted growth of *vam6*^{-/-} strain from Figure 4A and Figure E. Please clarify 3nM rapamycin treated *vam6* has growth in 3 dilutions but less concentration rap con at 2nm inhibits the growth with first dilution. Please clarify.

Thanks for your comments. The images concerning to the sensitivity of rapamycin have been updated. In Fig. 4a, YPD agar was used to determine the sensitivity of *C. albicans* to rapamycin. In Fig. 4c, Spider medium was used to examine the inhibitory effect of rapamycin on hyphal formation in SC5314. Here, we showed the morphology of colonies grown on the Spider agar. The reduced size of colonies indicated that the growth of *C. albicans* was reduced in the presence of 5 nM or 10 nM rapamycin. But by comparing the reduction of width (W) and depth (D) in colonies, we found the depth of clonal invasion was reduced by about 50% in response to 5 nM and 10 nM rapamycin, but the width was only reduced by about 20% and 30%. Therefore, we concluded that the inhibition of rapamycin on the hyphae invasion was more significant than that on proliferation.

Fig. 9 The morphology of colonies treated with 5 nM or 10 nM rapamycin in Spider agar.

The reason for the different growth of *C. albicans* in the presence of 3 nM or 2 nM rapamycin: The effective concentration of rapamycin is extremely low, capable of inhibiting the proliferation of *C. albicans* below 10 nM. In our experiments, a 1 mM storage solution was typically diluted to the working solution. However, due to the slight variations in the volume of medium used in each preparation, there might be potential dilution errors leading to differential inhibition under the theoretically identical concentrations. As your suggestion, we have updated representative pictures in Fig. 4a and 4e from our repeated experiments.

Fig. 10 Fig. 4a and 4e in revised manuscript. **a** Rapamycin sensitivity examined by spot assay. *C. albicans* strains were serial 5-fold diluted, spotted on YPD agar containing 1 nM, 3 nM and 5 nM rapamycin and cultured at 30 °C for 48 h. **e** Rapamycin sensitivity examined by spot assay. The wild type SC5314, *vam6Δ/Δ* and *GTR1^{OE}-vam6Δ/Δ* mutants were spotted on YPD agar containing 2.5 nM rapamycin and cultured at 30 °C for 48 h.

- Figure 5: Line 304-305”: Using continuous live cell imaging, we captured vacuolar fusion in hyphae. Especially in the hyphal tip, two vacuoles fused to form a larger one within 90 s (Fig. 5C).” No data provided showing this for the mutants.

Thanks for your suggestion, we have monitored the dynamic changes of fragmented vacuoles within 10 min in the hyphae of the *vam6* Δ/Δ , *vps41* Δ/Δ and *ypt72* Δ/Δ mutants. During this period, the small vacuoles migrated and changed their position, without fusing to form larger vacuoles. Figures and descriptions in the main text are as follows, “In contrast, there was no fusion of the fragmented vacuoles in the *vam6* Δ/Δ , *vps41* Δ/Δ and *ypt72* Δ/Δ mutants (Supplementary Video 5, 6 and 7).” in line 389-390.

- Figure 6: Fig 6d, 5×10^5 CFU/mouse via tail has same virulence for WT than with 1.6×10^6 . Please clarify. Mutant is more virulent than in Figure 3d which seems contradictory since the inoculum is 10 times lower in Fig 6d. Why did the author look at the fungal burden at day 2 while in Figure 3 they looked at day 5 with a higher inoculum. Should be consistent.

Thanks for your comments. According to your suggestion, in order to increase the comparability of our results, the survival and fungal load in female ICR mice infected with 1.6×10^6 CFU of *C. albicans* SC5314, *vam6* Δ/Δ and *GTR1*^{OE}-*vam6* Δ/Δ mutants have been performed and determined. The fungal load and PAS staining in kidneys were examined five days after infection. The repeated results are described as follows, “However, in the model of systemic *C. albicans* infection, mice infected with either *GTR1*^{OE}-*vam6* Δ/Δ or *vam6* Δ/Δ mutant exhibited higher survival rates and decreased fungal burdens in kidneys compared to the wild type group. No significant difference was observed among mice infected with *GTR1*^{OE}-*vam6* Δ/Δ or *vam6* Δ/Δ mutants (Fig. 6d-6e). Consistently, kidney sections stained by PAS showed that *GTR1*^{OE}-*vam6* Δ/Δ and *vam6* Δ/Δ mutants were unable to form elongated radial hyphae, and the extent of kidney damage caused by the mutants was significantly smaller than that caused by the wild-type SC5314 (Fig. 6f-6g).” in line 442-449.

Fig. 11 Fig 6d-6g in our revised manuscript. **d** The survival percentage of mice infected with *C. albicans*. Female ICR mice (n=10) were infected with SC5314, *vam6Δ/Δ* and *GTR1^{OE}-vam6Δ/Δ* mutants (1.6×10^6 CFU/mouse) via tail vein. The survival time of mice was observed for 3 weeks. **e** The fungal burden in kidneys of mice (n=5) were determined 5 days after fungal inoculation. Data were presented as mean \pm SD. **f** The kidneys of mice were taken out 2 days after fungal inoculation and made pathological sections with PAS staining. Scar bars = 100 μ m. **g** The hyphal length in kidneys was measured by Image J. At least 20 visible long hyphae were measured. Data were presented as mean \pm SD. n = 20.

Minor

-Figure 1A Please provide the scale bar and please provide sufficient information on how an image is acquired, magnification, and length was measured.

Thank you for your suggestion. We have added the scale bar of the images shown in Fig. 1a. Stained hyphae in Fig 1a were photographed with 1.5-fold magnification on a Axiovert 200 microscope with a 63 \times objective. The information of instrument and magnification have been added in the methods in page 25, line 710-713 “Stained strains were observed and photographed with 1.5-fold magnification on a Zeiss Axiovert 200 microscope with a 63 \times objective, or on a Leica TCS SP8 STED scanning microscope with 40 \times water-immersion objective.”. We used the scalebar and the length measurement function in image J software to quantify. We described the profile of hyphae by lines, and then determined the length of the profile. The diagram is as follows:

Fig. 12 Fig,1a in our new manuscript. The scale bar was added in the bottom right corner of the images. b. the diagram for measuring the length of hyphae.

The authors should provide the statistics for Figure 1d.

Thanks for your comments. We have selected six hyphae in a microscopic field and measured their length at each time point. The statistics has been analyzed by the mixed linear model calculated by SPSS.

Authors should provide detailed information for Figure 1e.

We have re-written the information in Fig 1e in line 124-130. “As shown in Fig. 1e, the defects of hyphae growth in the *vam6*-disrupted mutant on the solid media were more obvious than those in the liquid media. On solid RPMI 1640 plus 10% FBS medium, the wild-type colony formed obvious hyphal extension around the smooth center. In contrast, the hyphal extensions in the *vam6Δ/Δ* mutant were much shorter and fewer. Similarly, on Spider medium, unlike the parental strain forming wrinkled colonies with hyphal extension around the colony, the *vam6Δ/Δ* mutant only formed wrinkles without hyphal extension.”

-Line No 166: remove the word “function” verified colocalization of target proteins.

The word “function” has been deleted. The sentence has been revised to “The co-localization of VacuRed with GFP-Vam6, GFP-Vps41, and GFP-Ypt72 (Fig. 2f), on the one hand, confirmed the specific distribution of VacuRed on the vacuolar membranes, and on the other hand, verified the colocalization of three target proteins on the vacuole membrane to mediate the fusion of vacuoles.”. line 193-197.

-The representative image for VAM6^{-/-} and the bar graph does not match Figure 2G. Please quantify the length or provide an alternative image.

Thank you for your careful observations. The *vam6* Δ/Δ mutant did not produce uniform regions of hyphal extension in all directions on the solid agar, thus making it difficult to distinguish the boundaries of C and P. We have also checked for other colonies in the *vam6* Δ/Δ mutant. The C and P cannot be distinguished clearly in all of the colonies. Therefore, when determining the value of P, we only collected a certain edge based on the invasion area. In our revised manuscript, we have annotated the position of the measurements at the edge of the colony as you suggested.

Fig. 13 The P value of colonies culture in Spider agar.

-VacuRed has no reference and I could not find anything about this staining for vacuole.

Thank you for your comments. As the structure of VacuRed has not been patented, we cannot disclose the specific synthesis method and the structure of the probe in this article. If you want to know more detailed information, we would provide it to you privately. Thanks for your consideration.

The description of VacuRed has been changed to “To further image the structure of vacuoles, the probe VacuRed, a silicon-substituted coumarin-based fluorophore developed by our group^[1, 2], targeting fungal vacuoles was designed and synthesized.” in line 159-161.

Ref:

1. Li, C., Wang, T., Li, N., Li, M., Li, Y., Sun, Y., Tian, Y., Zhu, J., Wu, Y., Zhang, D., and Cui, X. Hydrogen-bonding-induced bathochromic effect of Si-coumarin and its use in monitoring adipogenic differentiation. 2019, Chem. Commun. 55, 11802-11805.
2. Li, C., Wang, T., Fan, M., Wang, N., Lin, X., Sun, Y., and Cui, X. Hydrogen Bond-Enhanced Nanoaggregation and Antisolvatochromic Fluorescence for Protein-Recognition by Si-Coumarins. 2022, Nano Lett. 22, 1954-1962.

- Line 217: “Overall, the HOPS complex is indispensable for active penetration and escape from phagocytes.” The use of the word indispensable is not accurate because there is no control

uninfected so indispensability cannot be determined. Participate could be use instead.

Thanks for your suggestion. The word “indispensable” has been deleted. The sentence has been revised to “Overall, the HOPS complex participates in active penetration through host cells.”. Line 261.

-Authors should clarify whether the microscope images provided in Figure 3b are 3h or 6h time points.

Thanks for your suggestion. The images provided in Fig. 3b was photographed by fluorescence microscopy at the 3 h time point. We have clarified this in figure legend as “Murine peritoneal macrophages were co-incubated with *C. albicans* (MOI=1) for 3 h, stained with 100 µg/ml of PI. The plate was photographed by fluorescence microscopy. Scar bars = 50 µm.” in line 1332-1334.

How do the authors differentiate PI-positive macrophage and PI+ *C. albicans*?

Thanks for your comments. In order to better distinguish the macrophages and *C. albicans*, the overlaid images were shown in Fig. 3b. The PI-positive nuclei of macrophages are much larger and thicker than those in the fungal hyphae. As shown in Fig. 3b, the tiny red dots in the rectangle boxes zoomed in represent the nucleus of *C. albicans* which were not counted as PI positive macrophages.

Fig. 14 Murine peritoneal macrophages were co-incubated with *C. albicans* (MOI=1) for 3 h, stained with 100 µg/ml of PI. The plate was photographed by fluorescence microscopy.

Reviewer #2 (Remarks to the Author):

The manuscript by Liu et al. describes the role of *C. albicans* orthologs of the yeast HOPS complex in hyphae initiation and penetration of solid agar. Consistent with the impaired hyphal penetration in HOPS knockout *C. albicans* strains, their virulence is reduced in co-incubation with human umbilical vein endothelial cells (HUVEC) and peritoneal macrophages (PM) and in mouse models. The authors observe that vacuoles are fragmented in HOPS knockout strains. Interestingly, the authors observe that the “interior of the hyphal tip was filled with a large vacuole”, which occupies the intracellular volume of the leading septum. Notably, in HOPS knockout strains (*vam6* Δ/Δ , *vps41* Δ/Δ , or *ypt72* Δ/Δ mutants) with a defect in vacuole fusion, these large vacuoles were absent from hyphae tips.

The study convincingly illustrates the role of the HOPS complex in *C. albicans* pathogenesis in human and mouse cells and mouse models. The manuscript presents an interesting hypothesis on the role of vacuole size in generating mechanical forces supporting hyphal penetration in *C. albicans*. However, the authors have not persuasively established that the defect in hyphal penetration in HOPS knockout *C. albicans* strains is due to vacuolar fragmentation and that large vacuoles indeed generate the mechanical forces required for hyphal penetration of *C. albicans*. To support their idea, the authors should show that abrogating vacuolar movement to the hyphal tip or inducing vacuolar localization to the tip is sufficient to change hyphal shape and penetration ability in solid media. Similarly, increasing or decreasing vacuolar size independent of HOPS function (such as by regulating vacuole fission) should be included to support the hypothesis that vacuolar size regulates the hyphal ability to penetrate.

Thanks for your comments. We have added more evidence about the large vacuoles generating the mechanical forces required for hyphal penetration in *C. albicans*.

1. In order to verify that large vacuoles are formed at growing tips of penetrating hyphae, we constructed Gfp-fused Yvc1 protein in the *vam6* null mutant. Yvc1 is a vacuolar Ca²⁺ channel protein, which locates on the vacuolar membrane in *C. albicans*. The in-situ observation of agar sections clearly revealed that certain hyphae invading into the agar exhibited the formation of large vacuoles at growing tips of hyphae. Combined with the inability to form large vacuoles in hyphae and the deficiency in penetrating solid agar in the HOPS-complex-deficient mutants, we argue that the formation of large vacuole is necessary to maintain hyphae penetration.

2. The medium of Spider plus Ca²⁺ or RPMI 1640 plus 10% fetal bovine serum (FBS) can induce different signal transduction during hyphae formation on solid plate. The hyphal length around the colony in the HOPS-complex-deficient mutants on the solid Spider plus Ca²⁺ or RPMI 1640 plus 10% FBS medium showed no relation to the nutrients in the culture medium. However, the HOPS-

complex-deficient mutants exhibited longer hyphae in medium containing less agar, indicating that the lack of large vacuoles mainly affect the mechanical force of hyphae to penetrate the solid agar, but not the signal transduction of hyphae extension.

3. The deletion of the HOPS complex mainly affects the fusion of small vacuoles, but has little effect on vacuolar function, suggesting that the deficiency of hyphae penetration and pathogenicity was mainly caused by the absence of large vacuoles.

4. The overexpression of *GTRI* restored the hyphal initiation and the ability of host cell damage in the *vam6* null mutants, but did not restore agar penetration or *in vivo* pathogenicity in mice. In contrast, our newly updated results showed that the *vac8* null mutants, with delayed hyphal initiation but normal agar penetration ability, was significantly more virulent for host cells and mice than the *vam6* null mutants. The comparison between the *GTRI*^{OE}-*vam6*Δ/Δ and *vac8*Δ/Δ mutants indicates that the hyphal penetration forces promoted by large vacuoles is the dominant factor in maintaining the pathogenicity of *C. albicans in vivo*.

According to your suggestion, we have added the main experimental evidence described above. We conclude that the HOPS complex mainly mediates vacuole fusion to generate large vacuoles without affecting the ability of expression or secretion of virulence factors. The decreased virulence in the HOPS-complex-deficient mutants is mainly due to the lack of mechanical forces provided by large vacuoles to promote hyphae penetration into agar and solid tissues, such as kidneys and skin.

-The data presented in Figs. 5a and b should be supported by quantification.

Thank you for your comments. We hope we have not misunderstood what you mean by quantification. We have counted the proportion of hyphae that contain large or fragmented vacuoles. In Fig. 5a, we can see vacuoles with larger diameters in almost every hyphal cell induced by liquid medium. Most of hyphae scraped from solid agar contained large vacuoles, although not all displayed large vacuoles at hyphal tips. For cells scraped from solid agar, most of them were yeast cells, with a minimal presence of hyphal cells. In order to enhance the robustness of our results, we employed the Yvc1-GFP strain in the revised manuscript to examine the vacuolar morphology in hyphal cells penetrated to the agar. We found that most of hyphae contained large vacuoles, while a subset of hyphae (indicated by yellow arrows) displayed swollen vacuoles at their tips. But the proportion of these hyphal cells was low. A small proportion of cells with swollen hyphal tips rendered our quantification exceedingly challenging due to the coexistence of numerous yeasts and pseudohyphal cells within the field. Therefore, the methodological error of quantification is expected to have a relatively significant impact.

The data presented in Fig. 5a and 5b were used to illustrate the morphology of vacuoles within

C. albicans hyphae induced in both liquid and solid media, providing evidence for the requirement of HOPS complex on the generation of enlarged vacuoles in hyphae. Although we did not adopt a quantification, we have included additional visual evidence in the Supplementary Fig. S5b in response to your suggestion. These images demonstrated that vacuoles predominantly exist as large vacuoles in the wild-type hyphae and primarily as fragmented forms in the HOPS-deficient mutants.

Fig. 1 Fig.5b in our revised manuscript. Imaging of Yvc1-GFP strains in hyphae sliced from Spider agar. *C. albicans* strains were spotted on Spider agar at 37 °C for 48 h, sliced into 2 mm thick, and then stained with 500 nM VacuRed for 10 min.

-The authors should rescue the HOPS expression in knockout strains and test whether the rescue strains show rescue of vacuolar size and localization at the hyphae tip and rescue of ability to penetrate solid media.

Thank you for your comments. The vacuolar morphology of *VAM6*, *VPS41* and *YPT72* revertant strains were showed in Fig. S2g and S2h. Our results demonstrated that the complement of these genes restored the presence of large vacuoles in both yeast and hyphae. Furthermore, we also observed that the vacuoles of the revertant strains localized at the hyphae tip, indicating that the rescue of HOPS complex restored the fusion of the fragmented vacuoles.

To validate the penetration of revertant strains, we investigated their hyphae growth on the solid agar. Either in RPMI1640+10%FBS or Spider medium containing 2% agar, all the three revertant strains restored the width and depth of colony effectively. The width (W) and depth (D) values of the revertant strains were similar to those of wild type *C. albicans*. And the depth of *vam6* Δ/Δ *VAM6*, *vps41* Δ/Δ *VPS41* and *ypt72* Δ/Δ *YPT72* was significantly larger than that of the null mutants.

Fig. 2 (a) The vacuolar morphology of HOPS deficient mutants and revertant mutants. (b) The hyphal morphology in Spider or RPMI 1640+10% FBS agar.

-Is the vacuolar localization to the hyphae tip also observed when *C. albicans* is co-cultured with HUVEC or macrophages? What happens in this experiment using HOPS knockout strains? This is an important experiment, as data from agar plates is not sufficient to conclude the role of vacuolar size in *C. albicans* infection under in vivo conditions.

Thanks for your constructive comments. According to your suggestion, we first stained *C. albicans* and HUVECs in the same confocal dishes with VacuRed dye. It turned out that the vacuoles of *C. albicans* cannot be stained by VacuRed in the presence of HUVECs. To better visualize the vacuolar morphology, we used the Yvc1-GFP strain. In *C. albicans*, *YVC1* encodes a calcium-channel protein that is abundantly localized on the vacuole membrane, which has been confirmed by the co-localization of VacuRed and Yvc1-GFP (Fig. 2g). Then, Yvc1-GFP strains were incubated with HUVECs. Solid bright spots were observed at the tips of certain hyphae that penetrated to HUVECs, indicating a fusion of numerous small vacuoles in this region. However, no conspicuous vacuolated vacuoles were observed at the hyphae tip. These results suggested that large vacuoles are not required to provide mechanical force to penetrate the cell membrane. The damage caused by *C. albicans* on HUVECs is mainly related to the slow initiation and elongation of hyphae in liquid medium.

C. albicans - HUVECs co-incubation

Yvc1-GFP

Fig. 3 Fig. S5a in our revised manuscript. The imaging of vacuoles during hyphal growth. *C. albicans* Yvc1-GFP was co-incubated with HUVECs for 3 h. Red arrows indicate the vacuoles of hyphae.

These results were also consistent with the conclusions obtained from the strains *GTR1^{OE}-vam6Δ/Δ*. In Fig. 6b and 6c, we found that *GTR1* overexpression significantly restored the damage of *vam6Δ/Δ* to HUVECs and macrophages. In terms of hyphae growth, overexpression of *GTR1* restored initiation and elongation rates in liquid medium, but did not fully restore the ability of hyphae penetration to the wild-type strains in solid agar. The results of *GTR1^{OE}-vam6Δ/Δ* also suggested that the mechanical force provided by large vacuoles is not essential for *C. albicans* penetration to cell membrane in liquid medium.

Fig.4 Fig. 4f, 4g, 6b, 6c and 4h in our revised manuscript. **a** The ratio of germination in *C. albicans* cultured in hyphal-inducing liquid media at 37 °C for 0.5 h. The ratio of germinated strains to all the strains observed in the microscopic field (approximately 50-100 CFU in each field under the 40× objective) was considered as the ratio of germination. The hyphal length quantified by Image J. *C. albicans* strains were cultured in RPMI 1640 + 10% FBS or Spider medium at 37 °C for 2 h. **b** The percentage of PI-positive murine peritoneal macrophages co-incubated with *C. albicans* for 6 h (MOI=1). Three biological replicates were carried out. **c** The LDH release rate of HUVECs co-incubated with *C. albicans* for 9 h (MOI=1). Three biological replicates were carried out. **d** The hyphal morphology of *C. albicans* strains on solid medium. *C. albicans* strains were spotted on the Spider agar and cultured at 37 °C for 5 days. The edge and the vertical section of a hyphal colony were observed by microscopy. Three biological replicates were carried out. One of the representative images are shown. Width (W) and depth (D) were quantified by Image J.

In our study, we found that the mechanical forces generated by large vacuoles mainly play a role in

penetrating solid agar or solid organ. First, we observed the large vacuoles of Yvc1-GFP at the hyphal tips in agar sections, but not in hyphae penetrating HUVECs (Fig. 5b and S5a in manuscript). Second, the overexpression of *GTRI* restored the hyphal initiation and the ability of host cell damage in the *vam6* null mutants, but did not restore agar penetration or *in vivo* pathogenicity in mice (Fig. 6 in manuscript). In contrast, our newly updated results showed that the *vac8* null mutants, with delayed hyphal initiation but normal agar penetration ability, was significantly more virulent for host cells and mice than the *vam6* null mutants (Fig.7 in manuscript). The comparison between the *GTRI*^{OE}-*vam6*Δ/Δ and *vac8*Δ/Δ mutants indicates that the hyphal penetration forces promoted by large vacuoles is the dominant factor in maintaining the pathogenicity of *C. albicans in vivo*. Collectively, our results suggest that large vacuoles can form at the hyphae tips when penetrating solid media and provide a supporting force to promotes hyphae invasion.

Fig. 5 Fig. 5b in our revised manuscript. Imaging of Yvc1-GFP strains in hyphae sliced from Spider agar. *C. albicans* strains were spotted on Spider agar at 37 °C for 48 h, sliced into 2 mm thick, and then stained with 500 nM VacuRed for 10 min.

Besides this major concern, there are some specific suggestions about some of the experiments and their presentation. The author can consider adding these suggestions to strengthen their major conclusions:

1) The *VAM6*, *VPS41*, and *ypt72* revertant strains should also be employed in Figure 3 to show rescue of *C. albicans* pathogenesis in cells or a mouse model.

Thank you for your comments. We have employed the role of *VAM6*, *VPS41*, and *YPT72* revertant strains in the cell and mouse models. All these results were showed in Fig.S3. After 3 hours and 6 hours of co-incubation, the percentage of peritoneal macrophages positive for PI was comparable between those co-incubated with the wild type *C. albicans* and those exposed to revertant strains. The PI positive rates were significantly higher than that of macrophages co-incubated with the HOPS-complex-deficient mutants.

Fig. 6 Fig. S3a and S3b in our revised manuscript. **a** The percentage of PI-positive murine peritoneal macrophages cells co-incubated with *C. albicans* strains for 3 h or 6 h (MOI=1). Three biological replicates were carried out. **b** Murine peritoneal macrophages were co-incubated with *C. albicans* (MOI=1) for 3 h, stained with 100 $\mu\text{g/ml}$ of PI. The plate was photographed by fluorescence microscopy.

In the systemic infection model, the survival of mice infected with wild-type and revertant strains was comparably indistinguishable, as all mice died within 4-9 days. Similarly, there was no significant difference in the fungal load in organs between mice infected with wild-type and the revertant strains, neither in kidneys of systemic infection nor in the skin of superficial infection. Taken together, these results indicated that the complement of *VAM6*, *VPS41*, and *YPT72* fully restore the pathogenicity of *vam6Δ/Δ*, *vps41Δ/Δ* and *ypt72Δ/Δ* mutants.

Fig 7 Fig. S3c-S3i in our revised manuscript. **c** The survival of mice infected with *C. albicans* strains. Female ICR mice were infected with *C. albicans* strains (1.6×10^6 CFU/mouse) via tail vein to conduct disseminated candidiasis model. The survival time of mice ($n=10$) was observed for 3 weeks. **d** The fungal burden in kidneys of mice ($n=5$) were determined 2 days after fungal inoculation. **e** The kidneys of mice were taken out 5 days after fungal inoculation and made pathological sections with PAS staining. The hyphal length in kidneys was measured by Image J. At least 20 hyphae were measured. **f** The exposed back skin with fungal-infected damage on day 7. **g** The fungal burden in damaged skin of mice ($n=5$) were determined 3 days after fungal inoculation. **h** The pathological images of infected skin stained by PAS. The dimension of skin damage was

quantified by the maximum depth of penetrated hyphae measured by Image J. **i** The hyphal length in damaged skin was measured by Image J. At least 20 hyphae were measured.

2) The description for Figs. 3B and 3C is not clear as to how the authors conclude using propyl iodide (PI) staining that the HOPS complex is required for escape from phagocytes. These lines from 209–217 need to be rewritten for better explanation.

Thanks for your comments. We have changed this sentence to “These data suggested that the HOPS complex was also required for maintenance the ability to kill macrophages. Overall, the HOPS complex participates in active penetration through host cells.” Line 259-262.

3) Fig. 5C: Live imaging of hyphal dynamics in the HOPS knockout strain should also be shown along with the WT strain to show whether vacuole fusion is observed or not in the hyphae in the HOPS knockout strains.

Thanks for your suggestion, we have monitored the dynamic changes of fragmented vacuoles within 10 min in the hyphae of the *vam6Δ/Δ*, *vps41Δ/Δ* and *ypt72Δ/Δ* mutants. During this period, the small vacuoles migrated and changed their position, without fusing to form larger vacuoles. Figures and descriptions in the main text are as follows, “In contrast, there was no fusion of the fragmented vacuoles in the *vam6Δ/Δ*, *vps41Δ/Δ* and *ypt72Δ/Δ* mutants (Supplementary Video 5, 6 and 7).” in line 389-390.

4) The description of Fig. 5F and G (lines 328–338) is unclear and needs to be rewritten. The graph shown in Fig. 5G should be described with more clarity in the text.

Thanks for your comments. We have rewritten the description of Fig 5F and 5G. “As shown in Fig. 5g, we adjusted the agar concentrations to 0.5%, 1%, and 2% in Spider medium with 10 mM calcium, respectively. With the concentrations of agar increased, the invading hyphae of all strains were sparser, and the length of hyphae around the colony was slightly reduced. The peripheral region (P) and central region (C) value of different colonies were shown in Fig. S5d and the ratios of peripheral region (P) of HOPS-complex-disrupted mutants to that of the wild type *C. albicans* at the same agar concentration were shown in Fig. 5h. In the *vam6Δ/Δ*, *vps41Δ/Δ*, and *ypt72Δ/Δ* mutants, the deficiency of hyphal invasion in 2% agar is significantly larger than that in 0.5% agar. Similar inverse relationship between the length of hyphal extension and agar concentration was further demonstrated in RPMI1640 plus 10% FBS medium supplemented with 0.5% to 2% agar. The hyphal invasion ratio in the *vam6Δ/Δ*, *vps41Δ/Δ*, and *ypt72Δ/Δ* mutants had greater deficiencies in 2% agar than in 0.5% or 1% agar (Fig. 5i and 5j, S5e).” in line 413-425.

Fig. 8 Fig.5g-5j in our revised manuscript. **g** Top view of hyphal colony on Spider + 10 mM CaCl₂ solid plates containing different concentrations of agar. *C. albicans* were spotted on Spider solid medium containing 0.5 %, 1 % or 2 % agar. Three biological replicates were carried out. One of the representative images are shown. **h** The hyphae invasion ratio calculated by PHOPS-complex-deficient mutants/*P*_{SC5314} in the same concentrations of agar. Parameters C and P of hyphal colonies in **g** were measured by Image J. **i** Top view of hyphal colony on RPMI 1640 + 10% FBS solid plates containing different concentrations of agar. *C. albicans* were spotted on RPMI 1640 + 10% FBS solid medium containing 0.5 %, 1 % or 2 % agar. Three biological replicates were carried out. One of the representative images are shown. **j** The hyphae invasion ratio calculated by PHOPS-complex-deficient mutants/*P*_{SC5314} in the same concentrations of agar. Parameters C and P of hyphal colonies were measured by Image J.

5) The mTORC phosphorylation activity (as shown in Fig. 4B) should be tested in the VAM6, VPS41, and ypt72 revertant strains to support the rapamycin spot assay data shown in Fig. S4.

Thank you for your comments. We have examined the activation of the TOR pathway in the revertant mutants after nitrogen starvation followed by the addition of arginine and ammonium sulfate. As shown in Fig. S4, the phosphorylation of S6 was deficient in *vam6Δ/Δ*, *vps41Δ/Δ* and *ypt72Δ/Δ* mutants compared with wide type *C. albicans*. However, in the *VAM6*, *VPS41* and *YPT72* revertant strains, the activation of pS6 were similar to that of wide type *C. albicans*, suggesting that rescue of the HOPS complex could efficiently promote the activation of TOR signaling.

Fig. 9 Fig. S4c and S4d in our revised manuscript. The phosphorylation of RPS6 protein (P-S6) examined by Western blotting. *C. albicans* strains were cultured in SD medium without nitrogen for 4 h and then stimulated for 60 min with the addition of 40 mM ammonium sulfate or 10 mM arginine.

6) Zoom-in images should be shown in Fig. 5B for clarity.

Thank you for your suggestion. We have showed the zoom-in images of vacuoles in the hyphae tips in our revised manuscript (Fig. 5B). The enlarged vacuoles in the hyphae tips of the wild-type strain and the fragmented vacuoles in HOPS complex deficient mutants were clearly observed.

Fig. 10 Fig. 5c in our revised manuscript. Vacuoles stained by VacuRed in hyphae scraped from Spider agar. *C. albicans* were spotted on Spider agar at 37 °C for 48 h, scraped, and then stained with 500 nM VacuRed for 10 min.

7) Line 142-143 states that “Meanwhile, Vam6 (Vps39), Vps41, Ypt72, and Class C proteins form the HOPS complex.” However, Ypt72 in Fig. 2a appears to be a Rab7 ortholog and not part of the hexameric HOPS complex. Ypt72 should be introduced properly in the manuscript result section.

Thanks for your suggestion. We have reintroduced the Ypt72 as “To test this hypothesis, one of the components of HOPS complex, *VPS41*, and the Rab GTPase *YPT72* in *C. albicans* were knocked out, respectively.” in line 153-154.

8) In the “Supplementary methods for strain construction” section, it states that “the presence of the product of primers LY22-17 and LY20-11 and the absence of the product of primers LY22-18 and LY22-18 were used to evaluate the correct strain by genomic PCR.”

Thanks for your comments. The primer names should be corrected to LY22-17 and LY-22-18 and the methods and primers have been corrected.

Reviewer #3 (Remarks to the Author):

The authors study the role of VAM6, VPS41, and YPT72 which encode components of the HOPS complex in *Candida albicans*. Here they demonstrate that mutants lacking these genes are defective in filamentation, particularly on solid media, and virulence in vivo. Rescue of the filamentation defect in the *vam6* mutant by overexpression of small GTPase encoded by GTR1, which encodes a subunit of the TORC1-activating EGO269 complex, did not restore hyphal penetration. Taken together, these results demonstrate an association between vacuolar integrity and the ability to invade solid agar, as an indirect measure of hyphal turgor pressure.

Thank you for your comments. According to your suggestions, we have added more evidence to confirm the requirement of large vacuoles to penetrate solid media. For the specific experiments we supplemented, please see our modified script and the response to major concerns.

Major Concerns:

The Introduction should be presented more comprehensively with additional details. For example, in line 62, Noble et al found that virulence and hyphal morphogenesis were de-coupled, indicating that virulence and morphogenesis are related in a complex, non-linear process. Also, in line 58, the transcriptional pathways in morphogenesis are listed but not described. In line 75, key proteins involved in morphogenesis on solid surfaces are also listed but not described. A broader and more detailed background to provide substantive context is recommended.

Thank you for your comments. We have added more specific information in the Introduction line 53-81. "Early studies of the ability of yeast and hyphal form switching as a virulence factor focused on the pathogenicity of *C. albicans* mutants defective in hyphal growth. The cyclic adenosine monophosphate/protein kinase A (cAMP/PKA), mitogen-activated protein kinases (MAPK), and pH pathways have been identified as the main signaling transduction cascades to regulate the yeast-hyphae transition. Downstream targets of these pathways are transcriptional activators including Efg1, Cph1, Tec1, Flo8 and Rim101, which result in hyphal-specific gene transcription. The cell cycle arrest Hgc1/Cdc28 pathway suppresses the degradation of the septin ring, promoting hyphal growth. Moreover, Tup1, Nrg1 and Rfg1 are repressors of filamentous growth. A large number of mutant libraries were used to screen for deficiencies in hyphal growth. For instance, Noble *et al.* have identified 115 virulence-attenuated *C. albicans* mutants. Among which, 53 mutants showed defects in morphogenesis, 18 mutants had slow proliferation, and 46 mutants exhibited deficiency in infectivity without defects in morphogenesis or proliferation, indicating that virulence and morphogenesis are related in a complex, non-linear process. In addition, many virulence factors associated with hyphal morphogenesis have been identified, such as GPI-anchored proteins,

degradative enzymes, and cell surface adhesins, among which the identification of candidalysin (Ece1) was influential². The invasive *ece1*Δ/Δ hyphae do not damage epithelia because the lack of candidalysin causes an inability to destroy cellular membranes.

In addition to studies on signal transduction and the function of virulence factors, mutants with different hyphae formation abilities in liquid and solid media have been of interest. The Blankenship group screened 124 *C. albicans* mutants *in vitro* for distinct filamentation induction conditions and concluded that filamentation in solid versus liquid media represent distinct biological and transcriptional programs. Subsequent studies showed that morphogenesis on solid surfaces depends on several proteins, such as Cwt1, Tpk1, Dfi1, Cek1, and Dck1/Rac1. These genes are required for invasive filamentous growth on agar but do not influence hyphae formation in liquid media. Notably, null mutants of some genes showed varying degrees of reduction in virulence in mouse models of systemic infection. For example, the virulence of the *cwt1* null mutant was not reduced, while the *dfi1* null mutant or the *cek1* null mutant was significantly attenuated.”

Since individual colonies can vary substantially, were the agar invasion assays (filament length, width/depth) done in multiple instances? If not, how were the presented colonies chosen as representative samples?

Thank you for your comments. Invasion hyphae on the solid agar were not characterized by plating the fungal suspension to obtain individual colonies. In order to avoid the differences between individual colonies, spot assays were used. Each mutant was adjusted to a concentration of OD₆₀₀ of 0.1, and then 1 μl of fungal suspension (about 1×10³ CFU) was dropped on the agar. As shown in Fig. 1, this operation resulted in more uniform colony morphology for the various spots on the same plate, which significantly reduced the errors in determined widths and depths. According to your suggestion, we have incorporated the data from multiple colonies and generated several bar graphs to effectively illustrate the disparity between wild-type and the HOPS-complex-deficient mutants.

Fig. 1 The morphology of colonies growth on the Spider + CaCl₂ agar.

The GTR1OE-*vam6* mutant seems to produce a somewhat incompletely restored phenotype, specifically, it does not appear to fully restore the filamentation defect of the *vam6* null mutant. Could this incomplete restoration be a contributing factor to the tissue penetration and virulence defect, either separately or in conjunction with the vacuolar defect? Can the authors more fully discuss this possibility? Are the filamentation defects fully rescued in the complemented control strains?

Thanks for your comments. We have revised our discussion in line 580-657. We systematically discussed the possible relationship among TOR pathway, hyphal initiation, and hyphal penetration ability in the maintenance of *C. albicans* virulence. “Overexpression of *GTR1* compensated for the reduced TOR activity as well as hyphal initiation and cell damage caused by hyphae, but it did not fully restore the filamentation defect in the *vam6* null mutant penetrating to agar or solid organs, such as kidney or skin. In contrast, the disruption of *VAC8*, which encodes a gene required for vacuolar movement, resulted in slower germination and hyphae growth in liquid medium. To our surprise, *vac8Δ/Δ* also formed large vacuoles in hyphae when penetrating agar. For penetrating solid kidneys and skin, the *vac8Δ/Δ* mutant also showed no obvious defect, and its pathogenicity *in vivo* was significantly higher than that of the HOPS-complex-deficient mutants. The opposite phenotypes of *GTR1*^{OE}-*vam6Δ/Δ* and *vac8Δ/Δ* mutants suggests that the production of the HOPS-complex-

mediated large vacuoles plays a more dominant role in maintaining *C. albicans* pathogenicity than hyphae initiation. The HOPS-complex-deficient mutants did not show significant difference in secretion and expression of virulence factors, so it was suggested that the large vacuoles may provide the mechanical support for hyphae penetrating the solid matter. In our study, the deficiency of vacuolar support is mainly manifested by the impaired hyphal invasion to the solid agars. With the decrease of agar concentration, the invasion zone of hyphae on solid agar was increased, which indirectly reflect the forces of hyphal penetration. When compared with the wild-type *C. albicans* invasion zones at the equivalent agar plates, the penetration deficiency of HOPS-complex-disrupted mutants in 0.5% agar were significantly smaller than that in 2% agar. These results suggest that the ability of *C. albicans* invading agar is mainly affected by the mechanical support provided by large vacuoles. Some studies have reported the driving forces and turgor pressure produced by vacuolar expansion. In *Basidiobolus ranarum*, the driving forces produced by enlarging vacuole pushes cytoplasm into developing conidia, which eventually break through the spore walls and disperse into the air. In the appressorium cell of *Magnaporthe grisea*, large vacuoles were also required for generating the massive turgor pressure to invade plant. The turgor pressure could promote *M. grisea* to penetrate plant tissue via the infection peg, which indicated the large mechanical forces provided by vacuoles. The turgor pressure was derived from the accumulation of glycerol in large vacuoles. In addition, plant vacuolar protein PIEZO could sense mechanical forces and modulate vacuole morphology. Many studies indicate that vacuoles are indispensable for sensing and regulating mechanical forces in fungi. Our study mainly demonstrates the necessity of large vacuoles in *C. albicans* hyphae to penetrate solid agar and organs. However, the detection and modulation of the vacuolar turgor pressure in *C. albicans* remains to be further investigated.”

Additional images with more views of SC5314 vacuoles for Fig. 5B would be of benefit. The current image shows only a single hyphal element with stained vacuoles, thus rendering the figure more difficult to interpret.

Thank you for your comments. Because of the long hyphae and the fact that most of fungal cells scraped from the surface of colonies were yeast cells, we have only presented one or two hyphae in representative fields. In the hyphae we observed, not all the hyphal cells were filled with large vacuoles at the tips. Since vacuoles are dynamically changing with fusion and division, not all hyphae showed enlarged tips. And our images are primarily designed to demonstrate the presence of large vacuolar enlargement in hyphal tips. Based on your suggestion, we realized that the presence of only one hyphal cell per field of view may lead to potential misinterpretations. Therefore, we provide a detailed explanation in the main text (line 377-378).

“Consistent with the hyphae scraped from agar, the tips of some but not all elongated hyphae were distributed with large vacuoles”

In addition, we also provided other field of view as shown in Fig. 5b. To better visualize the morphology of vacuoles in agar, a strain with GFP-tagged the calcium channel protein Yvc1 was used. Yvc1 located on the vacuolar membrane. When the agar was sectioned and imaged by Yvc1-GFP, we found that the tips of some hyphae, but not all hyphae, were enlarged and distributed with large vacuoles. This result further clarifies that the large vacuoles fused at the hyphae tips during penetration to solid agar.

Fig. 2 Fig. 5b in our revised manuscript. Imaging of Yvc1-GFP strains in hyphae sliced from Spider agar. *C. albicans* strains were spotted on Spider agar at 37 °C for 48 h, sliced into 2 mm thick, and then stained with 500 nM VacuRed for 10 min.

The Discussion should provide more information and context on the mechanical role of vacuolar function and filamentation in *Candida* species, and importantly to also include studies in filamentous fungi such as *Neurospora* or several plant pathogens.

Thanks for your comments. We have added information in discussion line 598-612 “These results suggest that the ability of *C. albicans* invading agar is mainly affected by the mechanical support provided by large vacuoles. Some studies have reported the driving forces and turgor pressure produced by vacuolar expansion. In *Basidiobolus ranarum*, the driving forces produced by enlarging vacuole pushes cytoplasm into developing conidia, which eventually break through the spore walls and disperse into the air³. In the appressorium cell of *Magnaporthe grisea*, large vacuoles were also required for generating the massive turgor pressure to invade plant. The turgor pressure could promote *M. grisea* to penetrate plant tissue via the infection peg, which indicated the large mechanical forces provided by vacuoles. The turgor pressure was derived from the accumulation of glycerol in large vacuoles⁴. In addition, plant vacuolar protein PIEZO could sense

mechanical forces and modulate vacuole morphology⁵. Many studies indicate that vacuoles are indispensable for sensing and regulating mechanical forces in fungi. Our study mainly demonstrates the necessity of large vacuoles in *C. albicans* hyphae to penetrate solid agar and organs. However, the detection and modulation of the vacuolar turgor pressure in *C. albicans* remains to be further investigated.”

The role of degradative enzymes in tissue penetration should also be mentioned, since they play a large role in tissue invasion. It would be straightforward to perform some simple indirect agar plate assays to assay for secreted aspartyl protease secretion, for example, in these mutants. In addition, the role of surface adhesins, which provide anchoring for *Candida* to form hyphae and invade tissues, should be mentioned, since defects in vacuolar regulatory proteins could potentially alter expression or trafficking of various secreted proteins. It would be very instructive to also include a comparison with hyphal-defective mutants unrelated to the vacuole. Overall, a tantalizing potential association between vacuolar morphology/function and hyphal penetration is revealed, but further experiments are needed to demonstrate causality related to the vacuole and turgor pressure.

Thank you for your comments. These suggestions you provided greatly contribute to understanding the mechanism underlying vacuolar fusion. The alteration of vacuolar morphology may affect its function.

According to your suggestion, we have examined the secretion of secretory aspartic proteases (SAPs) using YCB+BSA agar. As shown in Fig. S4a, the halo around the colonies were cause by SAP-mediated BSA hydrolysis. By measuring the diameters, we found that there was no significant difference between the wild type *C. albicans* and the HOPS-complex-deficient mutants. These results indicated that the diminished pathogenicity of *vam6Δ/Δ*, *vps41Δ/Δ* and *ypt72Δ/Δ* is not directly associated with the secretion of SAPs.

Fig. 3 Fig. S4a in our revised manuscript. SAPs secretion assay. *C. albicans* were spotted on the YCB-BSA agar and cultured at 30 °C for 48 h. The colonies morphology of *C. albicans* were imaged by camera. The size of each halo was measured by Image J.

In addition to examining the effect of HOPS complex on SAPs secretion, we also examined the

expression of virulence-related genes during co-incubation of *C. albicans* with human umbilical vein endothelial cells (HUVECs). The major up-regulated genes were *ALS3*, *HWPI*, *ECE1*, and *SAP6*, when *C. albicans* was incubated with RPMI1640+10% FBS at 37 °C with or without HUVECs co-incubation. And, the HOPS-complex-deficient mutants also up-regulated the expression of these genes, which was similar to that of the wild type *C. albicans*. Our results indicated that the HOPS-complex deficiency did not affect the expression of some important virulence genes during *C. albicans* invasion.

Fig.4 Fig.S4e in our revised manuscript. The expression of *ALS1*, *ALS3*, *ALS5*, *HWPI*, *ECE1*, *SAP2* and *SAP6* examined by qRT-PCR. Fungal mRNA was extracted from *C. albicans* incubated with/without HUVECs (MOI=1) cultured in DMEM medium for 1.5 h or 3 h.

According to your suggestion, we also investigated the vacuolar morphology and virulence of yeast-blocked *efg1/cph1Δ/Δ* mutants. The *efg1/cph1Δ/Δ* mutant exhibited normal large vacuoles through VacuRed staining. The pathogenicity of *efg1/cph1Δ/Δ* mutant was assessed in a systemic *C. albicans* infection model. Mice infected with 1.6×10^6 CFU of the *efg1/cph1Δ/Δ* mutant showed a survival rate of 100% and no detectable fungal load in their kidneys after being infected for 5 days. These results indicate that the pathogenicity of yeast-blocked *efg1/cph1Δ/Δ* mutants is lower than that of HOPS-complex-deficient mutants. We did not show the data in our main text, but we referred these results in our revised discussion.

“During determining the *in vivo* virulence of *C. albicans* mutants, we also used the yeast-blocked *efg1/cph1Δ/Δ* mutants, which showed normal vacuolar morphology in yeast cells. However, mice infected with 1.6×10^6 CFU of *efg1/cph1Δ/Δ* mutants did not show any abnormal locomotion and death during systemic infection. There was also no detectable fungal load in the kidney (data not shown). Compared with the yeast-blocked *efg1/cph1Δ/Δ* mutants, HOPS-complex-deficient

mutants is more virulent. But our study indicates that in addition to secreting virulence factors, hyphae can also cause tissue damage through mechanical penetration, which suggested that targeting vacuolar fusion could be explored as a novel antifungal strategy to reduce damage to host tissues.” in line 646-657.

To further confirm the important role of large vacuoles, the disruption of *VAC8* was constructed in our study (S6a-S6b). As show in Fig 7a, the hyphal germination rate of *vac8Δ/Δ* mutants was lower than that of wild-type *C. albicans* after a 40-minute induction period, similar to that observed in the *vam6Δ/Δ* mutants. However, after being induced for 4 hours, the length of hyphae formed by the *vac8Δ/Δ* mutants was longer compared to that in the *vam6Δ/Δ* mutants (Fig. 7b-7c). For agar penetration, the initial behavior of *vac8Δ/Δ* mutants on solid agar resembles that of *vam6Δ/Δ*. After 48 hours of cultivation, the invaded depth of *vac8Δ/Δ* and *vam6Δ/Δ* was significantly smaller than that of wild-type *C. albicans*. But to our surprise, when the cultivation time extended to 5 days, the *vac8Δ/Δ* mutant exhibited a significantly greater penetration depth compared to the *vam6Δ/Δ*, indicating that mechanical penetration of *vac8Δ/Δ* was not significantly impaired (Fig. 7d-7e).

The vacuolar morphology in the *vac8* null mutant were then explored. In consistence with the previous studies, in the hyphae induced for 1.5 h in liquid RPMI 1640 plus 10%FBS medium, the disruption of *VAC8* resulted in an augmentation of vacuole volume in yeast cells, while both the number and volume of vacuoles decreased in the hyphal tubes. However, when the hyphae were induced for 4 hours, there were no significant differences observed in the number or size of hyphal vacuoles between the *vac8Δ/Δ* mutants and wild-type *C. albicans* (Fig.7f). The *vac8Δ/Δ* mutants scraped from solid agar were also filled with large vacuoles in the hyphal tubes and tips (Fig. 7g). These results were contrary to our initial hypothesis that the *vac8* null mutants would lack apical vacuoles, but our findings suggested that the newly synthesized and fused large vacuoles within the hyphae were efficient to facilitate the hyphae extension and penetration into the agar.

The pathogenicity of the *vac8Δ/Δ* mutant was consistent with the agar-invasive phenotypes. There was no apparent reduction in its virulence compared to the wild-type *C. albicans*. In the *C. albicans*-peritoneal macrophages co-incubation assays, the proportion of PI positive cells damaged by *vac8Δ/Δ* was not different from that by the wild type *C. albicans*, but was higher than that damaged by *vam6Δ/Δ* (Fig. 7h-7j). In the systemic infection model, mice infected with *vac8Δ/Δ* from the tail vein had a survival rate of approximately 30% within 21 days, which was higher than that in mice infected with the wild type *C. albicans* (Fig. 7k), yet lower than that in mice infected with the HOPS-complex-deficient mutants (Fig. 3d). These results indicated that the pathogenicity of the *vac8Δ/Δ* mutant was reduced compared with that of wild type *C. albicans*, but was still stronger than that of

the HOPS-complex-deficient mutants. The renal fungal load did not show significant difference between *vac8Δ/Δ* and the wild type *C. albicans* infected mice for 5 days (Fig. 7l). Consistently, the length of hyphae in kidneys of mice infected with *vac8Δ/Δ* mutant was similar to that of mice infected with wild type *C. albicans* (Fig. 7m-7n). Subsequently, we assessed the pathogenicity and invasion of *vac8Δ/Δ* in the murine skin infection model. Notably, no significant difference was observed in the dorsal skin lesions and fungal load within the skin homogenates between mice infected with *vac8Δ/Δ* and the wild type *C. albicans* (Fig. 7o and 7p). Through the PAS staining, we found that the hyphae length of *vac8Δ/Δ* mutants invading to the skin was similar to the wild-type *C. albicans* (Fig. 7q and 7r). Collectively, although the hyphae initiation was slowed, but the ability to penetrate agar was not significantly reduced in the *VAC8*-disrupted mutant. As a result, the lethality of *vac8Δ/Δ* in systemic infection models was reduced in comparison with the wild type strain, but its pathogenicity was still higher than that of the HOPS-complex-deficient strains. The results of *GTR1^{OE}-vam6Δ/Δ* (normal hyphae initiation but defect in hyphae penetration) and *vac8Δ/Δ* (slowed hyphae initiation but normal in hyphae penetration) mutants indicated that both hyphae initiation and penetration are required for the pathogenicity of *C. albicans*, but the large vacuoles produced by vacuole fusion play a decisive role in maintaining the penetration of *C. albicans* into solid tissues.

Fig.5 Fig.7 in our revised manuscript. **The *vac8* null mutant with large vacuoles showed normal ability to penetrate solid media and slightly decreased virulence.**

a The ratio of germination in *C. albicans* cultured with hyphal-inducing liquid media at 37 °C for

40 min. The ratio of germinated cells to all the cells observed in the microscopic field was considered as the ratio of germination. In each field, approximate 50-100 CFU were observed under the 40× objective. Data were presented as mean ± SD; n = 6. **b-c** The hyphal length quantified by Image J. *C. albicans* were cultured in RPMI 1640 + 10% FBS or Spider medium at 37 °C for 2 h (b) or 4 h (c). At least 50 separate hyphae were measured. Data were presented as mean ± SD. n =50. **d-e** Vertical sections of hyphal colonies on solid agar. *C. albicans* SC5314, *vam6Δ/Δ* and *vac8Δ/Δ* mutants were cultured on RPMI 1640 medium with 10% FBS at 37 °C for 5 days. Three biological replicates were carried out. One of the representative images are shown. Colony Width (W) and depth (D) were quantified by Image J. Data were presented as mean ± SD. n = 3. **f-g** Vacuoles in hyphae of *C. albicans* cultured in liquid and solid media. *C. albicans* SC5314 and *vac8Δ/Δ* strains were cultured in liquid RPMI 1640 + 10% FBS medium for 1.5 h or 4 h and on Spider agar for 48 h at 37 °C. Hyphae in liquid medium or scratched from agar were stained with 500 nM VacuRed for 10 min. Scar bars = 10 μm. **h** Murine peritoneal macrophages were co-incubated with *C. albicans* (MOI=1) for 3 h, stained with 100 μg/ml of PI. The plate was photographed by fluorescence microscopy. Scar bars = 50 μm. **i-j** The percentage of PI-positive murine peritoneal macrophages cells co-incubated with *C. albicans* strains for 3 h or 6 h (MOI=1). Five biological replicates were carried out. Data were presented as mean ± SD. n = 5. **k** The survival percentage of mice infected with *C. albicans*. Female ICR mice (n=10) were infected with SC5314 and *vac8Δ/Δ* mutants (1.6×10^6 CFU/mouse) via tail vein. The survival time of mice was observed for 3 weeks. **l** The fungal burden in kidneys of mice (n=5) were determined 5 days after fungal inoculation. Data were presented as mean ± SD. **m** The kidneys of mice were taken out 2 days after fungal inoculation and made pathological sections with PAS staining. Scar bars = 100 μm. **n** The hyphal length in kidneys was measured by Image J. Data were presented as mean ± SD. n = 35. **o** The exposed back skin with fungal-infected damage on day 7. **p** The fungal burden in skin of mice (n=5) were determined 3 days after fungal inoculation. Data were presented as mean ± SD. **q** The pathological images of infected skin stained by PAS. The dimension of skin damage was quantified by the maximum depth of penetrated hyphae measured by Image J. Scar bars = 100 μm. **r** The hyphal length in damaged skin was measured by Image J. Data were presented as mean ± SD. n = 25. *p < 0.05; **p < 0.01; *** p < 0.001; **** p < 0.0001. Log-rank (Mantel-Cox) test (**k**), Unpaired t-test (**a**, **b**, **c**, **e**, **i**, **j**, **l**, **n**, **p**, **r**). The asterisk above the columns indicates the statistical difference between the mutant and the wild type SC5314. Source data are provided as a Source Data file.

Minor Concerns:

Strain construction: expected PCR sizes should be indicated and more details should be provided in

Supplementary Methods and Supplemental Figure Legends. As currently presented, the expected gel findings are difficult to follow. There also seems to be a typographical error: "the absence of the product of primers LY22-18 and LY22-18".

Thanks for your suggestion. We have checked and revised the description of Supplementary Methods and Supplemental Figure Legends.

Please use either "VacoRed" or "VacuRed" for consistency, not both. More details of this vacuolar probe should be provided.

Thank you for your suggestion. All "VacoRed" has been changed to "VacuRed" in our revision text. As the structure of VacuRed has not been patented, we cannot disclose the specific synthesis method and the structure of the probe in this article. If you want to know more detailed information, we will provide it to you privately. Thanks for your consideration.

The description of VacuRed has been changed to "To further image the structure of vacuoles, the probe VacuRed, a silicon-substituted coumarin-based fluorophore developed by our group^[1, 2], targeting fungal vacuoles was designed and synthesized." in line 160-161.

Ref:

1. Li, C., Wang, T., Li, N., Li, M., Li, Y., Sun, Y., Tian, Y., Zhu, J., Wu, Y., Zhang, D., and Cui, X. Hydrogen-bonding-induced bathochromic effect of Si-coumarin and its use in monitoring adipogenic differentiation. 2019, *Chem. Commun.* 55, 11802-11805.
2. Li, C., Wang, T., Fan, M., Wang, N., Lin, X., Sun, Y., and Cui, X. Hydrogen Bond-Enhanced Nanoaggregation and Antisolvatochromic Fluorescence for Protein-Recognition by Si-Coumarins. 2022, *Nano Lett.* 22, 1954-1962.

Fig. 2D, it is difficult to compare length or other morphology of the filaments in SC5314 (compared to the mutant strains), due to clumping and the relatively low magnification used. A higher magnification image would be helpful. It would also be beneficial to include a scale bar.

Thanks for your suggestion. In order to better measure the hyphal length, we used protease K to separate hyphae. The representative hyphae shown in Fig. 2D were replaced with new Fig. 2h. The images were acquired by Leica confocal microscopy and shown with scale bar.

Fig. 6 Fig. 2h in our revised manuscript. The hyphal morphology of *C. albicans* strains. *C. albicans* strains were cultured in liquid RPMI 1640 + 10% FBS at 37 °C for 4 h and stained with Calcofluor white.

Fig. 3j, please use the term “sacrificed” or “killed”, not “executed”.

Thanks for your suggestion, we have changed the expression of “executed”.

Fig. 3h, the image resolution is sub-optimal; are higher resolution images available so the hyphae can be more adequately seen?

Thanks for your suggestion. We have provided a high quality of images and zoom-in images to distinguish the morphology of hyphae in kidneys in Fig. 3h, 3m, 6f, 6j, 7m, 7q.

Fig 5. Fig 3h in our new manuscript. The hyphae in kidneys stained by PAS.

There are formatting/layout issues with the Supplemental Figures, such that it is difficult to discern which label belongs to which figure. These figures should be more clearly separated.

Thanks to your suggestions, we have reorganized the supplementary material. There is a one-to-one correspondence between the supplementary and main figures in our revised manuscript. The results related to the strain constructions and the phenotypes of revertant strains have been listed in the supplemental figures.

REVIEWERS' COMMENTS

Reviewer #1 (Remarks to the Author):

The authors carefully revised the manuscript and addressed all questions of this reviewer. One minor point should be considered. In Fig. 3 i and n the authors measured hyphal length within kidneys and the skin. While this reviewer agrees with the observed differences (PAS staining) measuring the length of hyphae in an infected organ seems questionable for 2 reasons: I) Hyphae cluster as seen in provided images, II) cutting sections may "destroy" hyphae. Thus, measuring hyphal length seems inaccurate because intact single hyphae are sparse. Furthermore, it remains unclear how many single hyphae were measured (not provided in figure legend).

Reviewer #2 (Remarks to the Author):

The authors have satisfactorily addressed all of my previous concerns with relevant experiments.

The manuscript abstract should be corrected as it wrongly states that "disruption of VAM6, VPS41, or YPT72, which are components of the homotypic vacuolar fusion and protein sorting (HOPS) complex"

Rab7 ortholog Ypt72 likely recruits HOPS complex to vacuolar membrane. It is not a component of the heterohexameric HOPS complex.

Reviewer #4 (Remarks to the Author):

The authors have substantially improved the depth and clarity of their manuscript. The additional experiments have shown a more convincing relationship between vacuolar localization, size, and ostensibly turgor pressure and tissue invasion. The figures provide additional information and are of higher technical quality.

REVIEWER COMMENTS

The authors carefully revised the manuscript and addressed all questions of this reviewer. One minor point should be considered. In Fig. 3 i and n the authors measured hyphal length within kidneys and the skin. While this reviewer agrees with the observed differences (PAS staining) measuring the length of hyphae in an infected organ seems questionable for 2 reasons: I) Hyphae cluster as seen in provided images, II) cutting sections may "destroy" hyphae. Thus, measuring hyphal length seems inaccurate because intact single hyphae are sparse. Furthermore, it remains unclear how many single hyphae were measured (not provided in figure legend).

Response: Thanks for your comments. The issues of the hyphal length in pathological sections of kidneys and skin have been considered while we were measuring, similar to the concerns raised by the reviewer, especially the difficulty in distinguishing intact single hyphae because of the hyphae cluster and inevitable destruction of cutting operation. To address these issues, we have tried to find out and measure the hyphae that dispersed and with intact structure in the fields. In such cases, the heads and tails of single hyphon were identified. However, it did not mean that we only selected longer hyphae. In fact, as we can see, even shorter hyphae may have intact tubes in the sections. We avoided selecting hyphae with obvious broken marks as much as possible, such as tapering at one or both ends, or other features of being cut. As the result of above considerations, numbers of hyphae we measured in the sections of one same sample were usually limited to 20 to 30, and the specific numbers of how many single hyphae we measured were provided in the figure legends. The large number of hyphae in the fields provided us with much more options, yet if there were fewer hyphae could be measured in one field, we also selected other fields from the same section, or even the same tissue, to measure more intact hyphae.

Reviewer #2 (Remarks to the Author):

The authors have satisfactory addressed all of my previous concerns with relevant experiments. The manuscript abstract should be corrected as it wrongly states that "disruption of VAM6, VPS41, or YPT72, which are components of the homotypic vacuolar fusion and protein sorting (HOPS) complex" Rab7 ortholog Ypt72 likely recruits HOPS complex to vacuolar membrane. It is not a component of the heterohexameric HOPS complex.

Response: Thanks for your comments. We have revised our abstract as follows, ‘Here, we found that the disruption of *VAM6* or *VPS41* which are components of the homotypic vacuolar fusion and protein sorting (HOPS) complex, or the Rab GTPase *YPT72*, all responsible for vacuole fusion, led to defects in hyphal growth in both liquid and solid media, but more pronounced on solid agar’.

Reviewer #4 (Remarks to the Author):

The authors have substantially improved the depth and clarity of their manuscript. The additional experiments have shown a more convincing relationship between vacuolar localization, size, and ostensibly turgor pressure and tissue invasion. The figures provide additional information and are of higher technical quality.

Response: Thanks for your comments.